# Lysine 68 acetylation directs MnSOD as a tetrameric detoxification complex versus a monomeric tumor promoter

Yueming Zhu[1,8], Xianghui Zou[1,2,8], Angela E. Dean[1], Joseph O' Brien[1], Yucheng Gao[1,2], Elizabeth L. Tran[1], Seong-Hoon Park[1,3], Guoxiang Liu[1], Matthew B. Kieffer [4], Haiyan Jiang[1], Melissa E. Stauffer[5], Robert Hart[6], Songhua Quan[1], Karla J.F. Satchell [4], Nobuo Horikoshi[1], Marcelo Bonini[6] & David Gius[1,2,7]

Manganese superoxide dismutase (MnSOD) functions as a tumor suppressor; however, once tumorigenesis occurs, clinical data suggest MnSOD levels correlate with more aggressive human tumors, implying a potential dual function of MnSOD in the regulation of metabolism. Here we show, using in vitro transformation and xenograft growth assays that the MnSOD-K68 acetylation (Ac) mimic mutant (MnSOD$^{K68Q}$) functions as a tumor promoter. Interestingly, in various breast cancer and primary cell types the expression of *MnSOD$^{K68Q}$* is accompanied with a change of MnSOD's stoichiometry from a known homotetramer complex to a monomeric form. Biochemical experiments using the MnSOD-K68Q Ac-mimic, or physically K68-Ac (MnSOD-K68-Ac), suggest that these monomers function as a peroxidase, distinct from the established MnSOD superoxide dismutase activity. *MnSOD$^{K68Q}$* expressing cells exhibit resistance to tamoxifen (Tam) and cells selected for Tam resistance exhibited increased K68-Ac and monomeric MnSOD. These results suggest a MnSOD-K68-Ac metabolic pathway for Tam resistance, carcinogenesis and tumor progression.

[1] Department of Radiation Oncology, Feinberg School of Medicine, Northwestern University, Chicago, IL 60611, USA. [2] Driskill Graduate Program in Life Sciences, Feinberg School of Medicine, Northwestern University, Chicago, IL 60611, USA. [3] Department of General and Applied Toxicology, Innovative Toxicology Research Center, Korea Institute of Toxicology (KIT), Daejeon 34114, Korea. [4] Department of Microbiology-Immunology, Feinberg School of Medicine, Northwestern University, Chicago, IL 60611, USA. [5] Scientific Editing Solutions, Walworth, WI 53184, USA. [6] Department of Medicine, University of Illinois at Chicago, Chicago, IL 60612, USA. [7] Department of Pharmacology, Robert H. Lurie Cancer Center, Feinberg School of Medicine, Northwestern University, Chicago, IL 60611, USA. [8]These authors contributed equally: Yueming Zhu, Xianghui Zou. Correspondence and requests for materials should be addressed to D.G. (email: david.gius@northwestern.edu)

The mitochondrial sirtuin, SIRT3, acts as a tumor suppressor (TS) protein that targets several metabolic proteins for deacetylation, including manganese superoxide dismutase (MnSOD)[1–4], to protect against metabolic damage[5]. Research from our laboratory, and others[1,6], has shown that acetylation of MnSOD disrupts normal cellular and mitochondrial metabolism, leading to a tumor-permissive phenotype, suggesting that MnSOD is an adaptive enzyme responding to cellular oxidative stress[7–9]. It has been proposed that the acetylation of MnSOD is a mechanistic link between the cellular and organismal physiology of aging, energy status, and metabolic stressors, such as reactive oxygen species (ROS), carcinogenesis, and resistance to anticancer agents[2–4,8]; however, the mechanism by which MnSOD acetylation directs these processes remains unclear.

Mammalian MnSOD is a mitochondrial matrix-localized, homotetrameric, antioxidant enzyme with four identical subunits each harboring a $Mn^{2+}$ atom[10]; the primary function of MnSOD is to scavenge superoxide generated from different metabolic processes. While multiple MnSOD acetylation sites have been identified, recent publications seem to suggest that K68 is central to the regulation of MnSOD superoxide dismutase activity[1,6,8,9,11–13]. However, the specific cell biological, biochemical, and/or physiological significance of MnSOD acetylation, and the underlying molecular mechanism regulating MnSOD detoxification activity and mitochondrial metabolism, remains to be fully determined. Thus, it has been proposed that MnSOD is a mitochondrial signaling hub that regulates how cells adapt to ROS-induced metabolic stress in addition to directing mitochondrial metabolism[14], which may play an important role in late-onset diseases[2,5].

Mice lacking Sirt3, and thus containing acetylated MnSOD (MnSOD-Ac), developed tumors[7], implying that SIRT3 may function as a tumor suppressor (TS). This raises a key biological question: what is the in vivo impact of MnSOD-Ac and how do elevated, and/or aberrant, stoichiometric levels disrupt normal mitochondrial metabolism leading to cellular damage and/or a tumor-permissive murine phenotype? Interestingly, female mice lacking Sirt3 spontaneously develop estrogen-positive (ER +), poorly differentiated, high Ki-67 mammary gland tumors that appear to be similar to human luminal B breast malignancies, which are often diagnosed in older women[2,5,7,15]. As compared to luminal A ER + breast cancers, luminal B subtypes tend to have increased proliferation markers and, most importantly, can exhibit an endocrine-resistant phenotype[5]. Interestingly, mice that have a monoallelic knockout for MnSOD ($MnSOD^{+/-}$) exhibit decreased MnSOD activity, increased oxidative stress, and decreased life span, as well as aging-related phenotypes, especially carcinogenesis[16]. This in vitro and in vivo evidence supports the possibility that there is a link between the mitochondrial acetylome, as directed by SIRT3, and ROS detoxification, mitochondrial metabolism, and carcinogenesis; however, rigorous mechanistic data supporting this intriguing idea has been limited.

In this regard, we present data showing that the acetylation status of MnSOD, specifically K68, directs ROS detoxification activity, as well as connects metabolic stress and mitochondrial reparative pathways that maintain metabolic balance. Our results show that MnSOD exists in both homotetrameric and monomeric forms, which function as a superoxide dismutase and a peroxidase, respectively. We show that the homotetramer is a TS, whereas the monomer, as modeled by enforced $MnSOD^{K68Q}$ expression, functions as a tumor promoter.

## Results

### $MnSOD^{K68Q}$ expression promotes a transformation phenotype.
MnSOD is a TS protein in vitro and in vivo[17,18], as well as in human tumor samples[19]. However, correlative findings in human tumor samples suggest that while MnSOD may function as a TS during the early stages of tumor initiation, once tumorigenesis progresses, MnSOD levels positively correlate with more aggressive human tumors[20], suggesting that specific isoforms of MnSOD, including potentially the acetylated form of MnSOD, may function as a tumor promoter. In addition, it also appears that, under specific conditions, there is a link between dysregulated MnSOD, aberrant cellular ROS levels[21–23], and resistance to tamoxifen (Tam)-induced cytotoxicity. These and other findings[24] suggest a mechanistic link between mitochondrial redox/ROS balance and the biology of ER + breast cancer.

To test this hypothesis, MnSOD K68 acetylation mimic ($MnSOD^{K68Q}$) and deacetylation mimic ($MnSOD^{K68R}$) mutants were made where the substitution of a lysine with a glutamine (Q) mimics an acetylated amino acid state, while substitution with an arginine (R) mimics deacetylation[8]. To determine if $MnSOD^{K68Q}$, a site-directed mutant that genetically mimics K68-Ac, may function as a tumor promoter, lenti-$MnSOD^{K68R}$ or lenti-$MnSOD^{K68Q}$ were co-infected into wild-type (WT) primary mouse embryonic fibroblasts (pMEFs) with lentiviral expression of either c-Myc or Ras. In these experiments, at least two oncogenes, i.e., c-Myc and Ras (WT Ras gene)[25], are required to immortalize and/or transform primary cells. pMEFs infected with lenti-$MnSOD^{K68Q}$, and either c-Myc or Ras, became immortalized (i.e., divided beyond 15 cell passages), as well as cells infected with both genes (Fig. 1a, bottom row). In contrast, infection with lenti-$MnSOD^{K68R}$ did not immortalize WT pMEFs infected with c-Myc or Ras, and interestingly, $MnSOD^{K68R}$ prevented immortalization in cells infected with both genes (Fig. 1a, middle row). As a control, pMEFs were immortalized by c-Myc and Ras together, but not with c-Myc or Ras alone (Fig. 1a, top row). In addition, pMEFs infected with lenti-$MnSOD^{K68Q}$ exhibited a more transformed in vitro phenotype as determined by growth in soft agar (Fig. 1b, top panel), a measure of anchorage-independent growth; increased colony formation when plated at low density (bottom panel), a measure of proliferative capacity; decreased doubling time, a measurement of proliferation rate (Supplementary Fig.1a, middle column); and the formation of xenograft tumors, a measure of an in vivo tumorigenic permissive phenotype (Supplementary Fig. 1a, right column).

To further characterize the link between MnSOD-Ac and its function, TS versus tumor promoter, pMEFs were co-infected with oncogenic lenti-Kras$^{G12V}$ (i.e., the oncogenic Kras gene) and lenti-MnSOD$^{WT}$, lenti-MnSOD$^{K68R}$, or lenti-MnSOD$^{K68Q}$. The pMEFs expressing $MnSOD^{K68Q}$ were immortalized (Fig. 1c, bottom row, second column), as well as exhibited a more transformed in vitro phenotype, as measured by doubling time in culture (22 versus 35 h, third column) and growth in soft agar (bottom row, right column). Interestingly, infection with lenti-MnSOD$^{K68R}$, the deacetylation mimic MnSOD mutant, prevented immortalization when co-infected with lenti-Kras$^{G12V}$ (middle row, second column). Finally, these experiments were repeated in immortalized NIH 3T3 cells, an established in vitro model, to determine in vitro transformation, and NIH 3T3 cells expressing $MnSOD^{K68Q}$ exhibited increased growth in soft agar (Fig. 1d, upper panels) and colony formation when plated at low density (bottom panels).

### MnSOD$^{K68Q}$ increases in vitro and xenograft proliferation. To determine the role of $MnSOD^{K68Q}$ expression on tumor growth properties, human mammary ER + MCF7 tumor cells infected with lenti-MnSOD$^{WT}$, lenti-MnSOD$^{K68R}$, or lenti-MnSOD$^{K68Q}$ were engrafted into nude mice. Normally, MCF7 cells will not

**Fig. 1** $MnSOD^{K68Q}$ expression promotes a transformation-permissive phenotype in vitro. **a** Immortalization, i.e., growth beyond 15 passages, of pMEFs infected with lenti-MnSOD$^{WT}$, lenti-MnSOD$^{K68R}$, and lenti-MnSOD$^{K68Q}$ and either lenti-Myc or lenti-Ras. **b** The cell lines above were tested for soft agar growth (upper) and colony formation (lower panels). **c** pMEFs infected with $Ras^{G12V}$ were tested for immortalization, doubling time, and soft agar growth. **d** NIH 3T3 cells expressing $MnSOD^{WT}$, $MnSOD^{K68R}$, and $MnSOD^{K68Q}$ were tested for growth in soft agar (upper) and colony formation (lower panels). Experiments done in triplicate. Scale bar: 20 μm

grow in nude mice without estrogen supplementation; however, MCF7 cells infected with lenti-MnSOD$^{K68Q}$ (MCF7-MnSOD$^{K68Q}$) grew tumors in nude mice without estrogen supplementation (Fig. 2a, b). In contrast, MCF7 cells infected with lenti-MnSOD$^{WT}$ (MCF7-MnSOD$^{WT}$) or lenti-MnSOD$^{K68R}$ (MCF7-MnSOD$^{K68R}$) did not form tumors (Fig. 2a, b). These results suggest increased growth characteristics in xenograft tumors that express $MnSOD^{K68Q}$; however, this could also reflect estrogen-independent growth properties. To address this, MCF7-MnSOD$^{K68Q}$ cells were injected into the hind limbs of nude mice, and these xenograft experiments showed that estrogen supplementation did not alter the tumor growth curve (Supplementary Fig. 1b).

Luminal B ER + breast cancer cells are more aggressive and display increased proliferation, as measured by Ki-67 staining, compared to luminal A cancer cells[5]. Tumors in mice lacking Sirt3, which contain MnSOD-Ac, display a luminal B-like tumor signature, including increased Ki-67[5]. Consistent with these observations, MCF7-MnSOD$^{K68Q}$ cells stained with an anti-Ki-67 antibody showed a significant increase in Ki-67 immunofluorescence (IF) staining, as compared to MCF7-MnSOD$^{K68R}$ or MCF7-MnSOD$^{WT}$ cells (Fig. 2c) and quantified using ImageJ analysis (Fig. 2d). MCF7-MnSOD$^{WT}$ cells exhibited the same Ki-67 staining as the control, non-infected MCF7 cells (Supplementary Fig. 2a). The raw data for all of the bar graph analysis and the full gel images are presented in the Source Data. In addition, experiments using a second ER + human breast cancer cell line, T47D, also showed increased Ki-67 staining for T47D-MnSOD$^{K68Q}$ as compared to T47D-MnSOD$^{K68R}$ and T47D-MnSOD$^{WT}$ cells (Fig. 2e, f). T47D-MnSOD$^{WT}$ cells exhibited the same Ki-67 staining as the control, non-infected T47D cells (Supplementary Fig. 2b). Finally, MCF7-MnSOD$^{K68Q}$ cells exhibited similar Ki-67 staining when exposed to either estrogen (Supplementary Fig. 2c, e) or Tam (Supplementary Fig. 2d, f).

**MnSOD$^{K68Q}$ is a monomer that exhibits peroxidase activity.** MnSOD consists of four subunits that form a homotetramer, each binds to a manganese ion (~88 kDa)[3,10]. To determine if the acetylation status of K68 alters the conformation of MnSOD, as well as its activity, MCF7-MnSOD$^{WT}$, MCF7-MnSOD$^{K68R}$, and

MCF7-MnSOD$^{K68Q}$ cells were harvested and cell lysates were crosslinked with glutaraldehyde, followed by sodium dodecyl sulfate polyacrylamide gel electrophoresis (SDS-PAGE) and immunoblotting with an anti-MnSOD antibody. These experiments showed that cells expressing $MnSOD^{K68Q}$ exhibited a significant decrease in the tetrameric form of MnSOD (Fig. 3a, left panel), with a slight increase in both the dimeric and monomeric forms. Similar results were also observed in T47D-MnSOD$^{K68Q}$ cells (right panel). In addition, these experiments were repeated in MCF7 and T47D cells infected with lenti-shSIRT3, and these cells also showed a significant decrease in the tetrameric form of MnSOD (Fig. 3b), in contrast to little or no change in the oligomerization status of MnSOD in the control cells.

It has been previously shown that MnSOD, under specific conditions and when significantly overexpressed, can exhibit peroxidase activity[26]; however, the mechanism to explain this intriguing observation is unknown. In this regard, a peroxidase assay showed that MnSOD$^{K68Q}$ mutant Immunoprecipitated (IPed) from MCF7 cells functions as a peroxidase (Fig. 3c). In contrast, IPed MnSOD$^{K68R}$ exhibited significantly less peroxidase activity (i.e., a roughly 50-fold difference), suggesting that this peroxidase activity may require acetylation of K68 (Fig. 3c). Finally, experiments confirmed that the MnSOD$^{-/-}$ MEFs expressing MnSOD$^{K68Q}$ exhibited a significant decrease in tetrameric MnSOD, as compared to cells expressing $MnSOD^{WT}$ or $MnSOD^{K68R}$ (Fig. 3d).

In the experiments above, we showed that $MnSOD^{K68Q}$, which is a genetic mutant that functions as a mimic for K68-Ac, leads to a tumor promoting phenotype (Fig. 1a, b). To more rigorously address this idea, immortalized, but not transformed, MnSOD$^{-/-}$ MEFs were infected with the MnSOD site-directed mutants described above. These experiments showed that MnSOD$^{-/-}$ MEFs infected with only lenti-MnSOD$^{K68Q}$, (i.e., a single gene) exhibited a more transformed phenotype (Fig. 3e, middle column), as compared to cells infected with lenti-empty vector, lenti-MnSOD$^{WT}$, or lenti-MnSOD$^{K68R}$ (Supplementary Fig. 3a, as measured by growth in soft agar, contact inhibition, and doubling time (Supplementary Fig. 3b–d). Since these results were done in cells lacking MnSOD, it seems reasonable to suggest that MnSOD$^{K68Q}$ functions as an in vitro tumor promoter.

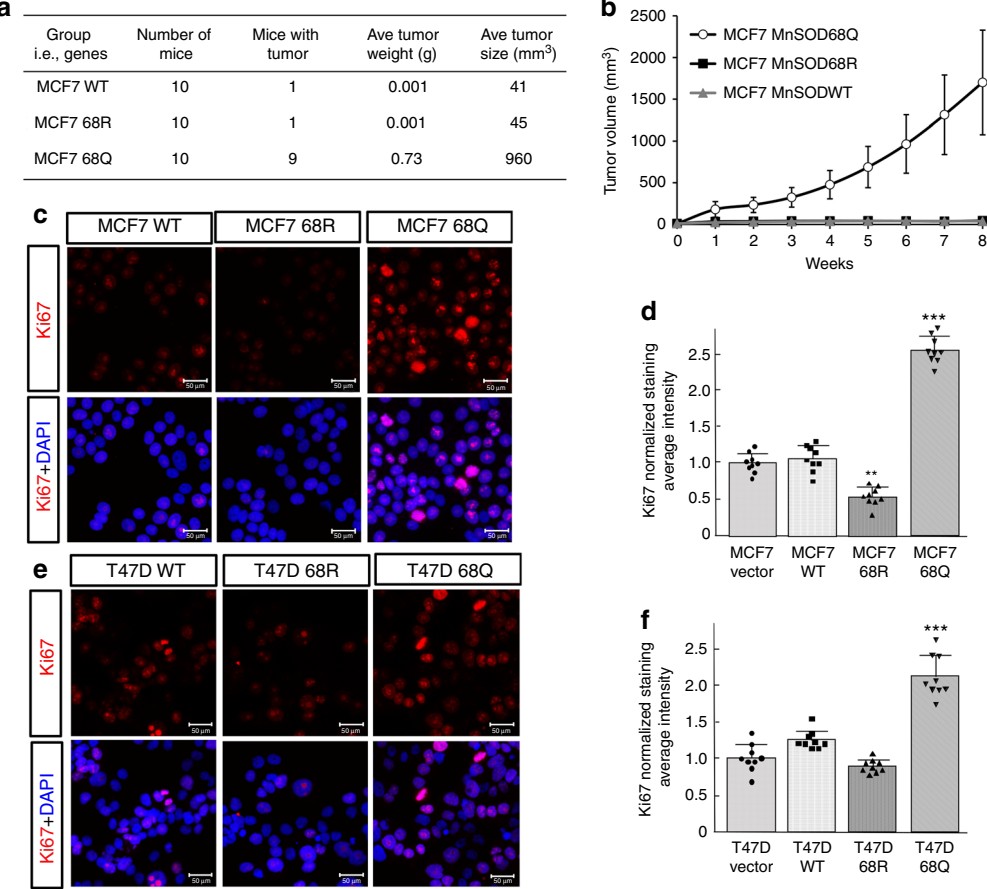

**Fig. 2** $MnSOD^{K68Q}$ expression increases xenograft tumor growth and in vitro proliferation. **a**, **b** MCF7 cells expressing $MnSOD^{WT}$, $MnSOD^{K68R}$, and $MnSOD^{K68Q}$ were implanted into the hind limb of nude mice ($n = 10$ per group) and tested for xenograft tumor growth. **c**, **d** MCF7 cells expressing $MnSOD^{WT}$, $MnSOD^{K68R}$, and $MnSOD^{K68Q}$ were **c** IF stained for Ki-67 and DAPI, and **d** quantified for Ki-67 intensity as determined by ImageJ analysis. **e**, **f** T47D cells expressing $MnSOD^{WT}$, $MnSOD^{K68R}$, and $MnSOD^{K68Q}$ were **e** IF stained for Ki-67 and DAPI, and **f** quantified for Ki-67 intensity. All experiments were done in triplicate. Error bars represent ±1 SEM. A one-way ANOVA analysis with Tukey's post-analysis was used. **p < 0.01 and ***p < 0.001

Finally, if MnSOD$^{K68Q}$ acts as a peroxidase, then removing cellular hydrogen peroxide, which is a necessary and required substrate for peroxidase activity, might prevent its ability to function as a peroxidase, as well as a tumor promoter. In this regard, co-infection with lenti-MnSOD$^{K68R}$ and AdMitoCat, which expresses catalase and decreases cellular hydrogen peroxide, prevented transformation (Fig. 3e, right column). Since we have shown that immortalized MnSOD$^{−/−}$ MEFs can be transformed by infection with $MnSOD^{K68Q}$, these results suggest that $MnSOD^{K68Q}$, which enriches for monomeric MnSOD, is potentially an in vitro tumor promoter that requires hydrogen peroxide.

**MnSOD-K68-Ac exhibits peroxidase activity**. The data presented above showed enrichment of monomeric MnSOD (Fig. 3a) and peroxidase activity (Fig. 3c) upon expression of $MnSOD^{K68Q}$ to mimic K68-Ac. However, it is also essential to show how the physical acetylation of K68 affects enzymatic activity. To initially address this issue, an established tissue culture system was used that enriches for acetylated, versus deacetylated, K68. In this system, transfected MnSOD$^{−/−}$ MEFs with $FLAG\text{-}MnSOD^{WT}$ were followed by the exposure to (i) 10 mM nicotinamide (NAM) and 1 μM trichostatin A (TSA), to inhibit SIRT3 deacetylase activity and enrich for K68-Ac, or (ii) 10 mM NAD +, to activate SIRT3 activity and enrich for deacetylated K68. As expected, whole-cell extracts harvested 40 h after transfection and IPed with

an anti-FLAG antibody showed that NAM/TSA exposure increased MnSOD-K68-Ac (Fig. 4a, top row, left two lanes), while NAD + exposure minimized MnSOD-K68-Ac (right two lanes). Similar results were observed in 293T cells (Supplementary Fig. 4a). The MnSOD-K68-Ac antibody specificity (Abcam, Inc, ab137037) was validated by two different methods[12].

These samples were subsequently separated, using a spin column, into fractions above or below 50 kDa. Immunoblotting with an anti-MnSOD antibody, the sample from cells grown in NAM/TSA showed an enrichment of MnSOD in the < 50 kDa fraction, suggesting that most of the MnSOD is in the monomeric form, with minimal MnSOD in the > 50 kDa fraction (Fig. 4a, 2nd and 3rd row, left two lanes). The enrichment of the monomeric MnSOD was confirmed when the < 50 kDa fraction was run on a semi-native gel followed by immunoblotting for MnSOD (Supplementary Fig. 4b, left two lanes) with minimal tetrameric MnSOD in the > 50 kDa fraction (right panel, left two lanes). In contrast, MnSOD from cells grown in NAD + showed increased levels of MnSOD in the > 50 kDa fraction (Fig. 4a, 2nd and 3rd row, right two lanes), with enrichment of tetrameric MnSOD in the > 50 kDa fraction (Supplementary Fig. 4b, right panel, right two lanes). These experiments confirm that samples enriched for MnSOD-K68-Ac contain predominantly monomeric MnSOD, and those with deacetylated MnSOD-K68 contain predominantly tetrameric MnSOD.

Biochemical analysis of the < 50 kDa fraction from cells exposed to NAM/TSA (i.e., enriched for MnSOD-K68-Ac and

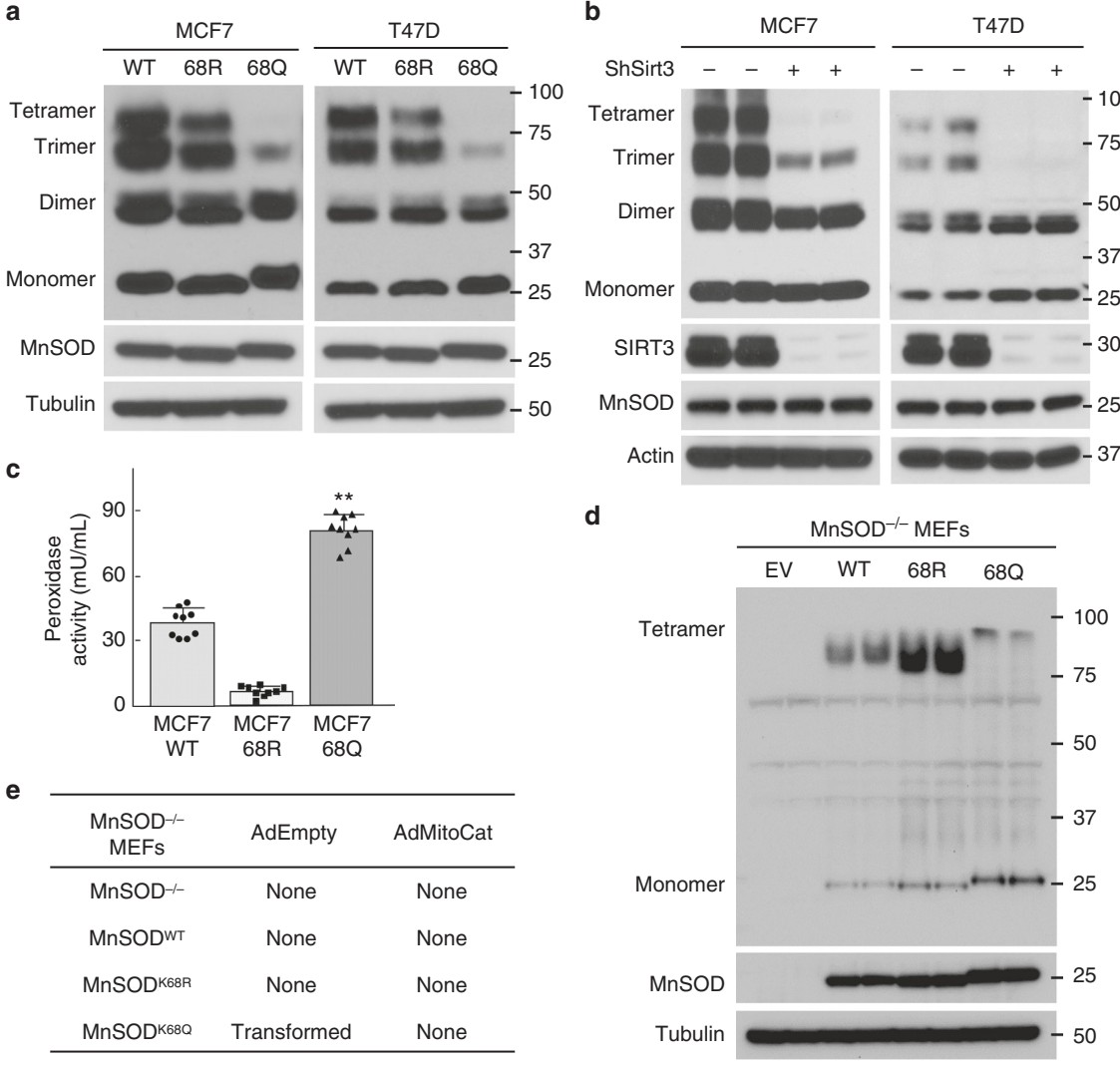

**Fig. 3** MnSOD-K68Q alters MnSOD conformation and exhibits peroxidase activity. **a** MCF7 (left panel) and T47D (right panel) cells expressing *MnSOD*[WT], *MnSOD*[K68R], or *MnSOD*[K68Q] were analyzed by semi-native crosslinking and blotting with an anti-MnSOD antibody. **b** MCF7 (left) and T47D (right) cells expressing *shSIRT3* were analyzed by crosslinking. **c** Flag-MnSOD[WT], Flag-MnSOD[K68R], and Flag-MnSOD[K68Q] expressed in MCF7 cells were measured for peroxidase activity. Error bars represent ± 1 SEM. **\*\****p* < 0.01. **d** Immortalized MnSOD[−/−] pMEFs expressing *MnSOD*[WT], *MnSOD*[K68R], or *MnSOD*[K68Q] were analyzed by semi-native crosslinking and immunoblotted with an anti-MnSOD antibody. **e** MnSOD[−/−] pMEFs expressing *MnSOD*[WT], *MnSOD*[K68R], or *MnSOD*[K68Q], without or with Ad-Mito-Cat or Ad-Empty, were measured for transformation. All experiments were done in triplicate. A one-way ANOVA statistical analysis with Tukey's post-analysis was used

monomeric MnSOD) showed elevated peroxidase activity compared to the < 50 kDa fraction from cells treated with NAD + (Fig. 4b). In contrast, MnSOD from both < 50 kDa fractions exhibited minimal MnSOD detoxification activity (Fig. 4c). Analysis of the > 50 kDa fraction from cells treated with NAD + (i.e., enriched for tetrameric MnSOD) exhibited elevated MnSOD detoxification activity compared to cells exposed to NAM/TSA (Fig. 4d). As expected, there was little MnSOD peroxidase activity in the > 50 kDa fraction from cells treated with either NAD + or NAM/TSA (Supplementary Fig. 4c).

A second, and more rigorous, method was also used to determine how the physical acetylation of MnSOD-K68 alters enzymatic activity. Recombinant MnSOD-K68-Ac was produced in *E. coli* transformed with both pEVOL-AcKRS, which expresses an acetyl-lysyl-tRNA synthetase/tRNA[CUA] pair from *M. barkeri*, and pET21a-MnSOD[K68TAG], a MnSOD bacterial expression vector that allows the site-specific incorporation of N-(ϵ)-acetyl-l-lysine into K68. The bacterially expressed proteins from

the control (carrying pET21a-MnSOD[WT]) and acetylated form (carrying pET21a-MnSOD[K68TAG]) were purified by nickel affinity columns followed by size exclusion chromatography (SEC)[13,27,28]. Purified wild-type MnSOD from bacteria eluted at a volume roughly corresponding to 92 kDa (Fig. 4e, peak 1/blue peak) on SEC consistent with the size of its known homotetrameric complex (Supplementary Fig. 5a, full chromatogram), as shown by others (Knyphausen et al., 2016)[13]. Purified MnSOD-K68-Ac from bacteria carrying pEVOL-AcKRS and pET21a-MnSOD[K68TAG] eluted at a volume consistent with the monomeric form of MnSOD (Fig. 4e, peak 2/red peak) roughly corresponding to 25 kDa (Supplementary Fig. 5b, full chromatogram).

Prior to further analysis, two eluted fractions corresponding to peak 1 (elution volumes 13 and 14 mL) and peak 2 (elution volumes 16 and 17 mL) were analyzed to confirm MnSOD. Immunoblotting (Fig. 4f, top panel) and Coomassie staining (bottom panel) for purified wild-type bacterial expressed protein

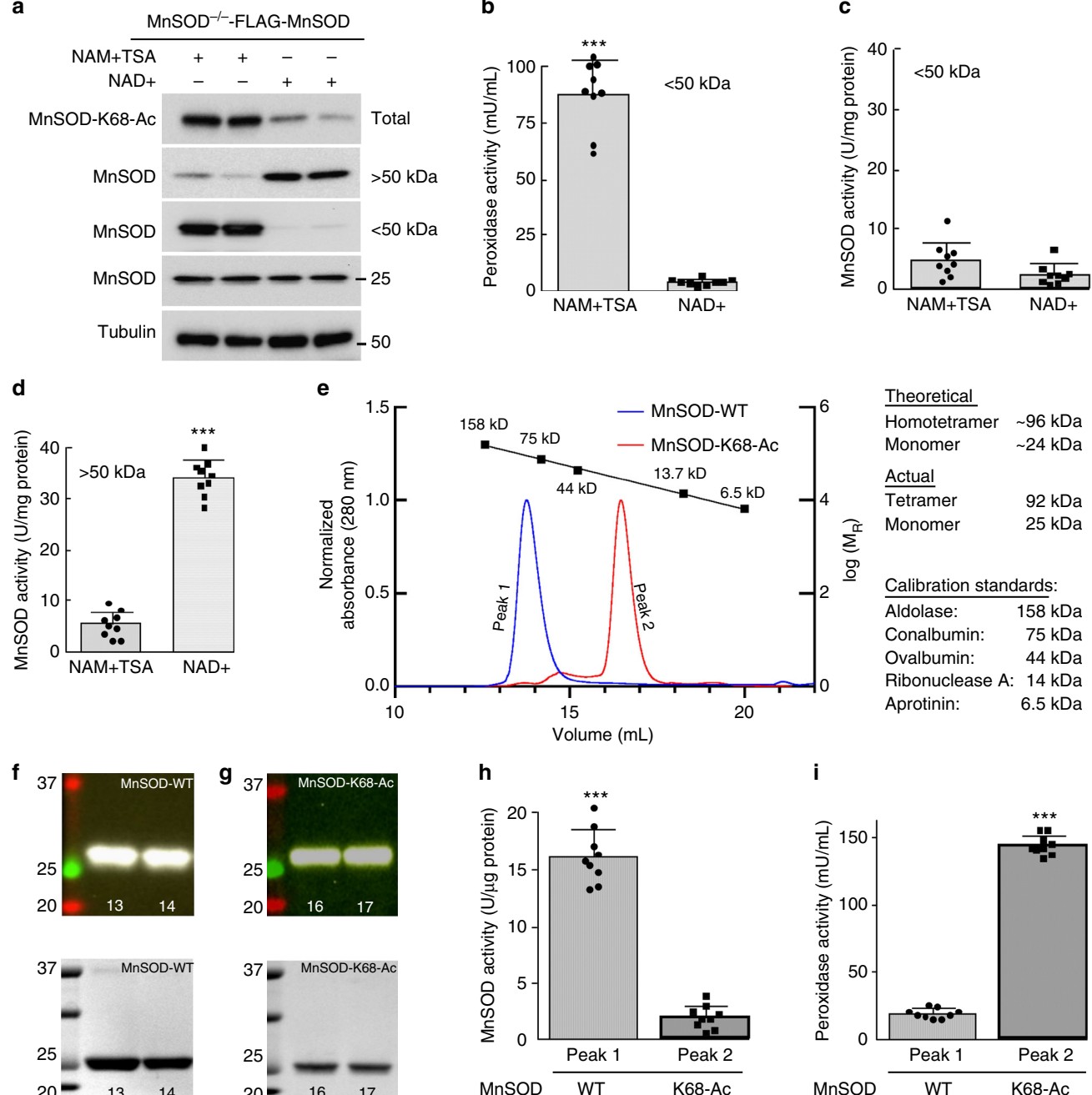

**Fig. 4** The physical acetylation of MnSOD-K68 produces peroxidase activity. **a–d** Immortalized MnSOD$^{-/-}$ MEFs expressing *Flag-MnSOD*$^{WT}$ were cultured in NAM + TSA or NAD +, separated using a 50 kDa molecular cutoff membrane and **a** MnSOD-K68-Ac, MnSOD, and actin immunoreactive protein levels were determined. **b** Peroxidase activity and **c** MnSOD activity in < 50 kDa fractions. **d** The > 50 kDa fractions were analyzed for MnSOD activity. **e** Bacterially produced and purified recombinant MnSOD-WT and MnSOD-K68-Ac proteins were characterized by size exclusion column chromatography. Standards are shown. **f**, **g** Elution volumes 13 and 14 mL, corresponding to peak 1/blue (**f**) and elution volumes 16 and 17 mL, corresponding to peak 2/red (**g**) from Fig. 4e were analyzed for MnSOD by MnSOD-K68-Ac immunoblotting (top panels) or Coomassie Brilliant Blue staining (bottom panels). **h**, **i** Peak 1 (elution volumes 13 and 14 mL) and peak 2 (elution volumes 16 and 17 mL) were analyzed for **h** superoxide dismutase activity **i** and peroxidase activity. All experiments were done in triplicate. Errors represent ±1 SEM. ***$p$ < 0.01. A *t*-test was used to compare means of the two groups

confirmed the presence of MnSOD. Similar experiments also confirmed the presence of MnSOD in bacteria carrying pET21a-MnSOD$^{K68TAG}$ (Fig. 4g, top and bottom panel). These samples were also analyzed via mass spectrometry (Supplementary Fig. 5c–e) and by staining with the anti-MnSOD-K68-Ac antibody (Supplementary Fig. 5f) confirming that peak 2 is enriched for MnSOD-K68-Ac protein.

Purified protein samples from the bacteria cells expressing pET21a-MnSOD$^{WT}$ (elution volumes 13 and 14 mL) showed significant superoxide dismutase activity (Fig. 4h, left bar) with minimal peroxidase activity (Fig. 4i, left bar). In contrast, recombinant MnSOD-K68-Ac protein from bacterial cells expressing pET21a-MnSOD$^{K68TAG}$ (elution volumes 16 and 17 mL) exhibited minimal superoxide activity (Fig. 4h, right

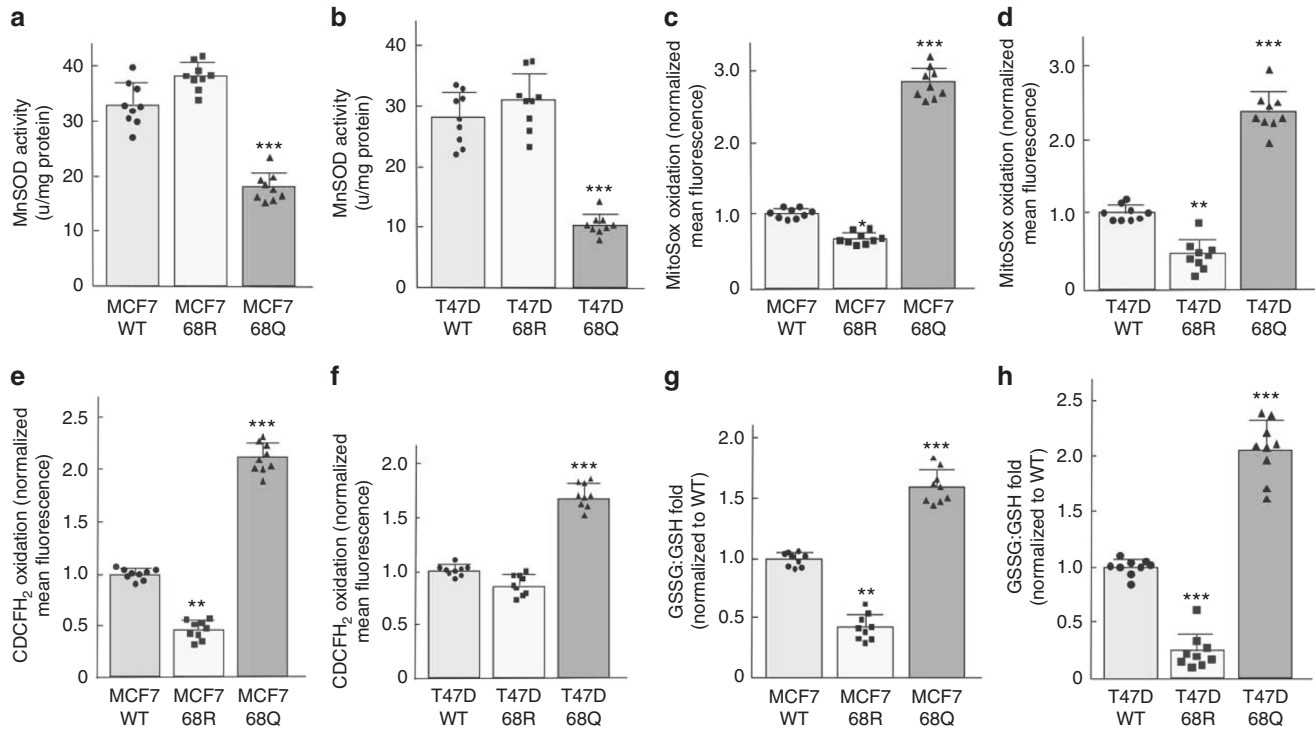

**Fig. 5** $MnSOD^{K68Q}$ expression leads to oxidative stress in human breast cells. **a**, **b** MnSOD activity: **a** MCF7-MnSOD$^{WT}$, MCF7-MnSOD$^{K68R}$, and MCF7-MnSOD$^{K68Q}$; and **b** T47D-MnSOD$^{WT}$, T47D-MnSOD$^{K68R}$, and T47D-MnSOD$^{K68Q}$ in whole-cell homogenates. **c**, **d** Steady-state levels of $O_2^{\bullet-}$ were measured in **c** MCF7-MnSOD$^{WT}$, MCF7-MnSOD$^{K68R}$, and MCF7-MnSOD$^{K68Q}$ and **d** T47D-MnSOD$^{WT}$, T47D-MnSOD$^{K68R}$, and T47D-MnSOD$^{K68Q}$ cells. **e**, **f** $H_2O_2$ levels was measured in these cells by CDCFH$_2$ oxidation via flow cytometry. **g**, **h** Glutathione levels were measured in whole-cell homogenates of these cells. All experiments were done in triplicate. Error bars represent ± 1 SEM. *$p < 0.05$, **$p < 0.01$, and ***$p < 0.001$. A one-way ANOVA statistical analysis with Tukey's post-analysis was used

bar) and significant peroxidase activity (Fig. 4i, right bar). These biochemical results show two different methods to isolate MnSOD where K68 is either physically acetylated (bacterial expression system) or enriched for K68 acetylation (transfection expression system) to confirm a switch to monomeric MnSOD and exhibits a peroxidase enzymatic function. However, more research is required to definitively identify that monomerization is the underlying molecular mechanism for the enzymatic switch to peroxidase activity.

**MnSOD-K68-Ac increases oxidative stress in breast cells**. The MCF7-MnSOD$^{K68Q}$ (Fig. 5a) and T47D-MnSOD$^{K68Q}$ (Fig. 5b) cells, which constitutively express $MnSOD^{K68Q}$, also exhibited a significant decrease in MnSOD superoxide detoxification activity, consistent with that shown by others[6,12]. Since the primary function of MnSOD is to detoxify mitochondrial superoxide ($O_2^{\bullet-}$), the mitochondrial oxidation/reduction status was measured in MCF7 and T47D cells expressing the various MnSOD acetylation mutants. Among these cell lines, MCF7-MnSOD$^{K68Q}$ and T47D-MnSOD$^{K68Q}$ cells exhibited a significant increase in: (1) MitoSox oxidation, a measure of mitochondrial $O_2^{\bullet-}$ (Fig. 5c, d); (2) CDCFH$_2$ oxidation, a measure of cellular hydroperoxide levels (Fig. 5e, f); and (3) GSSG/GSH ratio, a measure of cellular oxidative stress (Fig. 5g, h), as compared to the MCF7-MnSOD$^{K68R}$, T47D-MnSOD$^{K68R}$, and control cell lines.

**Tumor cells expressing $MnSOD^{K68Q}$ exhibit Tam resistance**. It has previously been shown that there is a link between dysregulated MnSOD[29–31] and aberrant cellular ROS levels and/or oxidative stress, due to several different mechanisms[23,32], and resistance to endocrine therapy. Based on these previous

publications, and our results above identifying $MnSOD^{K68Q}$ as an in vitro tumor promoter, it seemed reasonable to propose that, similar to other oncogenes, enforced expression of $MnSOD^{K68Q}$ may also lead to, either indirectly or directly, resistance to Tam.

To test this idea, MCF7, MCF7-MnSOD$^{K68R}$, and MCF-MnSOD$^{K68Q}$ cells were treated with 1 µM hydroxy-Tam for 5 days, and clonogenic survival assays were performed. The results of these experiments showed that MCF7 (Fig. 6a) and T47D (Supplementary Fig. 6a) cells constitutively expressing $MnSOD^{K68Q}$ exhibited significant resistance to the cytotoxicity of hydroxy-Tam, as compared to cells expressing $MnSOD^{K68WT}$ or $MnSOD^{K68R}$. In addition, MCF7 (Fig. 6b) and T47D (Supplementary Fig. 6b) cells expressing shSIRT3, which results in increased cellular MnSOD-K68-Ac, also exhibited resistance to hydroxy-Tam cytotoxicity. These experiments suggest a potential link, either directly or indirectly, between $MnSOD^{K68Q}$ expression and hydroxy-Tam-resistant tumor cells. These results also add to the literature implicating the role of the MnSOD pathway[29–31], as well as ROS levels[23,32], in Tam resistance.

**Tam-resistant breast cells exhibit a MnSOD-K68-Ac signature**. Since breast cancer cells expressing $MnSOD^{K68Q}$ exhibited resistance in vitro to Tam-induced cytotoxicity, it seems plausible that MCF7 cells selected for resistance to hydroxy-Tam might also display a MnSOD-K68-Ac signature. To address this idea, MCF7 (Fig. 6c) and T47D (Supplementary Fig. 6c) cells were cultured in the presence of 1 µM hydroxy-Tam for 3 months to generate hydroxy-Tam-resistant (HTR) cells. Both MCF7-HTR and T47D-HTR cells showed an increase in MnSOD-K68-Ac (Fig. 6d, e). In addition, staining with antibodies for several other SIRT3 deacetylation targets (MnSOD-K122-Ac, IDH2-K413-Ac, and OSCP-K139-Ac), which are a proxy for SIRT3 activity, also

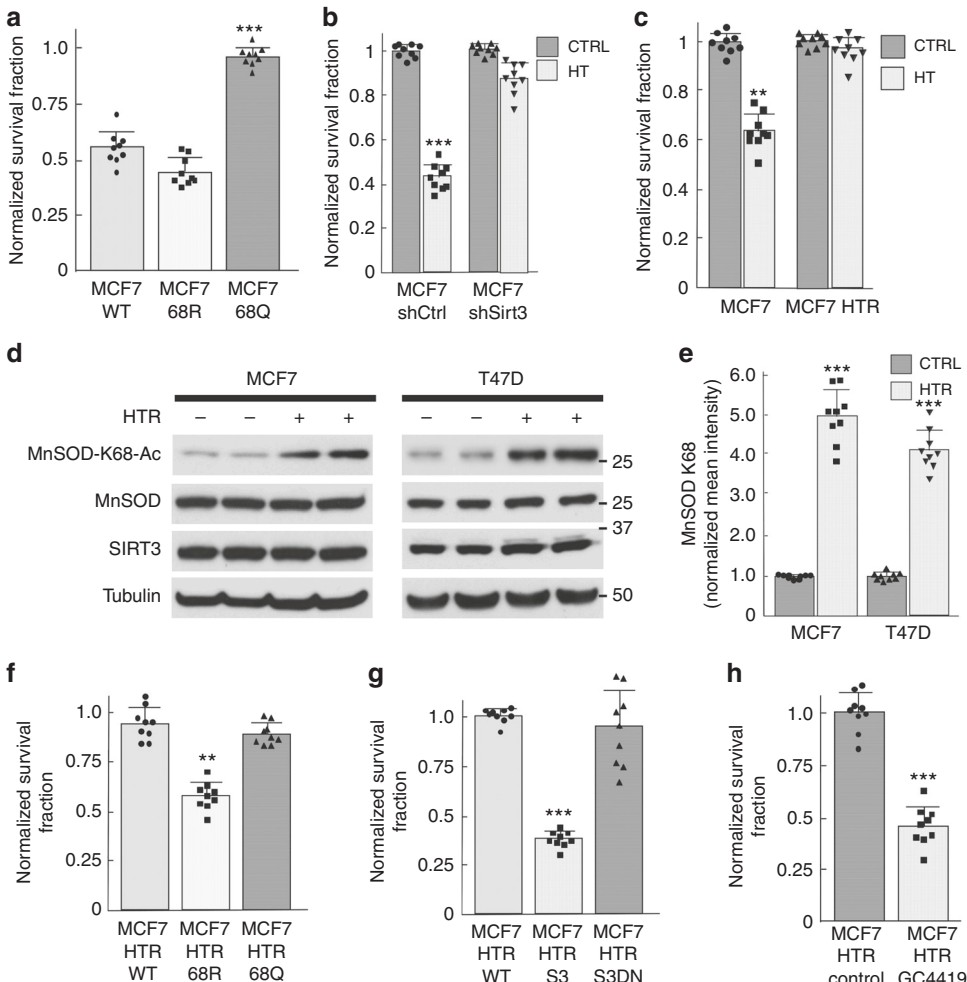

**Fig. 6** Hydroxy-Tam-resistant breast cancer cells exhibit a MnSOD-K68-Ac signature. **a–c** Clonogenic cell survival experiments for: **a** MCF7-MnSOD$^{WT}$, MCF7-MnSOD$^{K68R}$, and MCF7-MnSOD$^{K68Q}$ cells; **b** MCF7-shCtrl and MCF7-shSIRT3 cells; and **c** MCF and MCF7-HTR cells, with and without exposure to 1 μM hydroxy-Tam for 120 h (HT), as measured by cytotoxicity. **d**, **e** MCF7 and MCF7-HTR, as well as T47D and T47D-HTR cell lysates, immunoblotted for MnSOD-K68-Ac, MnSOD, SIRT3, and actin; and **e** Immunoreactive protein levels were quantified. **f–h** Clonogenic cell survival experiments for MCF7-HTR cells. **f** Cells expressing *MnSOD$^{WT}$*, *MnSOD$^{K68Q}$*, or *MnSOD$^{K68R}$* were treated with 1 μM hydroxy-Tam for 120 h. **g** Cells expressing *SIRT3$^{WT}$* or *SIRT3$^{DN}$* (S3DN; deacetylation-null *SIRT3* gene) were treated with 1 μM hydroxy-Tam. **h** Cells were treated with 5 μM GC4419 for 5 days. All experiments were done in triplicate. Error bars represent ± 1 SEM. **\*\***$p < 0.01$, and **\*\*\***$p < 0.001$. Three groups were analyzed via a one-way ANOVA statistical analysis with Tukey's post-analysis, and two groups were analyzed by a *t*-test

showed increased acetylation (Supplementary Fig. 6d), suggesting decreased SIRT3 activity. The results of these experiments suggest that ER + breast cancer cell lines selected for resistance to Tam exhibit a MnSOD-K68-Ac signature, which may also serve as a potential molecular biomarker.

**Tam resistance is reversed by *MnSOD$^{K68R}$* expression**. To further show that MnSOD-K68-Ac is a potential marker of Tam resistance, HTR cells were infected with lenti-MnSOD$^{WT}$, lenti-MnSOD$^{K68Q}$, and lenti-MnSOD$^{K68R}$, and hydroxy-Tam resistance was measured by clonogenic cell survival assays. The results showed that infection with lenti-MnSOD$^{K68R}$, but not with lenti-MnSOD$^{WT}$ or lenti-MnSOD$^{K68Q}$, reversed the hydroxy-Tam resistance (Fig. 6f and Supplementary Fig. 6e). Furthermore, when MCF7-HTR and T47D-HTR cells were infected with lenti-SIRT3$^{WT}$ (Fig. 6g and Supplementary Fig. 6f), which will result in MnSOD deacetylation, or treated with 5 μM GC4419 (Fig. 6h and Supplementary Fig. 6g), a compound that chemically removes superoxide versus the catalytic mechanism used by homo-tetrameric MnSOD, they became sensitive to hydroxy-Tam.

These results suggest that MnSOD-K68-Ac is a potential molecular biomarker and/or tumor signature for resistance to Tam.

**Tam exposure increases oxidative stress and monomeric MnSOD**. MnSOD activity is tightly correlated with mitochondrial metabolism, and HTR cells exhibit a MnSOD-K68-Ac signature (Fig. 6d, e); thus it seemed reasonable to determine the mitochondrial metabolic profile in HTR ER + cells. In this regard, MCF7-HTR (Fig. 7a) and T47D-HTR (Supplementary Fig. 7a) cells exhibited a decrease in MnSOD activity, an increase in mitochondrial O$_2$•$^-$ levels (Fig. 7b), and increased cellular hydroperoxide, as measured by CDCFH$_2$ oxidation (Fig. 7c), as well as an increase in the GSSG/GSH ratio (Fig. 7d and Supplementary Fig. 7b). Finally, monomeric MnSOD is enriched in MCF7-HTR and T47D-HTR cells (Fig. 7e), which is consistent with the decrease in MnSOD activity (Fig. 7a) and increase in MnSOD-K68-Ac (Fig. 6d), as compared to control MCF7 and T47D cells.

Furthermore, to determine if this increased oxidative stress in HTR cells is due to the acetylation status of MnSOD-K68, MCF7-

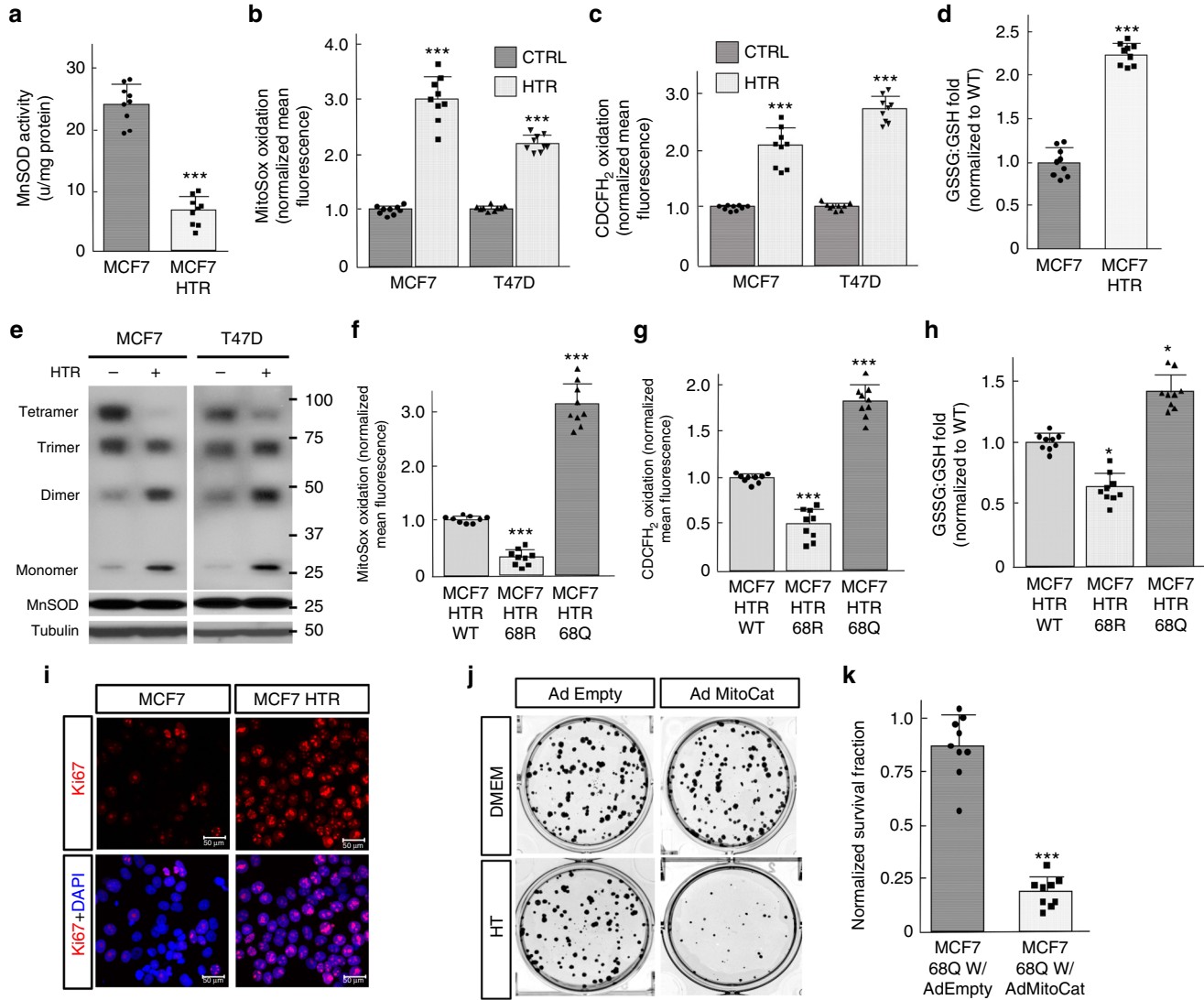

**Fig. 7** Hydroxy-Tam exposure increases oxidative stress. **a** MCF7 and MCF7-HTR whole-cell lysates were used for analysis of total MnSOD activity. **b**, **c** MCF7 and MCF7-HTR, and T47D and T47D-HTR whole-cell homogenates, were used to determine: **b** steady-state levels of $O_2^{\bullet-}$, by MitoSox oxidation; and **c** $H_2O_2$ by CDCFH$_2$ oxidation. **d** Glutathione levels in MCF7 and MCF7-HTR whole-cell homogenates. **e** Semi-native gel analysis of MCF7 and MCF7-HTR, as well as T47D and T47D-HTR cell lysates. **f**–**h** Whole-cell homogenates of MCF7-HTR cells expressing $MnSOD^{WT}$, $MnSOD^{K68Q}$, or $MnSOD^{K68R}$ were analyzed for: **f** steady-state levels of $O_2^{\bullet-}$, **g** $H_2O_2$, and **h** glutathione levels. **i** MCF7 and MCF7-HTR cells were stained for Ki-67 and DAPI. **j**, **k** Clonogenic survival experiments for MCF7-MnSOD$^{K68Q}$ cells expressing AdMitoCat. Cell were treated with 1 μM hydroxy-Tam for 120 h and **k** quantified. All experiments were done in triplicate. Error bars represent ± 1 SEM. *$p < 0.05$ and ***$p < 0.001$. Three groups analyzed via a one-way ANOVA statistical analysis with Tukey's post-analysis and two groups analyzed by a $t$-test

HTR and T47D-HTR cells were infected with lenti-MnSOD$^{WT}$, lenti-MnSOD$^{K68R}$, and lenti-MnSOD$^{K68Q}$. These experiments showed that enforced expression of $MnSOD^{K68R}$ reversed the increase in mitochondrial $O_2^{\bullet-}$ (Fig. 7f and Supplementary Fig. 7c), intracellular hydroperoxide (Fig. 7g and Supplementary Fig. 7d), and GSSG/GSH ratio (Fig. 7h and Supplementary Fig. 7e), as compared to cells expressing $MnSOD^{K68Q}$ or $MnSOD^{WT}$. These data show that HTR increases MnSOD-K68-Ac, suggesting that there may be a Tam resistance tumor signature that also includes changes in cellular ROS profiles, which has been shown by others[23,30].

**Tam-resistant MCF7 and T47D cells exhibit increased Ki-67.**
MCF7-HTR (Fig. 7i and Supplementary Fig. 8a) and T47D-HTR cells (Supplementary Fig. 8b, c), which display a MnSOD-K68-Ac signature (Fig. 6d), exhibit increased Ki-67 levels, similar to

luminal B breast malignancies, similar to MCF7-MnSOD$^{K68Q}$ and T47D-MnSOD$^{K68Q}$ cells (Fig. 2c, d). In addition, MCF7-HTR (Supplementary Fig. 8d, e)) and T47D-HTR (Supplementary Fig. 8f, g) cells treated with GC4419 or hydroxy-Tam and GC4419 exhibited decreased Ki-67 IHC staining. GC4419 or hydroxy-Tam and GC4419 also reversed the increase in Ki-67 IHC staining in MCF7-MnSOD$^{K68Q}$ (Supplementary Fig. 9a, b) and T47D-MnSOD$^{K68Q}$ (Supplementary Fig. 9c, d) cell lines, suggesting that chemically replacing the SOD activity of MnSOD reverses the increase in Ki-67.

To determine if hydrogen peroxide is necessary for the HTR observed in the MCF7-MnSOD$^{K68Q}$ cells, we infected these cells with AdMitoCat, which removed and/or significantly reduced mitochondrial hydrogen peroxide levels, a critical and necessary substrate for peroxidase enzymatic activity. The results of clonogenic cell survival experiments demonstrated that decreased mitochondrial hydrogen peroxide levels reversed the HTR

observed in MCF7 cells that constitutively express MnSOD$^{K68Q}$ (Fig. 7j, k). These results suggest that, either indirectly or directly, cells expressing the MnSOD acetylation mutant require hydrogen peroxide to maintain resistance to Tam.

**Tam-resistant xenografts exhibit a more aggressive phenotype.** To test if MCF7-HTR cells, which exhibit a MnSOD-K68-Ac signature (Fig. 6d), form more aggressive in vivo xenograft tumors, MCF7 and MCF7-HTR cells were injected into immunodeficient mice, and tumor growth was monitored. Without estrogen supplementation, control MCF7 cells were not able to form tumors in vivo, as expected. In contrast, MCF7-HTR cells formed tumors averaging 859 mm$^3$ in 6 weeks without estrogen supplementation (Fig. 8a, b), and xenograft engraftment was 100% (Supplementary Fig. 9e), indicating that these cells exhibited a highly tumorigenic phenotype. Finally, the MCF7-HTR cells were used to construct a Tet-On expression system for the inducible expression of the deacetylation mimic mutant (MnSOD$^{K68R}$). As such, MCF7-HTR cells were initially infected

with pTet-DualOn (Clontech) and selected with puromycin, followed by infection with pTre-Dual2-MnSOD$^{K68R}$ and hygromycin selection, and finally, these cells were validated for MnSOD$^{K68R}$ Tet-induction (Supplementary Fig. 10a, b). MCF7-HTR-Dual2-MnSOD$^{K68R}$ xenografts were grown to 100 mm, and mice were exposed to doxycycline to induce MnSOD$^{K68R}$ expression. These experiments showed that enforced expression of MnSOD$^{K68R}$ inhibited in vivo MCF7-HTR xenograft tumor cell growth (Fig. 8c).

**Human luminal B tumors exhibit high levels of MnSOD-K68-Ac.** Mice lacking *Sirt3* develop mammary tumors with a luminal B-like phenotype that are ER +, poorly differentiated, and display high levels of Ki-67[5,7,33]. To determine if there is a subgroup of human ER + tumors that display a loss of SIRT3/MnSOD-Ac signature, we analyzed breast cancer patient tissue microarray (TMA) slides containing all four subtypes of breast malignancies. The TMA was stained using anti-MnSOD-K68-Ac (see (Supplementary Fig. 4a, b for antibody specificity) and anti-SIRT3

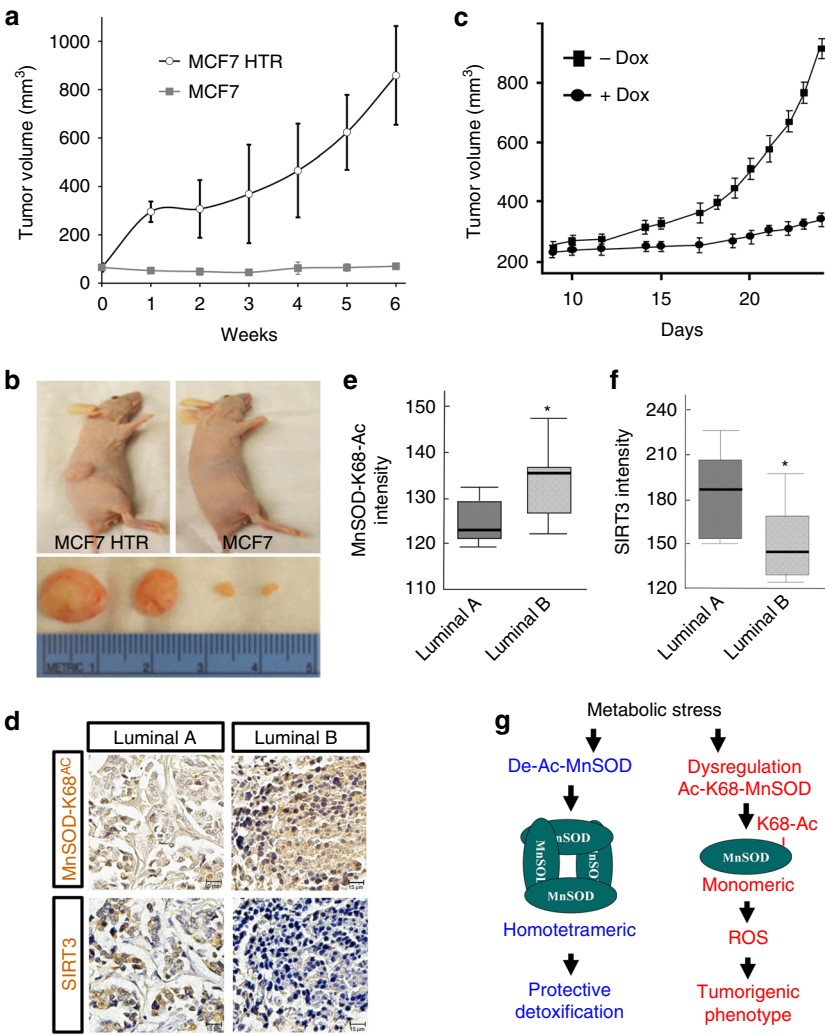

**Fig. 8** Luminal B human breast tumors exhibit a SIRT3/MnSOD-K68-Ac signature. **a, b** MCF7 and MCF7-HTR cells (5.0 × 10$^6$) were implanted into both hind limbs of nude mice and tumor volumes were measured for 6 weeks. **a** Error bars represent ±1 SEM. **b** Representative images of the tumors from MCF7-HTR (left panel) and MCF7 cells (right panel) at 6 weeks. **c** MCF7-HTR doxycycline-inducible MnSOD$^{K68R}$ cells were implanted into hind limbs of nude mice and tumor volumes were monitored for 4 weeks. Error bars represent ±1 SEM. **d** Luminal breast cancer sample TMA stained with anti-MnSOD-K68-Ac or anti-SIRT3 antibodies. **e, f** Quantified TMA consisting of luminal A (n = 37) and luminal B (n = 38) samples immunostained for **e** MnSOD-K68-Ac; and **f** SIRT3. The shaded boxes represent the interquartile range; whiskers represent the 10th–90th percentile range. Experiments were done in triplicate. *p < 0.05. A t-test was used to compare data between the two groups. **g** Schematic of the dichotomous role for MnSOD in normal cells (i.e., protection, blue) versus tumor promoter and/or and Tam resistance (red)

antibodies, and representative IHC images for luminal A and B tumor samples are shown (Fig. 8d and Supplementary Fig. 10c, d). Staining intensity was subsequently quantified using automated HistoQuest software that revealed that MnSOD-K68-Ac levels are significantly higher (Fig. 8e), and SIRT3 protein levels are markedly lower (Fig. 8f) in the luminal B samples, as compared to luminal A tumor samples. In addition, stratification of the staining intensities from the luminal A versus luminal B TMAs into low, intermediate, and high staining suggests that there may be a subgroup of luminal B tumors that exhibit significant MnSOD-K68-Ac staining (Supplementary Fig. 10c, d). These results suggest that the SIRT3/MnSOD-Ac signature may be a useful marker to identify a specific subgroup of women with luminal B breast cancer.

## Discussion

A long-standing paradigm in cancer biology is that tumor cells exhibit disruption of cellular antioxidant enzymes, including MnSOD, as compared to normal cells. In this regard, MnSOD functions to detoxify the mitochondria and prevent the damaging effects of the accumulation of superoxide. Loss of its activity is accompanied by increased cellular oxidative metabolites, creating a damage-permissive phenotype[5,16]. We now know that lysine acetylation, which has emerged as an important and perhaps primary mitochondrial post-translational modification, significantly decreases MnSOD activity[3,6,7]. However, the molecular mechanism regarding how MnSOD acetylation contributes to the regulation of MnSOD activity is unclear.

In this study, we have shown that: (1) $MnSOD^{K68Q}$ functions as an in vitro site-directed mutant that mimics an acetylation oncogenic post-translation modification, based on tissue culture and xenograft models; (2) breast cancer cells expressing $MnSOD^{K68Q}$ exhibit a loss of tetrameric MnSOD and an enrichment of the monomeric form; (3) IPed MnSOD-K68Q and MnSOD-K68-Ac, isolated from a bacterial expression system, exhibit peroxidase activity; (4) breast cancer cells selected for resistance to Tam exhibit a MnSOD-K68-Ac tumor cell signature; and (5) a subgroup of human breast malignancies displays a SIRT3/MnSOD-Ac signature, suggesting that this axis may be a potential target for new therapeutic modalities. Finally, the results in this manuscript suggest that under specific physiological conditions when K68 becomes acetylated, MnSOD may function as a tumor promoter, consistent with data implying that MnSOD levels positively correlate with aggressive tumor phenotypes[14]. In this regard, several publications in the last few years have shown a connection between disruption of the MnSOD-Ac axis and human illnesses[4], including aging[34], neurodegeneration[35], cardiovascular disease[36], and insulin resistance[37]. Finally, our results also suggest that ER + breast cancer tumor cells, selected for resistance to Tam over 3 months, exhibit a MnSOD-K68-Ac signature, similar to results previously published, suggesting a role of MnSOD levels[29–31], and/or aberrant ROS levels[23,32], in endocrine resistance in ER + tumor cells.

Although many studies have suggested that human malignancies originating from a diverse range of tissues exhibit dysregulated or decreased MnSOD activity[17], recent studies also identified high MnSOD expression in some advanced tumors[38]. In this regard, these studies suggest a potentially oncogenic role for MnSOD, depending on the specific cellular and metabolic conditions. Since endocrine therapy resistance is a representative phenotype of luminal B breast cancer, these results led us to further investigate the role of MnSOD acetylation in luminal B breast cancer. Our results suggest that a MnSOD-K68-Ac signature is associated with a poor clinical outcome in human breast cancer and thus, this signature may be a molecular biomarker in women with pathologically aggressive luminal B cancer.

Lysine acetylation neutralizes the positive charge of this amino acid, potentially altering the 3-dimensional structure of a protein and its enzymatic function[3,39]. In this regard, a previously published structural analysis of MnSOD, using Adaptive Poisson-Boltzmann Solver, showed the protein near K68-Ac exhibits decreased positive surface charge, as compared to non-acetylated K68, suggested that K68-Ac modulates the electrostatic potential near the active center[40]. Our data seem to confirm this idea (see Supplementary Fig. 11), suggesting that K68 may play an important role in the structure-function relationship of MnSOD.

The Free Radical Theory of Aging suggests that aberrant and/or prolonged intracellular oxidative stress contributes to the carcinogenic process[41,42]. Although the mechanism behind this process is likely to be complex and multifactorial, it is well-established that increased levels of ROS, whether from damaged and/or impaired mitochondrial function or from altered cell metabolism, plays a pivotal role in the aging, as well as the onset of carcinogenesis and tumor cell resistance[43,44]. In this study, we have added another layer to this theory by proposing that mitochondrial protein acetylation may determine cellular ROS and play a role in aging. This would be consistent with studies showing that caloric restriction (CR), which is associated with lower rates of cancer, can decrease mitochondrial acetylation[45].

In this manuscript, we found that acetylation of K68 drives MnSOD conformational changes and induces a significant increase in the monomeric form of MnSOD. In addition, our biochemical tissue culture and bacterial experiments show that MnSOD can, under specific condition, function as a peroxidase; however, this is not, a priori, causative data that monomerization is the underlying molecular mechanism for the observed enzymatic switch towards a peroxidase. In this regard, several manuscripts in bacteria and yeast suggest, based on context that MnSOD binding between manganese and iron are exchangeable, suggesting this may be, at least in a part, a potential additional mechanism.

These results raise an important question: what is the in vivo impact of K68-Ac and how do changes in stoichiometry of MnSOD-K68-Ac drive biological processes? In this regard, one possible explanation might be that MnSOD exists as in a dynamic tetrameric:monomeric equilibrium based on the K68-Ac. Since it is well-established that MnSOD acetylation responds to changes in nutrient availability, such as CR[12], it could be proposed that the stoichiometry of MnSOD-K68-Ac directs this tetrameric:monomeric equilibrium to program mitochondrial energy pathways to match organismal physiological conditions, including fasting or feasting. However, there is much to be done to determine if this idea is correct and the physiological consequence of such a hypothesis.

In conclusion, our results suggest that MnSOD-K68-Ac may determine the tetramer:monomer equilibrium in the cell, and K68-Ac may shift this equilibrium toward the monomeric form of MnSOD. While the mechanism for how this may play a role in carcinogenesis will require further study, this work does suggest that the dysregulation of the SIRT3-MnSOD-Ac-ROS axis may be a signature, as well as a driver, of a damage-permissive cellular phenotype. Overall, this study provides a framework for interpreting the link between the acetylation-dependent structural forms of MnSOD (i.e., tetramer or monomer) and their roles in transformation and tumorigenesis (Fig. 8g).

## Methods

**Cell lines**. The ER + MCF7 and T47D human breast cells, which were all obtained from ATCC, authenticated using STR profiling with CellCheck 9 Plus by IDEXX Bioresearch, and tested for mycoplasma using Mycoplasma Detection Kit,

InvivoGen, Inc in April 2016, were cultured in Dulbecco's Modified Eagle's Medium (DMEM, Gibco) supplemented with 10% fetal bovine serum (FBS; Sigma) and Antimycotic solution (Sigma). Cells were maintained in a humidified 37 °C environment with 5% CO₂. pMEFs were isolated from E14.5 isogenic $SIRT3^{+/+}$ mice (through a protocol that was approved by the Institutional Animal Care and Use Committee (IACUC) and complied with related animal research ethical regulations) and maintained in a 37 °C incubator with 5% CO₂ and 6% oxygen, except when otherwise noted. MCF7 and T47D cells were grown for 3 months in 1 µM hydroxy-Tam to create MCF7-HTR and T47D-HTR permanent cell lines, and several different subclones were frozen. MCF7-HTR and T47D-HTR were not used for >5 passages, and new cell lines were used. All experiments were done using exponentially growing cell cultures at 50% confluence.

**Virus plasmids and short-hairpin RNA (shRNA) constructs and mutagenesis.** To package lentivirus, 293T cells (obtained from ATCC) were transfected with 5 µg DNA, 5 µg psPAX2 packaging plasmid, and 500 ng VSV.G envelope plasmid. Viral supernatant was collected after 72 h and filtered through a 0.45 µm filter (Corning). Lenti-SIRT3^WT and the deacetylation-null mutant (lenti-SIRT3^DN) were gifts from Dr. Toren Finkel (NIDDK). pLKO.1 human SIRT3 shRNA was purchased from OpenBiosystem. Lenti-MnSOD plasmid (human) was used as the MnSOD^WT plasmid and for site-directed mutagenesis, i.e., K68 to arginine (R: deacetyl mimic) or glutamine (Q: acetyl mimic) (Bioinnovatise,). MCF7, T47D, and NIH3T3 cells were infected with 5 MOI of lentivirus and selected with DMEM containing 2 µg/mL puromycin (Invitrogen) or 100 µg/mL G418 sulfate (Invitrogen) for 14 days. After a 2-week selection period, cells were grown in DMEM with 10% FBS. All the primers used to make the recombinant plasmids are shown in Supplementary Table 1.

**Transduction of antioxidant enzymes.** Replication-incompetent adenoviral vectors, AdCMV Bgl II (AdBglII) and AdCMV Mito-Catalase (AdMitoCat) were received as a gift from Dr. Douglas Spitz (University of Iowa) and Dr. Marcelo Bonini (Medical College of Wisconsin, WI). Cells were plated the day before virus administration. The desired number of viral particles was added for 24 h, and then the media was changed to fresh media and left for another 48 h prior to each experiment.

**In vitro cell transformation assay.** For this study, spontaneous immortalization of pMEFs is the ability to continue dividing past passage 15. For in vitro immortalization experiments, MnSOD, or one of its site-directed mutants (K-R or K-Q), was co-infected with c-Myc and/or Kras into third-passage pMEFs. Cells were cultured and split every 2 days to prevent confluency and plated onto a new 100 mm dish at $3.0 \times 10^5$ cells. After 15 additional passages (18 total), cells were considered immortalized.

**Clonogenic cell survival assay.** For the clonogenic survival analysis, exponentially growing cells were replated using appropriate dilutions, and clonogenic survival was evaluated after 14 days in regular growth medium. Cells were stained with crystal violet, and colonies of >50 cells were counted and utilized to calculate clonogenic survival[46].

**Soft-agar colony formation assay analysis.** Ten-thousand cells were plated on 0.3% agar in growth medium over a 0.6% base agar foundation layer in growth medium[7,8]. After 21 days, the colonies were visualized under a ×20 microscope (Zeiss), and images were acquired.

**Xenograft in vivo tumorigenesis analysis.** Five million MCF7, MCF7-HTR, or MCF7 cells (obtained from ATCC) expressing $MnSOD^{K68WT}$, $MnSOD^{K68R}$ or $MnSOD^{K68Q}$ were injected into Foxn1nu athymic nude mice (Jackson Laboratory) that were 6-weeks-old (through a protocol that was approved by the Institutional Animal Care and Use Committee (IACUC) and complied with related animal research ethical regulations). Tumor sizes were examined using a Vernier caliper every 2–3 days, and the volumes were calculated using $V = 1/2 \times W^2 \times L$. When the sizes of tumors reached an average of 1000 mm³, the mice were sacrificed, and the tumors were collected for weight and size analysis.

**TetOn inducible system for MCF7-HTR MnSOD^K68R xenografts.** MCF7-HTR cells were infected with pLenti-CMV-IE-Tet-On Advanced-IRES2-ZsGreen1-P2A-Puro plasmid (Clontech, Mountain View, CA, modified by Bioinnovatise, Inc., Baltimore, MD) and selected under puromycin (1 µg/mL) and for green color (MCF7-HTR TetOn). Subsequently, MCF7-HTR TetOn cells were infected with pLenti-SV40 promotor-HygroR-SV40 poly(A)/pTreDual2MnSODK68R-mCherry (MCF7-HTR TetOn MnSODK68R; Clontech, Inc.) and were selected for mCherry and with hygromycin (50 µg/mL). To confirm the activation of the TetOn system, 1 µg/mL of doxycycline (Acros Organics, New Jersey) was added and after 24 h, expression was verified by the presence of mCherry, and fluorescent images were taken. Western blots for were used to validate that expression of $MnSOD^{K68R}$ was activated. Eight-week-old nude mice (Jackson Labs) were injected with $5 \times 10^6$ MCF7-HTR TetOn MnSOD^K68R cells into the right hind flank at the time of

tamoxifen pellet placement (5 mg pellet, Innovative Research of America, Sarasota, FL). The experimental group was given feed containing doxycycline (625 ppm, Envigo Teklad Diets, Madison, WI); the control group remained on standard feed provided by Northwestern's Animal Facility. Feed was changed every three days for the duration of the experiment. Tumors were measured every other day, and at the end of the experiment, tumors were removed for analysis through a protocol that was approved by the Institutional Animal Care and Use Committee (IACUC) and complied with related animal research ethical regulations.

**Immunohistochemistry staining and analysis.** Breast cancer tissue array slides (Biomax) were immersed twice in 100% xylene (Sigma) for 5 min and 100% ethanol (Sigma) for 5 min. Slides were sequentially immersed with 95%, 80 and 50% ethanol for 5 min before immersion in water and fixation in 95 mL 95% ethanol and 5 mL of 37% formaldehyde for 2 min. Slides were then treated with 1% Triton X-100 in 1x PBS (Corning) for 20 min, washed three times in 1x PBS for 5 min, and quenched in 0.3% H₂O₂ in 1x PBS for 20 min. Slides were blocked with 10% donkey serum (Sigma), 1% bovine serum albumin (BSA; Sigma) and 0.3% Triton X-100 (Sigma) in 1x PBS for 2 h before treatment with an anti-MnSOD-K68-Ac antibody (1:250 dilution, Abcam #ab137037) for 48 h at 4 °C. These slides were subsequently incubated at room temperature for 1 h before being washed three times with 1x PBS for 5 min. Rabbit secondary antibody (1:200 dilution, A0545, Sigma) was diluted in antibody solution and applied to slides for 1 h before being washed three times with 1x PBS for 5 min each. The slides were treated with VECTASTAIN ABC kit (Vector Laboratories) for 45 min following the manufacturer's protocol to detect avidin/biotinylated enzyme complexes. Slides were treated using the DAB peroxidase substrate kit (Vector Laboratories), per the manufacturer's protocol, and stained in hematoxylin (Sigma) for 10 min. Then slides were destained with 100 mL of 70% ethanol and 1 mL of 37% hydrochloric acid before dehydration. The intensities were quantified using HistoQuest software (Tissuegnostics).

**Peroxidase activity assay.** One-million cells expressing Flag-tagged $MnSOD^{WT}$, $MnSOD^{K68R}$, and $MnSOD^{K68Q}$ were lysed for 30 min in 25 mM Tris-HCl pH 7.4, 150 mM NaCl, 1 mM EDTA, 0.1% NP-40, 5% glycerol, protease inhibitors (Bio-Tool) and TSA (Trichostatin A, Sigma). Lysates were quantified with the Bradford assay (BioRad) and IPed using anti-Flag antibody (Sigma). The peroxidase enzymatic activities of these IPed proteins were determined by using pyrogallol as the substrate. In the reaction mix, the final concentrations were 14 mM potassium phosphate (Sigma), 0.027% (v/v) hydrogen peroxide (Sigma), and 0.5% (w/v) pyrogallol (Sigma). The plate was tapped to mix the sample and reaction reagent, incubated for 10 min at room temperature, and then read at OD 420 nm. The increase in A420 was recorded every 3 min. The ΔA420/20s was obtained using the maximum linear rate or 0.5-min interval for all the test samples and blanks. The peroxidase activity was calculated using this equation: Units/mL = [(ΔA420/20 s Test Sample – ΔA420/20 s Blank) (reaction volume) (dilution factor)]/[(12)(0.1)]. The methods for the bacterial expression and mammalian systems that was used for the peroxidase assays is in the Supplementary Information section.

**Glutathione analysis.** One-million cells at 70–80% confluency were lysed in 1.34 mM diethylenetriaminepenta-acetic acid (DETAPAC, Sigma) and dissolved in 143 mM sodium phosphate (Sigma). Then 6.3 mM EDTA (Sigma) and 5% 5-sulfosalicylic acid (Sigma) were added to the lysates. Fifty microliters of lysate were mixed with 700 µL 0.298 mM NADPH (Sigma) dissolved in sodium phosphate buffer, 100 µL 6 mM 5,5'-dithio-bis-2-nitrobenzoic acid (DTNB, Sigma) in sodium phosphate buffer, 100 µL water, and 50 µL 0.023 U/µL glutathione reductase (GR) dissolved in water (Sigma). Kinetic absorbance was read at 412 nm every 15 s for 2.5 min using an xMark™ microplate absorbance spectrophotometer (BioRad), and the rates were compared to a standard curve. Tumors were lysed in DETAPAC buffer before being assayed, and protein concentrations were measured for standardization of GSH levels that were normalized using the BCA method.

**MnSOD/SOD enzymatic activity.** Total SOD and MnSOD activity were determined by an indirect competitive inhibition assay[47]. Superoxide is generated from xanthine by xanthine oxidase and detected by recording the rate of reduction of nitroblue tetrazolium (NBT). SOD scavenges superoxide and competitively inhibits the reduction of NBT. One unit of SOD activity is defined as the amount of protein required to inhibit 50% of the maximal NBT reduction. To obtain the amount of MnSOD activity, 5 mM sodium cyanide was added to inhibit the CuZnSOD enzyme activity. The protein levels in each sample were measured using the BCA protein assay[48–51].

**Western blot analysis.** Cells and tissues were washed three times with cold 1x PBS, harvested, and lysed for 30 min in 25 mM Tris-HCl pH 7.4, 150 mM NaCl, 1 mM EDTA, 0.1% NP-40, 5% glycerol with protease inhibitors (BioTool) and TSA (Sigma), then quantified by Bradford assay and immunoblotted with: anti-MnSOD (1:1000 dilution, Millipore, #06–984), anti-MnSOD-K68-Ac (1:1000 dilution, Abcam, #Ab137037), anti-MnSOD-K122-Ac (1:500, Abcam, #Ab214675), anti-SIRT3 (1:1000 dilution, Cell Signaling, #D22A3), anti-IDH₂ (1:1000 dilution, Cell Signaling, #56439), anti-IDH2-K413-Ac (1:1000 dilution, Epitomics, Inc,

Burlingame, CA (this company has been bought by Abcam, Inc.)), anti-OSCP (1:1000 dilution, Santa Cruz Biotechnology, #sc-365162), anti-OSCP-K139-Ac (1:1000 dilution, Epitomics, Inc, Burlingame, CA), and anti-actin (1:10,000 dilution, Cell Signaling, #4970). Secondary antibody includes anti-rabbit and anti-mouse (1:10,000 dilution, Cell Signaling, #7074, #7076). For the MnSOD tetramerization assay, lysed cells were treated with 0.1% glutaraldehyde for 10 min at room temperature before samples were immunoblotted with anti-MnSOD antibody.

**Determination of cellular superoxide levels using MitoSox.** Steady-state levels of mitochondrial $O_2^{\bullet-}$ were estimated using the oxidation of a fluorescent dye, dihydroethidium (DHE) (Life Technologies). Cells were trypsinized, washed, and then labeled in 5 mM pyruvate containing 1x PBS with MitoSox Red (2 μM in 0.1% DMSO) for 20 min at 37 °C. After labeling, cells were kept on ice. Samples were analyzed using a Fortessa flow cytometer (Becton Dickinson Immunocytometry System, Inc., Mountain View, California; excitation 488 nm, emission 585, 25 nm band-pass filter). The mean fluorescence intensity (MFI) of 10,000 cells was analyzed in each sample and corrected for autofluorescence from unlabeled cells. The MFI data were normalized to control levels.

**Estimation of cellular $H_2O_2$ levels using $CDCFH_2$ oxidation.** Steady-state levels of $H_2O_2$ were estimated using the oxidation-sensitive 5-(and 6)-carboxy-2′,7′-dichlorodihydrofluorescein diacetate ($CDCFH_2$) (Life Technologies). The cells were trypsinized and washed with 1X PBS once and then labeled with $CDCFH_2$ or CDCF (10 μg/mL, in 0.1% DMSO, 15 min) at 37 °C. After being labeled, the cells were kept on ice. Samples were analyzed using a Fortessa flow cytometer (Becton Dickinson Immunocytometry System, Inc., Mountain View, California; excitation 488 nm, emission 530 nm, 25 nm band-pass filter). The MFI of 10,000 cells was analyzed in samples and corrected for autofluorescence from unlabeled cells. The MFI data were normalized to control levels.

**Survival experiments using MnSOD mimetic GC4419 treatments.** To test parameters indicative of oxidative stress, a clonogenic assay with hydroxy-Tam and antioxidant treatments was performed. Cells were plated at a density of 50,000 cells per 60-mm dish and treated with 1 μM hydroxy-Tam (Sigma) and 5 μM GC4419 (Galera Therapeutics) for a total of 120 h. This protocol was repeated with a fresh medium change every 24 h for 5 days. On day 6, the cells were trypsinized, counted, and replated in control medium using appropriate dilutions, and clonogenic survival was evaluated.

**Incorporation of N-(ε)-acetyl-lysine into K68.** BL21 (DE3) pMAGIC chemically competent E. coli cells, which were a kind gift from Andrzej Joachimiak, Argonne National Labs, were co-transformed with pEVOL-AcKRS and pET21a-MnSODK68$^{TAG}$ plasmids or pET21a-wtMnSOD to express MnSOD-K68-Ac and MnSOD-WT proteins. The cells harboring pEVOL-AcKRS and pET21a-MnSODK68$^{TAG}$ plasmids were incubated in 100 mL LB with 300 μg/mL ampicillin, 50 μg/mL kanamycin, and 50 μg/mL chloramphenicol (37 °C, 220 rpm) for 3 h at 37 °C, and 50 mM nicotinamide (Sigma) was added to this culture. When $OD_{600}$ reached 1.1, 2 mM Ne-acetyl-lysine (Sigma) was added to the culture and cells were induced by the addition of 0.4 mM IPTG and 0.2% arabinose (25 °C, 180 rpm) for another 20 h. (The bacterial MnSOD expression and lysine acetylation tRNA mutant plasmids used to make physically acetylated MnSOD-K68-Ac were a kind gift from Dr. Jiangyun Wang, Institute of Biophysics, Chinese Academy of Sciences, Beijing, China).

The BL21 (DE3) cells harboring pET21a-wtMnSOD plasmid were cultured in 5 mL LB media with 300 μg/mL ampicillin and 50 μg/mL kanamycin and 1 mL of this culture was incubated in 100 mL LB with 300 μg/mL ampicillin and 50 μg/mL kanamycin (37 °C, 200 rpm) overnight. The next day, 1 L of LB with 300 μg/mL ampicillin and 50 μg/mL kanamycin was inoculated with 10 mL of overnight culture, for ~2.5 h until $OD_{600}$ = 0.6, and then cells were induced with 0.4 mM IPTG (25 °C. 180 rpm) overnight. All purification steps were performed on ice. E. coli cells in 1 L LB were harvested by centrifugation (6000 rpm, 10 min, 4 °C) and washed with 50 mL Buffer I containing 20 mM imidazole, 50 mM Tris-HCl, 200 mM NaCl, 5 mM $MgCl_2$, 50 mM nicotinamide, pH = 8.0. Then pellets after centrifugation were suspended with 50 mL Buffer I supplemented with 1 mM PMSF and ~1 mg/mL lysozyme, and the lysates were incubated at 4 °C for 10 min. Then protein was extracted by sonication cycling (5 s on, 6 s off, 25 min). The extract was clarified by centrifugation (13,000 × g, 30 min, 4 °C) and the pellet discarded. In all, 0.2 mL Ni$^{2+}$-NTA beads were added to the supernatant and incubated with agitation at 4 °C for 1 h.

Beads were transferred into a column and washed three times with Buffer I containing increasing imidazole gradient (50, 75, 100 mM) and protein was eluted in 1 mL Buffer I supplemented with 200 mM imidazole. The proteins were analyzed by SDS-PAGE and then concentrated using Ultra-15 Centrifugal Filter Unit (10 kDa, Millipore Amicon™, USA UFC800324). The eluted protein was then re-buffered to Buffer II (50 mM Tris-HCl, 200 mM NaCl, 5 mM $MgCl_2$, 50 mM nicotinamide, pH 8.0) and loaded onto an equilibrated Ni$^{2+}$-NTA ÄKTA FPLC Purifier system with GE HisTrap HP columns (Product # GE17524701) and further purified by a Superdex 200 Increase 10/300 GL column (GE Healthcare, Product #

GE28-9909-44) in a buffer containing 50 mM potassium phosphate (pH = 7.8). Peak fractions were collected using an automated fraction collector. $A_{280}$ as a function of elution volume/time was also recorded[13,27,28]. The peak protein fraction concentrations were determined and immunoblotted with anti-MnSOD and anti-MnSOD-K68-Ac. The remaining purified proteins were measured for peroxidase activity and MnSOD activity. The eluted fractions were then subjected to further analysis. A calibration curve was generated using a gel filtration low and high molecular weight kit (GE Healthcare) according to the manufacturer's instructions and is shown Fig. 4e, which was used to determine the relative size of peak 1 and peak 2.

**Immunofluorescence sample preparation and image acquisition.** Cells seeded on glass coverslips were fixed in 4% paraformaldehyde and then blocked with 1% BSA and 10% normal goat serum in 1x PBS. Cells were incubated with anti-Ki-67 (c-bioscience) antibody in 1x PBS followed by incubation with goat anti-rabbit IgG conjugated with Alexa Fluor 647 (Invitrogen) in 1x PBS with 5% goat serum. Cells were washed in 1x PBS, mounted, and imaged with a fluorescence microscope. Fluorescence images were captured using a laser scanning confocal microscope (Nikon A1R). The paired images in all the figures were collected at the same gain and offset settings. Post-collection processing was applied uniformly to all paired images. The images were either presented as a single optic layer after acquisition in z-series stack scans from individual fields or displayed as maximum intensity projections to represent confocal stacks.

**Ethical regulations for animal testing and research.** All research in this manuscript and the data obtained was done in compliance with all the relevant ethical regulations for animal testing and research at Northwestern University. The mouse work was done under protocol # IS00001989, De Novo # IS00001989_IM13. This is titled: Loss of mtSIRT3, Decreased MnSOD Activity, and IR Induced Genomic Instability. This protocol was re-approved on 3rd August, 2018, by the Office for the Institutional Animal Care and Use Committee for 2996 mice with funding from NIH 5R01CA152601.

**Statistical analysis.** Statistical analysis was performed using GraphPad Prism for Windows (GraphPad Software, San Diego, CA). Error bars indicate mean ± SEM. One-way ANOVA analysis with Tukey's post-analysis was used to study the differences among three or more groups. For two column bar graphs (i.e., is a significant difference between the means of two groups), a t-test was used. All experiments were repeated at least three times. Statistical significance was assumed at $p < 0.05$.

**Reporting summary.** Further information on research design is available in the Nature Research Reporting Summary linked to this article.

## Data availability

Data supporting the findings of this manuscript are available from the corresponding author upon reasonable request. A reporting summary for this article is available as a Supplementary Information file. The source data underlying Figs. 2c–f, 3a–d, 4a–d,f–i, 5a–h, 6a–h, 7a–h, k and Supplementary Figs. 2e, f, 3b, 4c–e, 5a, b, f, 6a–f, 7a–f, 8a, c, e, g, 9b, d, 10b are provided as a Source Data file.

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

## Acknowledgements

D.G. is supported by 2R01CA152601-A1, 1R01CA152799-01A1, 1R01CA168292-01A1, 1R01CA214025-01, the Avon Breast Cancer Foundation, the Lynn Sage Cancer Research Foundation, the Zell Family Foundation, and the Chicago Biomedical Consortium, as well the Searle Funds at The Chicago Community Trust. M.B. is supported by a U.S. Department of Defense grant 67263-RT-REP and an NIH 1RO1HL125356 award. Y.Z. is supported by a Robert H. Lurie Comprehensive Cancer Center Translation Bridge Fellowship Award. Y.G. was supported in part by the Chicago Cancer Baseball Charities at the Lurie Cancer Center of Northwestern University. K.J.F.S. is supported by 5R01AI092825-07. Imaging work was performed at the Northwestern University Center for Advanced Microscopy, which is generously supported by NCI CCSG P30 CA060553, awarded to the Robert H. Lurie Comprehensive Cancer Center. Proteomics services were performed by the Northwestern Proteomics Core Facility, generously supported by NCI CCSG P30 CA060553 awarded to the Robert H. Lurie Comprehensive Cancer Center and the National Resource for Translational and Developmental Proteomics supported by P41 GM108569. The structural analyses of MnSOD were performed at the Northwestern University Structural Biology Facility, which is generously supported by NCI CCSG P30 CA060553 awarded to the Robert H. Lurie Comprehensive Cancer Center. The bacterial MnSOD expression and lysine acetylation tRNA mutant plasmids were a gift from Jiangyun Wang, Institute of Biophysics, Chinese Academy of Sciences, Beijing, China.

## Author contributions

D.G. designed, supervised, and directed all experiments. Y.Z., X.Z., A.E.D, J.O., Y.G., E.L.T., S.-H.P, G.L., M.B.K., H.J., S.Q., and N.H. carried out experiments. Y.Z., A.E.D., Y.G., R.H., K.J.F.S, N.H., M.B., and D.G. analyzed and interpreted data, and Y.Z., A.E.D., Y.G., E.L.T., M.E.S., N.H., and D.G. wrote the manuscript, supplemental section, and made the figures.

## Additional information

**Competing interests:** The authors declare no competing interests.

