## [Peer Review File · Nature Communications]

Reviewers' comments:

Reviewer #1 (Remarks to the Author):

Zhu et al. showed that MnSODK68Q promoted transformation and cancer cell proliferation, but MnSODK68R did not. K68Q and SIRT3 inactivation prevented the formation of SOD tetramer. K68Q gained peroxidase activity and its transformation activity could be rescued by mitochondrial catalase. K68Q overexpression conferred reduced SOD activity and increased oxidative stress compared to WT or K68R. K68Q or SIRT3 inactivation resulted in hydroxyl tam resistance. Hydroxyl tam resistant cells had increased level of K68 acetylation, reduced SOD activity, oxidative stress, reduced tetramer, increased proliferation and tumorigenesis, some of which could be rescued by K68R but not K68Q. SIRT3 expression was reduced and K68-acetylation was increased in luminal B breast cancer patient samples compared to luminal A. The authors concluded that acetylation switches MnSOD from anti-tumorigenic homotetramer to a tumor-promoting monomer.

In recent years, acetylation has been increasingly recognized as a posttranslational modification of numerous proteins. There is much interest in understanding the physiological implications of this PTM and the molecular mechanism by which this PTM changes protein function. While there have been some indirect evidence implicating the physiological relevance of protein acetylation, direct evidence is rare. This study provides one of the first direct evidence that protein acetylation is physiologically relevant, by testing MnSOD mutants that mimic acetylated or deacetylated forms in tumorigenesis. Mechanistically, this paper proposes that acetylation affects the protein oligomerization and intriguingly, converting a protein to possess a different enzymatic activity and elicit opposing effect in tumorigenesis. The mechanism is novel. In light of these conceptual advances, this study is a good candidate for Nature Communication.

Overall, the experiments are well rationalized and data are robust. This reviewer is particular impressed that some experiments were tested in different cell lines. I recommend its publication after minor revision.

1. The introduction needs to be improved.

Line 75: "Studies have shown that SIRT3-guided deacetylation at MnSOD lysines 68 (K68)⁶ and 122 (K122)^{8,9} directs MnSOD enzymatic activity to protect cells from mitochondrial ROS-induced cellular damage." K53 and K89 have also been identified as acetylation sites targeted by SIRT3 (Qiu, Cell Metabolism 2010) and they have been shown to be physiologically important, i.e. in maintaining stem cell function (Brown, Cell Reports 2013).

2. Figure legends need to be improved to ease the reading.

Fig 1 title: "MnSODK68Q overexpression promotes a transformation-permissive phenotype both in vitro and in vivo." In vitro but not in vivo data are included in Figure 1. The title should delete in vivo.

Fig1 legend: "Errors represent \pm 1 SEM. **p < 0.01, and ***p < 0.001." No data shown have error bars and statistical tests. These notes should be removed.

Fig 3 legend: "*p < 0.05, **p < 0.01, and ***p < 0.001." No data show * or ***. Should be deleted.

Fig 7 legend: "(f) Quantification of the results shown in S5a." Needs to state what is done in the experiment.

Fig 8a labels are missing. Which one is MCF7 which one is MCF7 HTR?

Reviewer #2 (Remarks to the Author):

Review to the manuscript:

Lysine 68 acetylation regulates dichotomous role of MnSOD: a protective tetrameric detoxification complex versus a monomeric oncoprotein by Zhu et al.

The lab of Dr. Gius presents interesting data showing that acetylation regulates MnSOD function. MnSOD has been shown to play a role as a tumor suppressor during initiation of neoplasia. However, it becomes an oncogene product upon tumor progression supporting the development of more aggressive tumors.

Using various cell lines, some of which were derived from patients, Zhu and co-workers analysed the effect of lysine-acetylation, predominantly at lysine 68 (K68), on MnSOD function and its impact on tumor development. They derived an interesting model suggesting that acetylation drives the formation of MnSOD monomers showing peroxidase activity, while the non-acetylated form exists as a tetramer with superoxide dismutase (SOD) activity. It is only the monomeric form that is on the one hand oncogenic and on the other hand promotes a Tamoxifen-resistance phenotype.

Overall, the experiments performed were sound and the results obtained are highly valuable for a huge scientific community. Zhu and co-workers suggest a novel role of post-translational lysine-acetylation to confer a gain-of-function, which is rarely the case, and furthermore they open up a novel therapeutic approach for tumor therapy approaching lysine-acetylation. However, there are several issues that would need to be addressed to significantly improve the manuscript prior to publication. I support a publication in Nature Communications if the authors without exception address all the major points shown below.

Point 1:

Although lysine 68 in MnSOD seems to be an important residue for MnSOD function, the major point of criticism of this manuscript is that Zhu and co-workers used lysine to glutamine (as acetylation mimic) and lysine to arginine (to conserve the non-acetylated state) mutations throughout. It is shown that these mutations can be poor mimics of lysine-acetylation at the molecular level. Therefore, the authors have to show that the MnSOD K68Q and K68R mutant proteins are reliable to study lysine-acetylation. One way would be to produce site-specifically lysine-acetylated MnSOD protein using the genetic-code expansion concept (see Knypausen et al., 2016). It is essential to show how acetylation at K68, the K68Q and the K68R protein interferes with MnSOD oligomerisation and catalytic activity. Analytical size-exclusion chromatography runs (or alternative methods) combined with enzymatic assays to show peroxidase and superoxide dismutase activity of the MnSOD proteins (MnSOD wildtype, acetylated at K68, K68Q and K68R) should be performed. These data will provide evidence for the hypothesis that the oligomerisation state of MnSOD is affected by K68-acetylation and how this determines MnSOD enzymatic activity (monomer: peroxidase; tetramer: superoxide dismutase). These experiments are essential as they show the validity of the central message of the paper. Zhu and co-workers describe an influence of lysine-acetylation but did not perform a single experiment with acetylated MnSOD protein. Without showing that the acetylated protein is doing what they suggest by performing *in vitro* experiments, most paragraphs entitled with "Acetylation of MnSOD..." must be rephrased in saying that mutation of K68Q in MnSOD results in a certain phenotype.

Point 2:

Along this line, it is reported that MnSOD acetylated at K122 is not a direct target for Sirt3 (Knypausen et al., 2016). The authors should be more precise when writing that acetylated K122 in MnSOD is deacetylated by Sirt3. They should at least write that K122-acetylated MnSOD is not a direct target. Using acetylated MnSOD K68 protein, the authors should analyse if Sirt3 deacetylates it *in vitro*. This will show if K68-acetylated MnSOD is a direct target for Sirt3.

Point 3:

The authors describe that mutation of K68Q in MnSOD results in monomerisation of the protein. For the *in vivo* impact of the acetylation at K68 it is essential that the acetylation occurs at sufficiently high stoichiometries. Using K68Q mutant, the authors obviously establish an

constitutively acetylated state of MnSOD. It is of course not clear which stoichiometries of K68-acetylation in MnSOD are achieved under physiological conditions. The authors could at least discuss this point. The authors show acetylated MnSOD K68 protein by western blotting, if I understood correctly by using a specific anti-MnSOD-K68-ac antibody, suggesting that K68-acetylated MnSOD is present. The authors should show the reliability of the antibody using purified non-acetylated and K68-acetylated MnSOD protein. However, it is not clear if this is a Sirt3 regulated site and in Fig. 5d there is no obvious difference in Sirt3 levels visible. Using nicotinamide to inhibit Sirt3 will show if the K68-acetylation is regulated by sirtuins.

Point 4:

Along the line of point 3, the stoichiometry of acetylation does not play a major role if the acetylation event creates a gain-of-function rather than a loss-of-function. In this context, the authors say that K68-acetylation of MnSOD is a gain-of-function modification. However, from my point of view the authors cannot deduce this from their data. It might be a gain-of-function, i.e. MnSOD monomerisation and increased peroxidase activity, that results in the phenotype observed. However, it can likewise be the loss in MnSOD oligomerisation and loss in SOD activity that supports the more tumorigenic phenotype. Therefore, the authors have to present additional data to show that it is the gain in peroxidase activity or increase in monomeric MnSOD that supports tumor development and not loss in MnSOD tetramerisation or loss in superoxid dismutase activity of MnSOD before making such statements.

Point 5:

The authors use Ras and c-Myc to immortalise or transform primary cells. When did they use the oncogenic G12V Ras and when wildtype Ras?

Point 6:

The authors say that monomeric MnSOD, driven by acetylation at K68, is more oncogenic and furthermore supports the Tamoxifen-resistance phenotype in MCF7 and T47D cells. They concluded this from observing a higher level on K68-acetylated MnSOD in these cells. The mechanism underlying this development of Tamoxifen-resistance is not clear and it is also not clear from the data presented what, acetylation or Tamoxifen-resistance, is cause and effect and if there is a direct relationship. Either the authors perform additional experiments to show if acetylation of MnSOD at K68 increases Tamoxifen-resistance or the other way around Tamoxifen-resistance drives acetylation of MnSOD at K68, or they rephrase it in the manuscript accordingly.

Point 7:

More aggressive tumors show elevated levels of MnSOD. This would drive the formation of MnSOD oligomers. According to the model presented by Zhu et al., acetylation of MnSOD counteracts this oligomerisation. The authors should show how the total MnSOD levels and the levels of K68-acetylation correlate with different tumor cells of various aggressiveness. The acetylation of MnSOD must counteract the concentration dependent oligomerisation of MnSOD, which might occur upon increase in MnSOD levels.

It is mechanistically difficult to understand why only overexpressed MnSOD shows elevated peroxidase activity. How can the author mechanistically explain this? Is only the overexpressed MnSOD acetylated and this in turn leads to its monomerisation and subsequent increase in peroxidase activity?

Point 8:

The authors performed crosslinking experiments using glutaraldehyde to show a MnSOD tetramerisation and monomerisation. However, this is only an indirect evidence for the oligomeric state as mutation of K68Q or K68 might affect other processes such as subcellular localisation or something else. The fact that they even observe MnSOD dimers, which I think do not occur in vivo, shows that using this assay the physiologically relevant forms cannot be represented. Proper analytical in vitro experiments such as ultracentrifugation, size-exclusion chromatography, isothermal titration calorimetry, etc. should be performed to show the oligomeric state.

Point 9:

Statements that peroxidase activity needs acetylation at K68 must be rewritten to be a bit more circumspect. There is no evidence that acetylation at K68 is needed for peroxidase activity. Maybe

it is the monomerisation, which elevates MnSOD peroxidase activity, but this monomerisation could also be driven by other events such as other post-translational modifications, etc. There are other statements in the manuscript that need also some rewriting to be more precise and circumspect. As an example, at several stages in the manuscript the authors say that MnSOD is a Sirt3 target (at various acetylation sites such as K122-ac, K139-ac, K68-ac) but there is no experiment showing this. Did the authors show at all, that Sirt3 knock-out or knock-down increases acetylation at MnSOD K68-ac? If not, they should at least do this to show that MnSOD K68-acetylation is either directly or indirectly affected by Sirt3. The authors state that their results show that acetylation at K68 in MnSOD promotes, at least in some part, a Tamoxifen-resistance phenotype. As shown above this statement is too strong and a proof is missing. The authors should be more concise and more circumspect with these kind of statements throughout their manuscript.

Point 10:

The authors performed an experiment expressing a mitochondrial catalase to reverse a Tamoxifen-resistance phenotype in an MnSOD K68Q background. It is not clear what the addition of catalase should have for an effect. The MnSOD K68Q cells should produce peroxidase which anyway removes hydrogen-peroxide, would it not? Could the authors explain the experiment they performed. Would it not be more suitable to increase hydrogen-peroxide levels and analyse if is possible to reverse the impact of MnSOD K68Q on Tamoxifen-resistance? How would a catalytically dead MnSOD K68Q dead mutant behave? These experiments will show the impact of MnSOD peroxidase activity on development of Tamoxifen-resistance.

Point 11:

For many experiments, particularly the western blottings, the primary data were not shown. As an example, Fig. 8 shows quantifications of Sirt3 levels in luminal A and B cells. Please provide the primary data (also for other experiments) and show it at least in the supplementary section.

Point 12:

The authors say in the discussion that they show that MnSOD K68 is a physiologically relevant Sirt3 target. This is exaggeration and as stated above I do not find any experiment to show this. Furthermore, they claim that K68-acetylation this is a gain-of-function modification. This might be the case and this is an interesting issue. However, as stated above the evidence is missing and the phenotype on tumor progression might also be explainable by a loss-of-MnSOD-function. The phenotype observed might be explained by a loss of superoxide detoxification upon acetylation of MnSOD and the following loss in superoxide dismutase activity resulting for example in a increase in genomic instability.

A figure showing the structure of MnSOD and the position of K68 would be helpful to envision a potential role of K68-acetylation on MnSOD function.

The statement that acetylation of MnSOD at K68, which affects MnSOD assembly and function, reprogramming mitochondrial function is exaggerated.

Please, be more circumspect with your statements and the conclusions drawn from the results throughout the manuscript.

Reviewer #3 (Remarks to the Author):

The study by is technically sound but incremental in its overall contribution to the literature. A role for MnSOD in drug resistance in breast and other cancers is clearly documented in the literature, as is its ability to confer resistance to Tamoxifen. Much of this work is not cited (e.g., see Cho SK et al, Biomaterials, 2013; Fu A et al., Oncotarget, 2016). Moreover, the role of ROS (and blocking with antioxidants like N-acetylcysteine) on ER function and Tamoxifen resistance is known, relevant to the work described here, and yet not cited (e.g., see Nass et al., BBA, 2016; Cook et al FASEBJ, 2014). This study strengthens the body of work already implicating this pathway and MnSOD in Tamoxifen resistance but provides little new regarding the role of oxidative stress in this phenotype. There are few new insights also beyond the studies with the isoforms described, relative to prior work with MnSOD.

Many genes make MCF-7 and T47D cells resistant to Tamoxifen when overexpressed but are not actionable clinically. Evidence that the relationships described in the experimental models here are relevant in patients is largely absent from the manuscript and is not directly supported by new data presented here.

The animal studies are technically sound but the number of animals/group is small. It is not clear how the statistical analyses of these studies account for the repeated measures nature of the data. While reanalysis may be needed to ensure the most appropriate tests are used, for the in vivo studies, this is unlikely to change the outcomes or interpretations.

Reviewer #4 (Remarks to the Author):

Overall this is an interesting study- and could be of great interest, but in its current form it is preliminary in nature and lacks proper controls to make its over-reaching conclusions. The connection between SIRT3 and MnSODK68Q is not mechanistically or conceptually well developed nor is the function of MnSODK68Q as an oncogene rigorously demonstrated.

Additional concerns:

- 1) There are some basic cancer biology concepts that are incorrectly measured. For example in Figure 1a, the authors are assessing the effects of cooperating oncogenes during cellular immortalization, while in Figure 1b&c soft agar colony formation is not a measure of invasiveness as indicated by the authors but rather a measure of anchorage-independent growth. On p6, the authors refer to MnSODK68Q as a tumor promoter or an oncogene. Why use both terms?
- 2) The study is premature as the authors don't actually conduct any transformation experiments (eg. showing that MnSODAC68Q cooperates to form tumors in vivo). In addition, the study lacks additional experiments to define the role of MnSODK68Q during transformation compared with its role in cancer cells once transformed. Currently, these two concepts are combined by using primary MEFs as well as human breast cancer cell lines.
- 3) Showing that MCF7 cells when infected with MnSODK68Q grow in nude mice compared to WT or control cells could imply that MnSODK68Q allows cells to grow without estrogen supplementation. This does not mean that MnSODK68Q is an oncogene.
- 4) Similarly, the increased proliferation conferred in ER+ cancer cell lines by overexpressing MnSODK68Q could mimic the effects of estrogen supplementation. A control should be done by treating the cells with antiestrogen (ICI or Tam) in combination as well as comparing Ki67 rates of cells treated with estrogen compared with MnSODK68Q.
- 5) Fig 3E. What assay was used to measure transformation? The table is not an adequate illustration of this experiment. In addition, the figure legend states this was an immortalization assay, not transformation, Therefore, it appears the authors conflate immortalization with transformation.
- 6) The levels of ER need to be assessed in cells expressing MnSODK68Q- as downregulation of ER could be a mechanism by which MCF7 and T47D cells become less sensitive to Tam as well as are able to grow in mice in the absence of estrogen.
- 7) Many studies have already shown that hydroxy-Tam resistant MCF7 cells form more aggressive in vivo xenograft tumors, so this is more of a control experiment than one that directly links high levels of MnSOD-K68-Ac with Tam resistance. The experiment that needs to be done is to inject hydroxy-Tam resistant cells with an inducible SIRT3 or inducible MnSOD-K68R with Tamoxifen and show that upon induction the tumors stop growing and are sensitized to anti-estrogens.

Dear Referees,

We are very appreciative of your thoughtful reviews, which were mostly supportive, but specifically recommended additional experiments and changes to the text to improve the overall quality and presentation of the manuscript. In this regard, significant changes were made and additional data have been added to the revised manuscript to address your suggestions and concerns. The additional results and control experiments uniformly support our original interpretation of the data, but they make a much stronger argument, thanks to you. As such, our revised manuscript now contains 55 data panels, including one additional schema panel, presented in 8 total figures. Thus, the manuscript has gone from 77 panels (53 in the figures and 24 in the supplement) in the initial submission to 101 panels in the resubmission (55 in the figures and 46 in the supplement). In addition, the manuscript has been reworked with additional text to outline the additional experiments in the manuscript to specifically address the concerns of all four reviewers.

In this regard, multiple changes have been made to the originally submitted manuscript that I believe significantly improve the presentation, as well as the overall scientific conclusions. First, the Introduction has been expanded to more completely present the previously identified SIRT3 lysine targets in MnSOD (i.e., K53, K89, K68, and K122), as well as the published references, in a way that I hope addresses the concerns of both reviewers 1 and 2. At the request of reviewer 2, a new Figure 4 has been added to the manuscript to show that the physical acetylation of monomeric MnSOD-K68 exhibits peroxidase activity, using two different experimental methods. The figure contains 6 new data panels (Figs. 4a-f), as well as one additional panel in the Supplement (Fig. S4). To accommodate this new data: (1) Figure 4 has been changed to Figure 5; (2) Figure 5 has been changed to Figure 6, (3) Figures 6 and 7 have been combined into a new Figure 7 that now contains 11 data panels; and (4) three panels from the old Figure 7 (Figs. 7c, d, e) have been moved into the Supplement as Figures S6a, b, c.

We have also added text to the Introduction outlining the previously published work showing that MnSOD can induce resistance to tamoxifen, as well as the potential link to altered cellular ROS levels, that clearly should have been included in the initial submission, as pointed out by reviewer 3 (what an oversight on our part!). We have also increased the number of mice used for the xenograft experiments from 5 to 10. A new Figure 8c has been added showing that the MCF7-HTR xenografts exhibit a significant decrease in growth due to the inducible, enforced expression of *MnSOD*^{K68R}, which is induced during xenograft growth using an inducible Tet-On expression system, as suggested by reviewer 4. In addition, the text and the legend describing Figure 1 have been significantly reworded to present the data in a clearer and more rigorous manner and to differentiate between experiments measuring immortalization versus transformation, in response to the requests of reviewers 2, 3, and 4 to be more precise in our wording. Finally, we have changed the text to be

more concise when we are presenting *MnSOD*^{K68Q} as an oncogene, which is consistent with our results, and completely removed the phrase “tumor promoter”.

The Supplement has also been significantly reworked to address the concerns of the reviewers. A new Figure S1 has been added to demonstrate a more transformed phenotype in pMEFs infected with lenti-*MnSOD*^{K68Q}, as part of the reorganization of Figure 1, which contains new xenograft data, showing that pMEFs infected with both lenti-Myc and lenti-*MnSOD*^{K68Q} form tumors in nude mice, as compared to cells infected with *Myc* alone or *Myc* and *MnSOD*^{K68R}. In addition, a new Figure S3 has been added to present the data showing that enforced expression of *MnSOD*^{K68Q} in *MnSOD*^{-/-} MEFs, which have no MnSOD activity, resulted in a more transformed tissue culture (*in vitro*) phenotype, as measured by increased growth in soft agar, increased growth when plated at low density, and a decrease in tumor cell doubling time. These results show that the transformation-permissive phenotype in cells expressing *MnSOD*^{K68Q} is not simply due to the loss of the tetrameric form of MnSOD, as well as its detoxification activity. A new Supplemental Figure S4 has also been added that shows that monomeric MnSOD-K68-Ac, as isolated by size exclusion chromatography, shows little to no MnSOD activity. Figures 7b, c, and d have been moved into the Supplement as figures S6a, b, c, to accommodate the addition of a new Figure 4 into the manuscript. Finally, a new Supplemental Figure S9 has been added to show the: (1) control experiments for the MCF7-HTR cells that were constructed that will inducibly express *MnSOD*^{K68R} upon doxycycline exposure; and (2) additional control data from the tissue microarrays stained with the anti-MnSOD-K68-Ac antibody and the raw western blot data.

Reviewer #1, General Comments:

Zhu et al. showed that MnSODK68Q promoted transformation and cancer cell proliferation, but MnSODK68R did not. K68Q and SIRT3 inactivation prevented the formation of SOD tetramer. K68Q gained peroxidase activity and its transformation activity could be rescued by mitochondrial catalase. K68Q overexpression conferred reduced SOD activity and increased oxidative stress compared to WT or K68R. K68Q or SIRT3 inactivation resulted in hydroxyl tam resistance. Hydroxyl tam resistant cells had increased level of K68 acetylation, reduced SOD activity, oxidative stress, reduced tetramer, increased proliferation and tumorigenesis, some of which could be rescued by K68R but not K68Q. SIRT3 expression was reduced and K68-acetylation was increased in luminal B breast cancer patient samples compared to luminal A. The authors concluded that acetylation switches MnSOD from anti-tumorigenic homotetramer to a tumor-promoting monomer. In recent years, acetylation has been increasingly recognized as a posttranslational modification of numerous proteins. There is much interest in understanding the physiological implications of this PTM and the molecular mechanism by which this PTM changes protein function. While there have been some indirect evidence implicating the physiological relevance of protein acetylation, direct evidence is rare. This study provides one of the first direct evidence that protein acetylation is physiologically relevant, by testing MnSOD mutants that mimic acetylated or deacetylated forms in tumorigenesis. Mechanistically, this paper proposes that acetylation affects the protein oligomerization and intriguingly, converting a protein to possess a different enzymatic activity and elicit opposing effect in tumorigenesis. The mechanism is novel. In light of these conceptual advances, this study is a good candidate for Nature Communications. Overall, the experiments are well rationalized and data are robust. This reviewer is particular impressed that some experiments were tested in different cell lines. I recommend its publication after minor revision.

Reviewer #1, Specific Comments:

Concern # 1 - *The introduction needs to be improved. Specifically, line 75: “Studies have shown that SIRT3-guided deacetylation at MnSOD lysines 68 (K68)6 and 122 (K122 directs MnSOD*

enzymatic activity to protect cells from mitochondrial ROS-induced cellular damage.” K53 and K89 have also been identified as acetylation sites targeted by SIRT3 (Qiu, *Cell Metabolism* 2010) and they have been shown to be physiologically important, i.e. in maintaining stem cell function (Brown, *Cell Reports* 2013). This issues is also addressed by reviewer number two who wrote “Along this line, it is reported that MnSOD acetylated at K122 is not a direct target for Sirt3 (Knyphausen et al., 2016). The authors should be more precise when writing that acetylated K122 in MnSOD is deacetylated by Sirt3. They should at least write that K122-Ac MnSOD is not a direct target”.

We appreciate these two reviewers’ concerns about this issue, which we totally agree needs to be addressed in the manuscript in some way that distinguishes the cell biological, biochemical, and phenotypic effects of these four identified MnSOD lysines, as published in Qiu, *Cell Metabolism* 2010; Tao, *Mol. Cell* 2010; Brown, *Cell Reports* 2013; Vassilopoulos, *ARS* 2014; and Knyphausen et al., 2016, *J. Biol. Chem.* As both reviewers clearly know, the issue with these four lysines, and their biochemical and physiological roles in SIRT3 biology, is likely due to the different methods and reagents used in these publications. As such, and I hope the reviewers agree, I think it might be best to present, in general terms, what has been published, and attempt to be as circumspect as possible about what has been previously shown. To address this specific issue in the Introduction, the following sentence has been removed from page 3, line 14 to line 16: “Studies have shown that SIRT3-guided deacetylation at MnSOD lysines 68 (K68)⁶ and 122 (K122)^{8,9} directs MnSOD enzymatic activity to protect cells from mitochondrial ROS-induced cellular damage.” In addition, two new sentences have replaced the original one on page 3, line 12 to line 18: “**Four MnSOD lysines have been identified as biochemical and/or physiological SIRT3 deacetylation targets—K53, K89^{1, 11}, K68^{6, 12, 13}, and K122^{8, 9}—using different methods, including site directed mutagenesis, physical lysine acetylation, and acetyl-lysine specific monoclonal antibodies. However, the specific cell biological, biochemical, and/or physiological significance of each of these lysines, as well as the underlying molecular mechanism by which they regulate MnSOD detoxification activity and mitochondrial metabolism, remains to be fully determined**”.

Reviewer #1, Specific Comments:

Figure legends need to be improved to ease the reading.

Fig 1 title: “MnSODK68Q overexpression promotes a transformation-permissive phenotype both in vitro and in vivo.” In vitro but not in vivo data are included. The title should delete in vivo.

The legend for Figure 1 has been changed to address this concern and more accurately reflect the data presented. It now reads “**MnSOD^{K68Q} expression promotes a transformation-permissive phenotype in vitro**”.

*Fig 1 legend: “Errors represent ± 1 SEM. **p < 0.01, and ***p < 0.001.” No data shown have error bars and statistical tests. These notes should be removed.*

Thank you for catching these errors. These has been removed from or changed in the figure legend.

*Fig 3 legend: “*p < 0.05, **p < 0.01, and ***p < 0.001.” No data show * or ***. Should be deleted.*

*p < 0.05 and ***p < 0.001 have been removed.

Fig 7 legend: (f) Quantification of the results in S5a. Needs to state what is done in the experiment.

This text in the legend of Figure 7 has been changed to more correctly outline the results. The text, in blue, has been changed to “**(k) The results from figure 7j were quantified as the number of**

MCF7-MnSOD^{K68Q} colonies exceeding 50 cells, without and with Ad-mitoCat infection, per 10 cm dish”.

Fig 8a labels are missing. Which one is MCF7 which one is MCF7 HTR?

This figure has been changed, and Figure 8a is now 8b. In addition, the figure is now labeled as “MCF7 or MCF7-HTR” xenograft, and this new text has been added to the legend of Figure 8: **(a-b)** The control MCF7 and Tam resistant MCF7-HTR cells (5×10^6) were implanted into both hind limbs of nude mice, and tumor volumes were measured every 3 days for 6 weeks. **(b)** Representative images from the tumors that formed in the nude mice of the MCF7 (left panel) and MCF7-HTR (right panel) at the end of 6 weeks are shown.

Reviewer #2, General Comments:

Lysine 68 acetylation regulates dichotomous role of MnSOD: a protective tetrameric detoxification complex versus a monomeric oncoprotein by Zhu et al. The lab of Dr. Gius presents interesting data showing that acetylation regulates MnSOD function. MnSOD has been shown to play a role as a tumor suppressor during initiation of neoplasia. However, it becomes an oncogene product upon tumor progression supporting the development of more aggressive tumors.

Using various cell lines, some of which were derived from patients, Zhu and co-workers analysed the effect of lysine-acetylation, predominantly at lysine 68 (K68), on MnSOD function and its impact on tumor development. They derived an interesting model suggesting that acetylation drives the formation of MnSOD monomers showing peroxidase activity, while the non-acetylated form exists as a tetramer with superoxide dismutase (SOD) activity. It is only the monomeric form that is on the one hand oncogenic and on the other hand promotes a Tamoxifen-resistance phenotype.

Overall, the experiments performed were sound and the results obtained are highly valuable for a huge scientific community. Zhu and co-workers suggest a novel role of post-translational lysine-acetylation to confer a gain-of-function, which is rarely the case, and furthermore they open up a novel therapeutic approach for tumor therapy approaching lysine-acetylation. However, there are several issues that would need to be addressed to significantly improve the manuscript prior to publication. I support a publication in Nature Communications if the authors, without exception, address all the major points shown below.

Reviewer #2, Specific Comments:

Concern # 1 - *Although lysine 68 in MnSOD seems to be an important residue for MnSOD function, the major point of criticism of this manuscript is that Zhu and co-workers used lysine to glutamine (as acetylation mimic) and lysine to arginine (to conserve the non-acetylated state) mutations throughout. It is shown that these mutations can be poor mimics of lysine-acetylation at the molecular level. Therefore, the authors have to show that the MnSOD K68Q and K68R mutant proteins are reliable to study lysine-acetylation. One way would be to produce site-specifically lysine-acetylated MnSOD protein using the genetic-code expansion concept (see Knypausen et. al., 2016). It is essential to show how acetylation at K68, the K68Q and the K68R protein interferes with MnSOD oligomerisation and catalytic activity. Analytical size-exclusion chromatography runs (or alternative methods) combined with enzymatic assays to show peroxidase and superoxide dismutase activity of the MnSOD proteins (MnSOD wild-type, acetylated at K68, K68Q and K68R) should be performed. These data will provide evidence for the hypothesis that the oligomerisation state of MnSOD is affected by K68-acetylation and how this determines MnSOD enzymatic activity (monomer: peroxidase; tetramer: superoxide dismutase). These experiments are essential as they show the validity of the central message of the paper. Zhu and co-workers describe an influence of*

lysine-acetylation but did not perform a single experiment with acetylated MnSOD protein. Without showing that the acetylated protein is doing what they suggest by performing in vitro experiments, most paragraphs entitled with “Acetylation of MnSOD.” must be rephrased in saying that mutation of K68Q in MnSOD results in a certain phenotype.

We appreciate the reviewer for raising this issue and suggesting the appropriate experiments that will improve this manuscript. The primary point in concern #1 is that the *MnSOD*^{K68Q} mutant may not physiologically reflect the actual physical acetylation of MnSOD at lysine 68. In this regard, the reviewer is completely correct, and we should have used more precise text to distinguish between expression of the *MnSOD* mutant (*MnSOD*^{K68Q}), which is an oncogene based on our *in vitro* and xenograft experiments, and actual MnSOD acetylation at lysine 68. As such, we have gone through the manuscript and changed the text (which is in blue font) to be more concise in describing these experiments and more precise when we are describing *MnSOD*^{K68Q} as an oncogene. We have also completely removed the phrase “tumor promoter”, replacing it with “oncogene”.

We have added new experiments to address concern #1, using two different methods to experimentally show “*how acetylation at K68 ... interferes with MnSOD ... catalytic activity*”. As suggested by the reviewer, first we have used a bacterial expression system (a kind gift from Dr. Jiangyun Wang, Institute of Biophysics, Chinese Academy of Sciences, Beijing, China) to isolate MnSOD protein with K68 physically acetylated at K68. We also used a size exclusion spin column to separate samples into fractions either above or below 50 kDa, and western blotting with anti-MnSOD and MnSOD-K68-Ac antibodies to validate this method.

In addition, we used a second method to isolate acetylated MnSOD, though it is not quite as controlled and rigorous as the bacterial expression system. For these experiments, we used an established tissue culture expression system in our laboratory where immortalized MnSOD^{-/-} MEFs are transfected with a vector expressing *FLAG-MnSOD*^{WT} in the presence of 10 mM nicotinamide (NAM) and 1 μM trichostatin A (TSA), to inhibit sirtuin deacetylase activity and thereby *enrich* these cells for acetylated MnSOD. In addition, transfected cells were also grown in 10 mM NAD⁺ to activate sirtuin activity and enrich these samples for deacetylated MnSOD. The exogenously expressed MnSOD was immunoprecipitated, and a 50 kDa protein size exclusion column was subsequently used to separate samples into fractions either above or below 50 kDa. These samples were also immunoblotted with anti-MnSOD, MnSOD-K68-Ac, and actin antibodies. In the text we stress that this model “enriches for MnSOD-Ac” but is not as rigorous as the bacterial system; however, it allows us to both enrich MnSOD-Ac and run a semi-native gel from identical samples.

The results of these two different methods to isolate MnSOD where K68 is either physically acetylated (bacterial expression system) or enriched for K68 acetylation (transfection expression system) uniformly confirm our previous data and extend these results to more rigorously make this scientific argument. The results of these experiments are now presented in a new Figure 4. The corresponding Results section is titled “*MnSOD-K68-Ac exhibits peroxidase enzymatic activity*”. We have also added four new paragraphs to the manuscript from page 9, line 9 to page 11, line 6 describing this additional data: “The data presented above showed enrichment of monomeric MnSOD (Fig. 3a) and peroxidase activity (Fig. 3c) upon expression of *MnSOD*^{K68Q} to mimic K68-Ac. However, it is also essential to show how the physical acetylation of K68 affects enzymatic activity. To initially address this issue, an established tissue culture system was used that enriches for acetylated, versus deacetylated, K68. This system transfects MnSOD^{-/-} MEFs with *FLAG-MnSOD*^{WT} followed by exposure to (i) 10 mM nicotinamide (NAM) and 1 μM trichostatin A (TSA), to inhibit SIRT3 deacetylase activity and enrich for K68-Ac, or (ii) 10 mM NAD⁺, to activate SIRT3 activity and enrich for deacetylated K68. As expected, whole cell extracts harvested 40 h after transfection and IPed with an anti-FLAG antibody showed that NAM/TSA exposure increased

MnSOD-K68-Ac (Fig. 4a, top row, left two lanes), while NAD⁺ exposure minimized MnSOD-K68-Ac (right two lanes). The MnSOD-K68-Ac antibody specificity was validated by two different methods (Supplemental Fig. S4a-b). Similar results were observed in 293T cells (Supplemental Fig. S4c).

These samples were subsequently separated, using a spin column, into fractions above or below 50 kDa. Immunoblotting with an anti-MnSOD antibody, the sample from cells grown in NAM/TSA showed an enrichment of MnSOD in the <50 kDa fraction, suggesting that most of the MnSOD is in the monomeric form, with minimal MnSOD in the >50 kDa fraction (Fig. 4a, 2nd and 3rd row, left two lanes). The enrichment of the monomeric MnSOD was confirmed when the <50 kDa fraction was run on a semi-native gel followed by immunoblotting for MnSOD (Supplemental Fig. S4d, left panel, left two lanes) with minimal tetrameric MnSOD in the >50 kDa fraction (right panel, left two lanes). In contrast, MnSOD from cells grown in NAD⁺ showed increased levels of MnSOD in the >50 kDa fraction (Fig. 4a, 2nd and 3rd row, right two lanes), with enrichment of tetrameric MnSOD in the >50 kDa fraction (Supplemental Fig. S4d, right panel, right two lanes). These experiments confirm that samples enriched for MnSOD-K68-Ac contain predominantly monomeric MnSOD, and those with deacetylated MnSOD-K68 contain predominantly tetrameric MnSOD.

Biochemical analysis of the <50 kDa fraction from cells exposed to NAM /TSA (i.e., enriched for MnSOD-K68-Ac and monomeric MnSOD) showed elevated peroxidase activity compared to the <50 kDa fraction from cells treated with NAD⁺ (Fig. 4b). In contrast, MnSOD from both <50 kDa fractions exhibited minimal MnSOD detoxification activity (Fig. 4c). Analysis of the >50 kDa fraction from cells treated with NAD⁺ (i.e., enriched for tetrameric MnSOD) exhibited elevated MnSOD detoxification activity compared to cells exposed to NAM/TSA (Fig. 4d). As expected, there was little MnSOD peroxidase activity in the >50 kDa fraction from cells treated with either NAD⁺ or NAM/TSA (Supplemental Fig. S4e).

A second, and more rigorous, method was also used to determine how the physical acetylation of MnSOD-K68 alters enzymatic activity by transforming *E. coli* with both pEVOL-AcKRS, which expresses an acetyl-lysyl-tRNA synthetase/tRNA^{CUA} pair from *M. barkeri*, and pET21a-MnSOD^{K68TAG}, a MnSOD bacterial expression vector that allows the site-specific incorporation of N-(ε)-acetyl-l-lysine into K68. The bacterially expressed protein from the control (i.e., expressing WT MnSOD) and pET21a-MnSOD^{K68TAG} plasmids was nickel column purified, and the acetylation of MnSOD was confirmed via immunoblotting with our anti-MnSOD-K68-Ac antibody (Fig. 4e). Finally, the purified protein samples from the cells transformed with pEVOL-AcKRS and pET21a-MnSOD^{K68TAG} exhibited a significant increase in peroxidase activity (Fig. 4f), as compared to purified protein samples from bacterial cells expressing the control expression plasmid, pET21a-MnSOD. The biochemical results show two different methods to isolate MnSOD where K68 is either physically acetylated (bacterial expression system) or enriched for K68 acetylation (transfection expression system) to confirm that monomeric MnSOD exhibits a new peroxidase enzymatic function”.

Concern # 2 - *Along this line, it is reported that MnSOD acetylated at K122 is not a direct target for Sirt3 (Knyphausen et al., 2016). The authors should be more precise when writing that acetylated K122 in MnSOD is deacetylated by Sirt3. They should at least write that K122-acetylated MnSOD is not a direct target. Using acetylated MnSOD K68 protein, the authors should analyse if Sirt3 deacetylates it in vitro. This will show if K68-acetylated MnSOD is a direct target for Sirt3.*

To address this question, and as described above, two new sentences have replaced the original one on page 3, line 12 to line 20: “Four MnSOD lysines have been identified, using either direct or indirect experimental techniques, as biochemical and/or physiological SIRT3 deacetylation targets—K53, K89^{1, 11}, K68^{6, 12, 13}, and K122^{8, 9}—using different methods, including site directed

mutagenesis, physical lysine acetylation, and acetyl-lysine specific monoclonal antibodies. However, the specific cell biological, biochemical, and/or physiological significance of each of these lysines, as well as the underlying molecular mechanism by which they regulate MnSOD detoxification activity and mitochondrial metabolism, remains to be fully determined”.

Finally, I am very open to other thoughts about how to present this concern in the Introduction, since I think this is a very important issue, as well as one that is complex and should be described in a way that respectfully but correctly presents what has been published by the various groups in the past.

Concern # 3a - *The authors show acetylated MnSOD K68 protein by western blotting, if I understood correctly by using a specific anti-MnSOD-K68-ac antibody, suggesting that K68-acetylated MnSOD is present. The authors should show the reliability of the antibody using purified non-acetylated and K68-acetylated MnSOD protein. However, it is not clear if this is a Sirt3 regulated site and in Fig. 5d there is no obvious difference in Sirt3 levels visible. Using nicotinamide to inhibit Sirt3 will show if the K68-acetylation is regulated by sirtuins.*

The construction of the MnSOD-K68-Ac antibody is complicated, and we initially made this antibody in a collaboration with Epitomics, Inc - The Rabbit Monoclonal Antibody Company, where the laboratory paid for making the antibody, and the company had the commercial rights to sell the antibody. We have made five antibodies with this company. Since then, Epitomics was bought by Abcam, who now sells this antibody. Of all our antibodies, MnSOD-K68-Ac appears to have the highest specificity.

To address the concern in regards to this antibody, we have added two new figure panels to the Supplement. Figure S4a shows the standard method we have used in the past, and published, to validate our lysine-specific anti-acetyl antibodies. For these experiments (Fig. S4a), Flag-tagged MnSOD expression vector was transfected into HEK 293T cells with TSA (1 μ M), and after 48 h Flag-MnSOD was IPed, and samples were incubated with purified SIRT3 protein without (lane 1) or with (lane 2) NAD (left panel). After 2 h, mixtures were immunoblotted with the anti-MnSOD-K68-Ac antibody, which showed a significant decrease in immunoreactive protein levels in extracts where SIRT3 and NAD⁺ were added. The middle immunoblot shows that a control peptide did not change the western blot, while in contrast, when samples were incubated with the peptide used to make the antibody (i.e., a 13 amino acid peptide with K68-Ac at position 7), the immunoreactive MnSOD band is no longer seen (right panel).

Experiments were also done with liver extracts from mice on an *ad libitum* diet versus caloric restriction (i.e., to activate SIRT3), and these results showed a decrease in MnSOD-K68-Ac that is shown in Figure S4b. These data are included to show a decrease in MnSOD-K68-Ac under physiological conditions thought to activate SIRT3, leading to MnSOD deacetylation. Lastly, we have included an immunoblot, which we did in in collaboration with Abcam, Inc. I have spoken to the post-doctoral fellow, and it is my understanding that we worked with them

to generate this Immunoblot using the anti-MnSOD-K68-Ac antibody. This western blot (shown to the right) shows that the addition of the 13 amino acid peptide used to make the antibody, similar to the experiments shown in Figure S4a, (<http://www.abcam.com/sod2mnsod-acetyl-k68-antibody-epvanr2-ab137037.html>) also decreased the immunoreactive levels using the MnSOD-K68-Ac antibody. Finally, text has been added to the manuscript to direct the reader to the data in Figure S4 that shows the specificity of the MnSOD-K68-Ac antibody on page 9, line 19 to line 20: “The MnSOD-K68-Ac antibody

specificity was validated by two different methods (Supplemental Section, Figs. S4a-b)".

Concern # 3b - *The authors describe that mutation of K68Q in MnSOD results in monomerisation of the protein. For the in vivo impact of the acetylation at K68 it is essential that the acetylation occurs at sufficiently high stoichiometries. Using K68Q mutant, the authors obviously establish a constitutively acetylated state of MnSOD. It is of course not clear which stoichiometries of K68-acetylation in MnSOD are achieved under physiological conditions. The authors could at least discuss this point.*

We agree with reviewer two, and I think this an important question in regards to our work. In this regard, we have added text to both the introduction and then the discussion. In the introduction we raise this question starting on page 3, line 22 to page 4, line 2 stating "Mice lacking *Sirt3*, and thus containing acetylated MnSOD (MnSOD-Ac), developed tumors", implying that SIRT3 may function as a tumor suppressor. Thus, this raises a key SIRT3 biological question: what is the *in vivo* impact of MnSOD-Ac and how does elevated, and/or aberrant, stoichiometric levels disrupt normal mitochondrial metabolism leading to a cellular damage and/or tumor permissive murine phenotype?".

We have also added text to the discussion to try to provide an potential explanation to this question starting on page 19, line 3 to line 11 stating "These results presented in this manuscript also raise an important question: what is the *in vivo* impact of the acetylation at K68 and how do changes in stoichiometry of MnSOD-K68-Ac drive biological processes? In this regard, one possible explanation might be that MnSOD exists as in a dynamic tetrameric:monomeric equilibrium based on the K68-Ac. Since it is well-established that MnSOD acetylation responds to changes in nutrient availability, such as CR¹², it could be proposed that the stoichiometry of MnSOD-K68-Ac directs this tetrameric:monomeric equilibrium to program mitochondrial energy pathways to match organismal physiological conditions, including fasting or feasting. However, there is much to be done to determine if this idea is correct and the physiological consequence of such a hypothesis".

Concern # 4 - *Along the line of point 3, the stoichiometry of acetylation does not play a major role if the acetylation event creates a gain-of-function rather than a loss-of-function. In this context, the authors say that K68-acetylation of MnSOD is a gain-of-function modification. However, from my point of view the authors cannot deduce this from their data. It might be a gain-of-function, i.e. MnSOD monomerisation and increased peroxidase activity, which results in the phenotype observed. However, it can likewise be the loss in MnSOD oligomerisation and loss in SOD activity that supports the more tumorigenic phenotype. Therefore, the authors have to present additional data to show that it is the gain in peroxidase activity or increase in monomeric MnSOD that supports tumor development and not loss in MnSOD tetramerisation or loss in superoxide dismutase activity of MnSOD before making such statements.*

As noted above, I agree and I think "gain of function" is not the correct way to present this data. The phrase "gain of function" has been removed, and the existing text has been changed throughout the manuscript to more precisely describe this data. In addition, to address the concern that it is the "increase in monomeric MnSOD that supports tumor development and not loss in MnSOD tetramerisation", we have added additional data using MEFs where MnSOD was genetically deleted and *a priori* have no MnSOD activity. These experiments showed that enforced expression of *MnSOD*^{K68Q}, by infecting *MnSOD*^{-/-} MEFs with lenti-*MnSOD*^{K68Q}, led to a more transformed *in vitro* phenotype, as measured by growth in soft agar, contact inhibition, and cell doubling times, suggesting that *MnSOD*^{K68Q} is a tissue culture oncogene. This data is included in the new

Supplemental Figure S3a-d. In addition, new text has been added to the manuscript starting on page 8, line 19 to line 24: “These experiments showed that MnSOD^{-/-} MEFs infected with only lenti-MnSOD^{K68Q}, (i.e., a single oncogene) exhibited a more transformed phenotype (Figure 3e, middle column), as compared to cells infected with lenti-empty vector, lenti-MnSOD^{WT}, or lenti-MnSOD^{K68R} (Supplemental Fig. S3a), as measured by growth in soft agar, contact inhibition, and doubling time (Supplemental Fig. S3b-d). Since these results were done in cells lacking MnSOD, it seems reasonable to suggest that MnSOD^{K68Q} functions as an *in vitro* oncogene”.

Finally, directly above this text on page 8, line 17, we have replaced the phrase “potential gain of function oncoprotein” to “is a potential oncogene”. Since these experiments were done by infection with lenti-MnSOD^{K68Q}, and as the reviewer points out, “oncogene” is the more correct and precise way to present data from *in vitro* transformation experiments. This change has been made throughout the manuscript.

Concern # 5 - *The authors use Ras and c-Myc to immortalise or transform primary cells. When did they use the oncogenic G12V Ras and when wildtype Ras?*

To address this concern, we have added new text to more specifically state when we are using wild-type *Ras* versus the oncogenic *Ras* gene. As such, on page 5, line 18, text was added stating “(WT *Ras* gene)” as well as text on page 6, line 7, stating “(i.e., the oncogenic *Kras* gene)”.

Concern # 6 - *The authors say that monomeric MnSOD, driven by acetylation at K68, is more oncogenic and furthermore supports the Tamoxifen-resistance phenotype in MCF7 and T47D cells. They concluded this from observing a higher level on K68-acetylated MnSOD in these cells. The mechanism underlying this development of Tamoxifen-resistance is not clear and it is also not clear from the data presented what, acetylation or Tamoxifen-resistance, is cause and effect and if there is a direct relationship. Either the authors perform additional experiments to show if acetylation of MnSOD at K68 increases Tamoxifen-resistance or the other way around Tamoxifen-resistance drives acetylation of MnSOD at K68, or they rephrase it in the manuscript accordingly.*

Upon close re-reading we must agree with the reviewer that our results do not, *a priori*, show a cause and effect or a direct relationship. To address this, we have changed the text to more clearly present our results that are really showing that MCF7 and T47D cells selected for resistance to tamoxifen exhibit a MnSOD-K68-Ac axis signature. Thus, the following changes have been made starting on page 12, line 14, where the section heading has been changed from “MnSOD-K68-Ac leads to hydroxy-Tam resistance in breast cancer cells” to “**Hydroxy-Tam-resistant breast cancer cells exhibit a MnSOD-K68-Ac signature**”.

In addition, we have changed text throughout the manuscript to highlight that the Tam-resistant MCF7 and T47D cells exhibit a MnSOD-K68-Ac signature. Text to address this has been added to: (1) page 12, line 22 to line 24: “The results of these experiments suggest that ER+ breast cancer cell lines selected for resistance to Tam exhibit a MnSOD-K68-Ac signature, which may also serve as a potential molecular biomarker”; (2) page 13, lines 3 to 5: “To further show that the MnSOD-K68-Ac is a potential Tam resistance signature / biomarker, HTR cells were infected with lenti-MnSOD^{WT}, lenti-MnSOD^{K68Q}, and lenti-MnSOD^{K68R}, and hydroxy-Tam resistance was measured by clonogenic cell survival assays”; (3) page 13, lines 11 to 12: “These results suggest that MnSOD-K68-Ac is a potential molecular biomarker and/or tumor signature for resistance to Tam”; (4) page 14, line 6 to line 8: “These data suggest that hydroxy-Tam treatment increases MnSOD-K68-Ac suggesting that there may be a Tam resistance tumor signature that also includes changes in cellular ROS profiles, which has been shown by others^{24,31}”; and (5) page 15, line 1 to line 2:

“These results suggest that, either indirectly or directly, that cells expressing the MnSOD acetylation mutant require hydrogen peroxide to maintain resistance to Tam”.

Concern # 7 - *More aggressive tumors show elevated levels of MnSOD. This would drive the formation of MnSOD oligomers. According to the model presented by Zhu et al., acetylation of MnSOD counteracts this oligomerisation. The authors should show how the total MnSOD levels and the levels of K68-acetylation correlate with different tumor cells of various aggressiveness. The acetylation of MnSOD must counteract the concentration dependent oligomerisation of MnSOD, which might occur upon increase in MnSOD levels.*

It is mechanistically difficult to understand why only overexpressed MnSOD shows elevated peroxidase activity. How can the author mechanistically explain this? Is only the overexpressed MnSOD acetylated and this in turn leads to its monomerisation and subsequent increase in peroxidase activity?

The reviewer is correct to inquire if more aggressive tumors exhibit increased MnSOD levels. In this regard, we have a manuscript in revision at *Nature Communications*, done in collaboration with Marcelo Bonini (He C, Hart PC, Zhu Y, Luelsdorf de Abreu A, Patel R, O'Brien J, Kastrati I, Tang B, Frasar J, Wakefield L, Ganini da Silva D, Gius D, and Bonini MG. Acetylation activates an alternative function of SOD2. *Nature Commun.* In revision.) I realize this is a less-than-ideal response to the reviewer's concern # 7, since he/she does not have access to this manuscript. However, the model suggested in this manuscript is that when the total levels of MnSOD accumulate in the cell, it can overwhelm the ability of SIRT3 to keep the protein in the deacetylated state, leading to an increase in the acetylated form of MnSOD, pushing the tetramer:monomer equilibrium towards the monomeric form, which dysregulates mitochondrial metabolism, as well as favoring a transformation-permissive phenotype. Finally, I would suggest that the new experiments presented in the new Figure S3a-d, as well as the new Figure 4, also address this idea.

Concern # 8 - *The authors performed crosslinking experiments using glutaraldehyde to show a MnSOD tetramerisation and monomerisation. However, this is only an indirect evidence for the oligomeric state as mutation of K68Q or K68 might affect other processes such as subcellular localisation or something else. The fact that they even observe MnSOD dimers, which I think do not occur in vivo, shows that using this assay the physiologically relevant forms cannot be represented. Proper analytical in vitro experiments such as ultracentrifugation, size-exclusion chromatography, isothermal titration calorimetry, etc. should be performed to show the oligomeric state?*

The reviewer is correct that when we use glutaraldehyde, we tend to see the addition of a dimer that is not seen using semi-native gel separation, and we agree this clearly is an artifact due to the harsher glutaraldehyde method to crosslink proteins. To address this concern, we have used the samples from the new experiments shown in Figure 4, isolated using a size exclusion column, and run a semi-native gel that shows only a tetramer from IPed Flag-MnSOD in cells grown in NAD⁺ in the >50 kDa fraction. In contrast, only a monomer is present for IPed Flag-MnSOD isolated from cells grown in NAM /TSA. This data is presented in the new Supplemental Figure S4d. Corresponding text has been added to the manuscript on page 10, line 2 to line 5: “The enrichment of the monomeric MnSOD was confirmed when the <50 kDa fraction was run on a semi-native gel followed by immunoblotting for MnSOD (Supplemental Fig. S4d, left panel, left two lanes) with minimal tetrameric MnSOD in the >50 kDa fraction (right panel, left two lanes)”.

Concern # 9 - *Statements that peroxidase activity needs acetylation at K68 must be rewritten to be a bit more circumspect. There is no evidence that acetylation at K68 is needed for peroxidase activity. Maybe it is the monomerisation, which elevates MnSOD peroxidase activity, but this*

monomerisation could also be driven by other events such as other post-translational modifications, etc.

There are other statements in the manuscript that need also some rewriting to be more precise and circumspect. As an example, at several stages in the manuscript the authors say that MnSOD is a Sirt3 target (at various acetylation sites such as K122-ac, K139-ac, K68-ac) but there is no experiment showing this. Did the authors show at all, that Sirt3 knock-out or knock-down increases acetylation at MnSOD K68-ac? If not, they should at least do this to show that MnSOD K68-acetylation is either directly or indirectly affected by Sirt3.

The author's state that their results show that acetylation at K68 in MnSOD promotes, at least in some part, a Tamoxifen-resistance phenotype. As shown above this statement is too strong and a proof is missing. The authors should be more concise and more circumspect with these kind of statements throughout their manuscript.

As written above we agree that more concise text is required to more correctly and/or precisely describe the results of the experiments in this manuscript, and we have gone through the text to address these issues. As one example, the phrase “Tam resistance permissive phenotype” or “Tam resistance phenotype” no longer appears in the text, and as discussed above, we have presented this data as a Tam resistance signature that I think more accurately describes our results. In addition, we have previously shown that MnSOD K122 (Tao et al., 2010, *Cancer Cell*) and K139 (Vassilopoulos et al., 2010, *ARS*) are SIRT3 deacetylation targets. For K68-Ac, please see the text above in reviewer 2, concern # 3. We have also tried to be more careful presenting this data, see Concern # 2 above.

Text to specifically address this has been added the legend of Supplemental Figure S5: “The cell lysates were analyzed by immunoblotting with anti-MnSOD-K122-Ac (validated as a SIRT3 deacetylation target in Tao et al., 2010, *Cancer Cell*), anti-MnSOD, anti-OSCP-K139-Ac (validated as a SIRT3 deacetylation target in Tao et al., 2010, *Cancer Cell*), anti-OSCP, anti-IDH2-K413-Ac (validated as a SIRT3 deacetylation target in Someya et al., 2010, *Cancer Cell*), anti-IDH2 and anti-actin.”

Concern # 10 - *The authors performed an experiment expressing a mitochondrial catalase to reverse a Tamoxifen-resistance phenotype in a MnSOD K68Q background. It is not clear what the addition of catalase should have for an effect. The MnSOD K68Q cells should produce peroxidase which anyway removes hydrogen-peroxide, would it not? Could the authors explain the experiment they performed? Would it not be more suitable to increase hydrogen-peroxide levels and analyse if is possible to reverse the impact of MnSOD K68Q on Tamoxifen-resistance? How would a catalytically dead MnSOD K68Q dead mutant behave? These experiments will show the impact of MnSOD peroxidase activity on development of Tamoxifen-resistance.*

We should have presented a clearer rationale for the experiments shown in Figure 7, as well as a more concise presentation of these results. The rationale for these experiments is that mitochondrial peroxidases require, *a priori*, hydrogen peroxide as a necessary substrate for enzymatic activity. Thus, significantly decreasing mitochondrial hydrogen peroxide levels, due to infection with Ad-mitoCat, will decrease mitochondrial peroxidase activity, converting the necessary substrate, H₂O₂, to O₂ and H₂O. Thus, if MnSOD-K68Q is a peroxidase, then Ad-mitoCat should decrease its activity, as well as Tam resistance, if this activity, either indirectly or directly, is required for the development of Tam resistance. Thus, by removing H₂O₂ through the use of Ad-mitoCat, this experiment functionally yields the same result as using a catalytically dead MnSOD. Without H₂O₂, MnSOD^{K68Q} does not have the substrate to function as a peroxidase. Lastly, we have made revisions

to the text that we believe now more precisely describes the results of these experiments. See page 14, line 19 to page 15 line 2: “To determine if hydrogen peroxide is necessary for the HTR observed in the MCF7-MnSOD^{K68Q} cells, we infected these cells with AdMitoCat, which will remove and/or significantly reduce mitochondrial hydrogen peroxide levels, a critical and necessary substrate for peroxidase enzymatic activity. The results of clonogenic cell survival experiments demonstrated that decreased mitochondrial hydrogen peroxide levels reversed the HTR observed in MCF7 cells that constitutively express MnSOD^{K68Q} (Fig. 7j,k). These results suggest that, either indirectly or directly, cells expressing the MnSOD acetylation mutant require hydrogen peroxide to maintain resistance to Tam”.

Concern # 11 - *For many experiments, particularly the western blotting, the primary data were not shown. As an example, Fig. 8 shows quantifications of Sirt3 levels in luminal A and B cells. Please provide the primary data (also for other experiments) and show it at least in the supplementary section.*

In this regard, we should have been clearer that the results in Figure 8 were obtained with a commercial tissue microarray, and the results in Figure 8d are two images from the IHC staining, while Figures 8e and 8f are the computer imaging quantification of the staining intensity. However, the reviewer is correct that we should have shown the primary data used to generate the bar graphs in Figures 8e and 8f, which have now been added into a new Supplemental Figure S9c,d. Text describing this is on page 15, line 24 to page 16, line 3 in the manuscript: “The TMA was stained using anti-MnSOD-K68-Ac (see Supplemental Section, Fig. 4a,b for antibody specificity) and anti-SIRT3 antibodies, and representative IHC images for luminal A and B tumor samples are shown (Fig. 8d and Supplemental Fig. S9c,d)”.

In addition, we have added a bit more text in regards to stratification of the staining intensities into low, intermediate, and high staining for luminal A, as compared to luminal B tumors, using the MnSOD-K68-Ac antibody. I think this more clearly shows that there is a subgroup of luminal B breast tumors that exhibit a MnSOD-K68-Ac signature. New text describing these results is on page 16, lines 6 to 9: “In addition, stratification of the staining intensities from the luminal A versus luminal B TMAs into low, intermediate, and high staining suggests that there may be a subgroup of luminal B tumors that exhibit significant MnSOD-K68-Ac staining (Supplemental Fig. S9c,d)”.

Concern # 12 - *The authors say in the discussion that they show that MnSOD K68 is a physiologically relevant Sirt3 target. This is exaggeration and as stated above I do not find any experiment to show this. Furthermore, they claim that K68-acetylation this is a gain-of-function modification. This might be the case and this is an interesting issue. However, as stated above the evidence is missing and the phenotype on tumor progression might also be explainable by a loss-of-MnSOD-function. The phenotype observed might be explained by a loss of superoxide detoxification upon acetylation of MnSOD and the following loss in superoxide dismutase activity resulting for example in an increase in genomic instability.*

A figure showing the structure of MnSOD and the position of K68 would be helpful to envision a potential role of K68-acetylation on MnSOD function.

The statement that acetylation of MnSOD at K68, which affects MnSOD assembly and function, reprogramming mitochondrial function is exaggerated.

Please, be more circumspect with your statements and the conclusions drawn from the results throughout the manuscript.

We agree that we need to be more precise and circumspect with our language throughout the manuscript, and we have altered our statements throughout the manuscript (in blue font) to address this as described in more detail above. A new Supplemental Figure S10 has been added describing

the location of K68 within MnSOD. Text outlining this is on page 18, line 15: “(see Supplemental Fig. S10)” and see directly below.

The text regarding mitochondrial reprogramming has been removed from the manuscript. In addition, the phrase “However, the molecular mechanism regarding how MnSOD acetylation contributes to reprogramming the mitochondrial metabolism”, has been changed to “**However, the molecular mechanism regarding how MnSOD acetylation contributes to the regulation of MnSOD activity is still somewhat unclear**” on page 17, line 8 to line 9. In fact, the word “reprogramming” no longer appears in the Results or Discussion, since I think the reviewer is correct that the phrase “mitochondrial reprogramming” is vague. In addition, and as discussed above, “gain of function” has also been removed from the text and replaced with more concise text throughout the manuscript.

Finally, we have removed and changed this portion of the Discussion: “The positive charge of these lysine residues may play an important role in MnSOD tetrameric assembly, stability, and substrate recognition. Thus, our results provide a new mechanistic rationale by which MnSOD, via K68-Ac, alters MnSOD assembly as well as function, which reprograms mitochondrial function” to “**Lysine acetylation neutralizes the positive charge of this amino acid, potentially altering the 3-dimensional structure of a protein and its enzymatic function^{3,34}. In this regard, a previously published structural analysis of MnSOD, using Adaptive Poisson-Boltzmann Solver, showed the protein near K68-Ac exhibits decreased positive surface charge, as compared to non-acetylated K68, suggested that K68-Ac modulates the electrostatic potential near the active center³⁵. Our data seem to confirm this idea (see Supplemental Fig. S10), suggesting that K68 may play an important role in the structure-function relationship of MnSOD**” on page 18, lines 10 to 16.

Reviewer #3, Specific Comments:

Concern # 1 - *The study by is technically sound but incremental in its overall contribution to the literature. A role for MnSOD in drug resistance in breast and other cancers is clearly documented in the literature, as is its ability to confer resistance to Tamoxifen. Much of this work is not cited (e.g., see Cho SK et al, Biomaterials, 2013; Fu A et al., Oncotarget, 2016). Moreover, the role of ROS (and blocking with antioxidants like N-acetylcysteine) on ER function and Tamoxifen resistance is known, relevant to the work described here, and yet not cited (e.g., see Nass et al., BBA, 2016; Cook et al FASEBJ, 2014). This study strengthens the body of work already implicating this pathway and MnSOD in Tamoxifen resistance but provides little new regarding the role of oxidative stress in this phenotype. There are few new insights also beyond the studies with the isoforms described, relative to prior work with MnSOD.*

First and foremost, we are profoundly embarrassed that we did not discuss these previously published manuscripts presenting a connection between MnSOD and/or ROS and resistance to tamoxifen. This has now been corrected in the manuscript in multiple places using the reviewer’s suggestions as a guide. As such, the suggested references, as well as Razandi, *Oncogene*, 2013, have been added to text in the Results section to present a more complete and thoughtful rationale for the experiments involving Tam resistance. As such, text to address this has been added to page 11, line 20 to page 12, line 2: “**It has previously been shown that there is a link between dysregulated MnSOD^{28,29,30} and aberrant cellular ROS levels and/or oxidative stress, due to several different mechanisms^{24,31}, and resistance to endocrine therapy. Based on these previous publications, and our results above identifying *MnSOD^{K68Q}* as an *in vitro* oncogene, it seemed reasonable to propose that, similar to other oncogenes, enforced expression of *MnSOD^{K68Q}* may also lead to, either indirectly or directly, resistance to Tam**”.

We have also added text to the Introduction to address this concern on page 5, line 7 to line 11: “In addition, it also appears that, under specific conditions, there is a link between dysregulated MnSOD, aberrant cellular ROS levels^{22,23,24}, and resistance to tamoxifen (Tam)-induced cytotoxicity. These and other findings²⁵ suggest a mechanistic link between mitochondrial redox/ROS balance and the biology of ER+ breast cancer”.

In addition, text has also been added to the Discussion to emphasize this point (page 17, line 19 to line 22): “Finally, our results also suggest that ER+ breast cancer tumor cells, selected for resistance to Tam over 3 months, exhibit a MnSOD-K68-Ac signature, similar to results previously published suggesting a role of MnSOD levels^{28,29,30}, and/or aberrant ROS levels^{24,31}, in endocrine resistance in ER+ tumor cells”.

Concern # 2 - *Many genes make MCF-7 and T47D cells resistant to Tamoxifen when overexpressed but are not actionable clinically. Evidence that the relationships described in the experimental models here are relevant in patients is largely absent from the manuscript and is not directly supported by new data presented here.*

We agree with reviewer 3, as well as reviewer 2, who both raised legitimate concerns that some of our text was not precise in describing the conclusions from the data. To address this, we have reworked the manuscript to modify our conclusions in regards to both the transformation data as well as the experiments involving resistance to Tam cytotoxicity. As such, and as presented above, we have made significant changes to the text throughout the manuscript to highlight that the data from the Tam resistant MCF7 and T47D cells does not suggest a new mechanism of Tam resistance but that these MCF7-HTR and T47D-HTR cells exhibit a MnSOD-K68-Ac signature. As the reviewer suggests, this is more in line with the data and also supports the manuscript’s transition into the human tissue microarray data showing that there may be a subgroup of human luminal B breast malignancies that also exhibit a MnSOD-K68-Ac signature. Finally, to address the reviewer’s concerns the text has been changed on: (1) page 12, line 22 to line 24: “The results of these experiments suggest that ER+ breast cancer cell lines selected for resistance to Tam exhibit a MnSOD-K68-Ac signature, which may also serve as a potential molecular biomarker”; (2) page 13, lines 3 to 5: “To further show that the MnSOD-K68-Ac is a potential Tam resistance signature / biomarker, HTR cells were infected with lenti-MnSOD^{WT}, lenti-MnSOD^{K68Q}, and lenti-MnSOD^{K68R}, and hydroxy-Tam resistance was measured by clonogenic cell survival assays”; (3) page 13, lines 11 to 12: “These results suggest that MnSOD-K68-Ac is a potential molecular biomarker and/or tumor signature for resistance to Tam”; (4) page 14, line 6 to line 8: “These data suggest that hydroxy-Tam treatment increases MnSOD-K68-Ac suggesting that there may be a Tam resistance tumor signature that also includes changes in cellular ROS profiles, which has been shown by others^{24,31}”; and (5) page 15, line 1 to line 2: “These results suggest that, either indirectly or directly, that cells expressing the MnSOD acetylation mutant require hydrogen peroxide to maintain resistance to Tam”.

Concern # 3a - *The animal studies are technically sound but the number of animals/group is small.*

We agree that the animal numbers were low and increasing them will improve the manuscript. To address this, we have repeated the animal studies and increased the total animal number to 10 per group. The subsequent results have improved the statistical analysis of the data, and the results of these additional experiments are consistent with our initial submission.

Concern # 3b - *It is not clear how the statistically analyses of these studies account for the repeated measures nature of the data. While reanalysis may be needed to ensure the most*

appropriate tests are used, for the in vivo studies, this is unlikely to change the outcomes or interpretations.

To address this we have added text to more better describe how the statistical analysis was done in the methods section starting on page, 26 line 16 to 19 starting “Statistical analysis was performed using GraphPad Prism for Windows (GraphPad Software, San Diego, CA). Data were expressed as mean SEM unless otherwise specified. One-way ANOVA analysis with Tukey’s post-analysis was used to study the differences among three or more means. Significance was determined at $p < 0.05$ and the 95% confidence interval”.

Reviewer #4, General Comments:

Overall this is an interesting study- and could be of great interest, but in its current form it is preliminary in nature and lacks proper controls to make its over-reaching conclusions. The connection between SIRT3 and MnSODK68Q is not mechanistically or conceptually well-developed nor is the function of MnSODK68Q as an oncogene rigorously demonstrated.

Reviewer #4, Specific Comments:

Concern # 1 - *There are some basic cancer biology concepts that are incorrectly measured. For example in Figure 1a, the authors are assessing the effects of cooperating oncogenes during cellular immortalization, while in Figure 1b&c soft agar colony formation is not a measure of invasiveness as indicated by the authors but rather a measure of anchorage-independent growth. On p6. The authors refer to MnSOD^{K68Q} as a tumor promoter or an oncogene. Why use both terms?*

Reviewer 4 is correct, as was also noted by reviewer 2, that the text included in the manuscript was not as precise as required to describe the results from the experiments presented. As such, we have added more precise text to refer to the MnSOD mutant (MnSOD^{K68Q}) as an oncogene in our *in vitro* and xenograft experiments, instead of saying MnSOD^{K68Q} is a tumor promoter or acetylation switches MnSOD to a “tumor promoter”. In this regard, the phrase “tumor promoter” no longer appears in the text of the manuscript. We have also revised the text to be more precise when we describe our experiments addressing immortalization of pMEFs versus transformation.

Concern # 2a - *The study is premature as the authors don’t actually conduct any transformation experiments (eg. showing that MnSOD^{K68Q} cooperates to form tumors in vivo).*

To address this, we have used the cells initially presented in Figure 1a for xenograft experiments to determine if MnSOD^{K68Q} can cooperate with Myc to form xenograft tumors in nude mice. These results have been added to the table in Supplemental Figure S1. In addition, text has been added to the manuscript to describe these *in vivo* transformation experiments on page 6, line 4 to line 5: “and the formation of xenograft tumors, a measure of an *in vivo* tumorigenic permissive phenotype (Supplemental Fig. S1a, right column)”.

Concern # 2b - *In addition, the study lacks additional experiments to define the role of MnSOD^{K68Q} during transformation compared with its role in cancer cells once transformed.*

To address this concern, we have added additional *in vitro* transformation experiments and the subsequent data using MEFs where MnSOD was genetically deleted (MnSOD^{-/-}). These cells are immortalized and only require one additional oncogene to become transformed. Thus, the MnSOD^{-/-} MEFs were infected with either lenti-MnSOD^{WT}, lenti-MnSOD^{K68R}, or lenti-MnSOD^{K68Q} and then cultured and selected in puromycin for 14 days. These experiments showed that enforced expression

of $MnSOD^{K68Q}$, as compared to cells infected with lenti- $MnSOD^{WT}$ or lenti- $MnSOD^{K68R}$, can drive a transformation phenotype *in vitro*, as measured by growth in soft agar, contact inhibition, and cell doubling times, suggesting that $MnSOD^{K68Q}$ is an *in vitro* oncogene. This data is included in a new Supplemental Figure S3a-d. In addition, new text has been added to the manuscript to present this data, as well as to more precisely describe the experiments and the results, on page 8, line 19 to line 24: “These experiments showed that $MnSOD^{-/-}$ MEFs infected with only lenti- $MnSOD^{K68Q}$, (i.e., a single oncogene) exhibited a more transformed phenotype (Figure 3e, middle column), as compared to cells infected with lenti-empty vector, lenti- $MnSOD^{WT}$, or lenti- $MnSOD^{K68R}$ (Supplemental Fig. S3a), as measured by growth in soft agar, contact inhibition, and doubling time (Supplemental Fig. S3b-d). Since these results were done in cells lacking $MnSOD$, it seems reasonable to suggest that $MnSOD^{K68Q}$ functions as an *in vitro* oncogene.

Concern # 3 - *Showing that MCF7 cells when infected with $MnSOD^{K68Q}$ grow in nude mice compared to WT or control cells could imply that $MnSOD^{K68RQ}$ allows cells to growth without estrogen supplementation. This does not mean that $MnSOD^{K68Q}$ is an oncogene.*

First, to address this concern, we have removed the text “suggesting that $MnSOD^{K68Q}$ (Ac-mimetic) appears to function as an oncogene” from the sentence on page 6, line 23. We have also repeated the xenograft experiments with the MCF7- $MnSOD^{K68Q}$ cells, without and with estrogen supplementation, and these results showed similar growth rates. This data is included in a new Supplemental Figure 1b, and new text has been added starting on page 7, line 1 to line 5: “These results suggest increased growth characteristics in xenograft tumors that express $MnSOD^{K68Q}$ however, this could also reflect cells exhibiting estrogen-independent growth properties. In this regard, MCF7 cells infected with lenti- $MnSOD^{K68Q}$ were injected into the hind limbs of nude mice, and these xenograft experiments showed that estrogen supplementation did not alter the tumor growth characteristic (Supplemental Section Fig. S1b)”.

Concern # 4 - *Similarly, the increased proliferation conferred in ER+ cancer cell lines by overexpressing $MnSOD^{K68Q}$ could mimic the effects of estrogen supplementation. A control should be done by treating the cells with anti-estrogen (ICI or Tam) in combination as well as comparing Ki67 rates of cells treated with estrogen compared with $MnSODK68Q$.*

This is an excellent point and frankly, one that I had not considered. We have now done this experiment, and similar to the xenograft experiments, we did not see a significant difference in the growth properties in the MCF7- $MnSOD^{K68Q}$ cells when exposed to either estrogen or Tam. I am not, *a priori*, sure this means these cells are estrogen-independent; however, we are working on this idea. In this regard, I would propose that we show these results and simply state the results without making any interpretive statements so that we can expand upon this idea in a future publication, if the reviewer agrees. This data is included in new Supplemental Figure S2c-f. In addition, new text has been added on page 7 line 16 to line 18: “Finally, MCF7- $MnSODK68Q$ cells exhibited similar growth characteristic when exposed to either estrogen (Supplemental section, Fig. S2c,d) or Tam (Supplemental section, Fig. S2e,f)”.

Concern # 5 - *Fig 3E. What assay was used to measure transformation? The table is not an adequate illustration of this experiment. In addition, the figure legend states this was an immortalization assay, not transformation, therefore, it appears the authors conflate immortalization with transformation.*

We agree that we were not as precise in the text of our original submission as we should have been when describing the difference between immortalization and transformation. As described above (concern 2b), we have changed quite a bit of text addressing this issue in our revised submission. In

addition, we have added additional data on MnSOD^{-/-} MEFs infected with lenti-empty vector, lenti-MnSOD^{WT}, lenti-MnSOD^{K68Q}, or lenti-MnSOD^{K68R}, and these experiments showed that cells expressing MnSOD^{K68Q} exhibited a more *in vitro* transformed phenotype, as measured by growth in soft agar, contact inhibition, doubling time, and xenograft growth. This data is presented in the new Supplemental Figure S3a-d, and new text has been added describing these experiments on page 8, line 21 to page 9, line 3 (see concern 2b above for quoted text).

Concern # 6 - *The levels of ER need to be assessed in cells expressing MnSODK68Q- as downregulation of ER could be a mechanism by which MCF7 and T47D cells become less sensitive to Tam as well as are able to grow in mice in the absence of estrogen.*

As the reviewer correctly points out, the process of identifying a more rigorous mechanistic link between MnSOD^{K68Q} and Tam resistance should begin with the regulation and/or dysregulation of the ER and its downstream targets. To address this concern, we have attached these results to the revision letter (see figure to the right). The MCF7-HTR and MCF7-MnSOD^{K68Q} cells were harvested and immunoblotted with the anti-ER and actin antibodies. However, in this regard, we would, very respectfully, request that the reviewers allow us to use these results in our next manuscript that we

are working on. The current manuscript is almost 100 data panels, and we expect that the downstream mechanism by which MnSOD^{K68Q} directs changes in ER signaling will be complex. I hope the reviewers will also agree this work could be included in our next submission. Thank you for considering this request.

Concern # 7 - *Many studies have already shown that hydroxy-Tam resistant MCF7 cells form more aggressive in vivo xenograft tumors, so this is more of a control experiment than one that directly links high levels of MnSOD-K68-Ac with Tam resistance. The experiment that needs to be done is to inject hydroxy-Tam resistant cells with an inducible SIRT3 or inducible MnSOD-K68R with Tamoxifen and show that upon induction the tumors stop growing and are sensitized to anti-estrogens.*

We agree that using an inducible expression of MnSOD^{K68R} to stop Tam resistant MCF7 cells from forming more aggressive *in vivo* xenograft tumors is essential. To address this concern, the MCF7-HTR cells were infected with lenti-Tet-DualOn (Clontech) and selected with puromycin, followed by infection with lenti-Tre-Dual2-MnSOD^{K68R} and selection with hygromycin, and then validated for MnSOD-K68R induction, which is shown in Supplemental Figure S9a,b. The subsequent xenografts were grown to 100 mm³, and mice were exposed to doxycycline to induce MnSOD^{K68R} expression. These *in vivo* results are consistent with our *in vitro* results showing that enforced expression of MnSOD^{K68R} prevents the continued proliferation of the MCF7-HTR cells. Text describing these results is presented on page 15, line 11 to line 18: “Finally, the MCF7-HTR cells were used to construct a Tet-On expression system for the inducible expression of the deacetylation mimic mutant (MnSOD^{K68R}). As such, MCF7-HTR cells were initially infected with pTet-DualOn (Clontech) and selected with puromycin, followed by infection with pTre-Dual2-MnSOD^{K68R} and hygromycin selection, and finally, these cells were validated for MnSOD^{K68R} Tet-induction (Supplemental Fig. S9a,b). MCF7-HTR-Dual2-MnSOD^{K68R} xenografts were grown to 100 mm, and mice were exposed to doxycycline to induce MnSOD^{K68R} expression. These experiments showed that enforced expression of MnSOD^{K68R} inhibited *in vivo* MCF7-HTR xenograft tumor cell growth (Fig. 8c)”.

In closing, the authors would like to thank each reviewer for their comments, which have very significantly improved the overall quality of the manuscript, as well as strengthened the *in vitro* and *in vivo* genetic, biochemical, and physiological conclusions.

Reviewers' comments:

Reviewer #1 (Remarks to the Author):

The authors have addressed the issues raised by this reviewer. I recommend its publication.

Reviewer #2 (Remarks to the Author):

Dr. Gius and co-workers submitted the revision to the manuscript "Lysine 68 acetylation regulates the dichotomous role of MnSOD: a protective tetrameric detoxification complex versus a monomeric oncogene". Overall, Zhu et al. worked quite a lot on the manuscript and it from my point of view improved.

Let me stress that the pure fact that they included now even more panels in the figures is per se no sign for improved quality. Moreover, they throughout say MnSOD K68Q-or the acetylated protein-is an oncogene. But does this mutation, i.e. MnSOD K68Q, really occur under disease conditions? Maybe I missed that. If not, from my point of view the authors cannot call a post-translational modification such as lysine-acetylation an oncogene.

Zhu et al. performed additional experiments that partly support their hypotheses. However, I am still not completely convinced that these data proof the mechanism they postulate underlying the observations that MnSOD K68Q mutation has on the phenotypes observed, i.e. acetylation at K68 affects MnSOD peroxidase versus superoxide dismutase activity via its impact on the oligomeric state (monomer versus tetramer). The authors prepared site-specifically K68 lysine-acetylated MnSOD protein by using an amber-suppression technology. This is from my point of view a good way to proof their model. But they only determined the peroxidase activity and compared it to the non-acetylated MnSOD prepared in the same way. Without also showing how the superoxide dismutase activity is affected by K68-acetylation I cannot recommend publication of this manuscript.

Along this line, the authors used 50 kDa cutoff ultracentrifugation units to show how acetylation affects the oligomeric state of MnSOD. However, if MnSOD exists in a monomer-tetramer equilibrium that is not predominantly lying on one site it is not surprising that they observe protein in the flow-through of both devices, namely <50 kDa and >50 kDa cutoff filters independently of the modification. The authors have to show by analytical size-exclusion chromatography that acetylation of MnSOD at K68 affects the oligomeric state of the enzyme. If they are not able without any doubt to show that acetylation of MnSOD at K68 affects the monomer or tetramer formation, the model presented cannot be correct. To this end, these experiments have to be performed before publication. It is from my point of view hard to understand why the authors did not perform these experiments if they have the proteins, i.e. non-acetylated MnSOD and K68-acetylated MnSOD, available. The experiments shown by Zhui eat al. using the filtration units are not convincing.

Finally, the authors should analyse the MnSOD K68-acetylated protein concerning the deacetylation by Sirt3 to draw conclusions about the regulation of MnSOD acetylation under physiological conditions.

Overall, the manuscript improved during the revision. However, these points need to be concisely addressed prior to publication by performing the experiments as written here. Without these data I do not recommend publication of this manuscript. The authors clearly show that there is an impact of MnSOD K68Q mutation on cancer cell development. However, the mechanism underlying this is not concisely explained and it might be the pure mutation-that does not occur in vivo - that by an unknown mechanism results in the phenotypes observed. The authors have to show that acetylation at K68 in MnSOD is doing what they propose on the protein: acetylation at K68 drives MnSOD monomer formation and shows predominantly peroxidase activity and deacetylation of MnSOD K68 drives tetramerisation and makes MnSOd a predominant superoxide dismutase. It is central to the physiological significance and the conclusions drawn that the recombinantly expressed and acetylated MnSOD protein behaves as postulated (monomer versus tetramer and SOD activity versus peroxidase activity). If this is not the case, the model presented cannot be valid and must be reevaluated.

Reviewer #4 (Remarks to the Author):

The authors have comprehensively addressed all of the concerns -- both stylistic and substantive -
- raised in the prior round of review, often doing so through the inclusion of significant new
experimental results where needed. At this stage, I support publication of the manuscript without
further revisions.

Dear Referees,

We are very appreciative of your thoughtful reviews, as well as reviewer number two who directed specific experiments to address her/his concerns. In this regard, the editor has requested that three specific scientific issues be addressed including: (1) Please discuss whether the MnSOD-K68Q mutation/acetylation of MnSOD occurs under disease conditions or as stated by reviewer number 2 “If not, from my point of view the authors cannot call a post-translational modification such as lysine-acetylation an oncogene”. We agree that calling a post-translational modification acetylation mimic mutant an “oncogene” is not the correct scientific manner to present this data. As such, changes in the text have been made to more precisely describe the function of MnSOD acetylation mimic mutant from an “oncogene” to a “tumor promoter”. In addition, text has been added to the discussion outlining literature suggesting a role for the MnSOD-K68-Ac signature in human disease. (2) Please provide data how superoxide dismutase activity is affected by acetylation. To address this we have one new panel to the figure 4 (Fig. 4e) that measure dismutase activity from MnSOD-K68-Ac and wild-type MnSOD isolated from bacteria purified using size exclusion chromatography. (3) Please provide the requested size exclusion chromatography runs. To address this we have added a new Supplemental Figure S5 containing four new panels (5a,b,c,d) that show the chromatograms for bacteria expressing MnSOD-WT and MnSOD-K68-Ac isolated using a Superdex GL column. The eluted fractions were then subjected to further analysis, as shown in the new figures 4e,f.

Reviewer #2, General Comments:

Dr. Gius and co-workers submitted the revision to the manuscript “Lysine 68 acetylation regulates the dichotomous role of MnSOD: a protective tetrameric detoxification complex versus a monomeric oncogene”. Overall, Zhu et al. worked quite a lot on the manuscript and it from my point of view improved. Zhu et al. performed additional experiments that partly support their hypotheses. However, I am still not completely convinced that these data proof the mechanism they postulate underlying the observations that MnSOD K68Q mutation has on the phenotypes observed, i.e. acetylation at K68 affects MnSOD peroxidase versus superoxide dismutase activity via its impact on the oligomeric state (monomer versus tetramer). The authors prepared site-specifically K68 lysine-acetylated MnSOD protein by using an amber-suppression technology. This is from my point of view a good way to proof their model. But they only determined the peroxidase activity and compared it to the non-acetylated MnSOD prepared in the same way. Without also showing how the superoxide dismutase activity is affected by K68-acetylation I cannot recommend publication of this manuscript.

Specific Comments:

Concern # 1 - *Please discuss whether the K68Q mutation/acetylation of MnSOD occurs under disease conditions. In this regard, the reviewer states “Moreover, they throughout say MnSOD K68Q-or the acetylated protein-is an oncogene. But does this mutation, i.e. MnSOD K68Q, really occur under disease conditions? Maybe I missed that. If not, from my point of view the authors cannot call a post-translational modification such as lysine-acetylation an oncogene”.*

In this regard, we do not see K68Q mutations in disease conditions, but it has been shown in recent publications that the MnSOD-Ac signature is associated with some diseases. To address this issue we have added a new sentence to the discussion as well as several additional references. This new text is on page 17, line 20 to line 22 stating “In this regard, several publications in the last few years have shown a connection between disruption of the MnSOD-Ac axis and human illnesses⁴, including aging³³, neurodegeneration³⁴, cardiovascular disease³⁵, and insulin resistance³⁶”.

In addition, I also think the reviewer is suggesting that we should not call a lysine post-translational modification an oncogene. As such, to address this we have changed the manuscript to remove the text suggesting that *MnSOD^{K68Q}* is an oncogene and replaced this text with the phrase “MnSOD^{K68Q}, a site directed mutant that genetically mimics K68-Ac, may function as a tumor promoter” on page 5, line 15 to line 16 and page 8, line 18 to line 19. In addition, in other sections of the manuscript we have changed “oncogene” to “tumor promoter” on: (1) page 2, line 10; (2) page 4, line 18; (3) page 5, line 7; (4) page 6, line 8; (5) page 9, lines 2, 5, and 9; (6) page 12, line 10; (7) page 17, line 19. Finally, we have changed the last word of the title to “tumor promoter”. I believe this is what the reviewer had in mind.

Concern # 2 - *Please provide data how superoxide dismutase activity is affected by acetylation.*

In order to address this concern, we used a bacterial protein expression system that allows the site-specific incorporation of N-(ε)-acetyl-l-lysine into K68 site of MnSOD (see concern 3 below). As such, we have added one new data panel to figure 4 (Figs. 4e) that shows the superoxide dismutase activity for MnSOD-K68-Ac (see right two bars). Text to address this has also been added to page 11, line 11 to 13 “In contrast, bacterial cells expressing pET21a-MnSODK68TAG and pEVOL-AcKRS (peak number 4) exhibited minimal superoxide activity (Fig. 4e, right bar) and significant peroxidase activity (Fig. 4f, right bar)”.

Concern # 3 - *Please provide the requested size exclusion chromatography runs.*

In order to address this concern, the overexpressed protein from bacterial transformed with WT MnSOD and pET21a-MnSOD^{K68TAG} plasmids were lysed and separated using dual nickel / Superdex Fast Protein Liquid Chromatography (FPLC). Bacteria expressing WT MnSOD that were lysed and separated using FPLC showed a peak (number 2) corresponding to 90 kDa (Supplemental Fig. S5a,b). Identical experiments using pET21a-MnSOD^{K68TAG} expressing bacteria showed a peak (number 4) corresponding to 23 kDa (Supplemental Fig. S5cd).

We have also added a new paragraph to the manuscript from page 11, line 1 to line 13 describing this additional data stating “The bacterially expressed protein from the control (i.e., expressing wild-type MnSOD) and pET21a-MnSODK68TAG plasmids was purified by nickel affinity followed by size exclusion chromatography (SEC). Purified protein from bacteria expressing wild-type MnSOD showed SEC elution peak 2 roughly corresponding to 90 kDa consistent with the size

of the tetramer (Supplemental Fig. S5a), which was enriched for MnSOD as measured by Coomassie Blue staining (Supplemental Fig. S5b). Identical experiments using protein from pEVOL-AcKRS and pET21a-MnSODK68TAG expressing bacteria showed an SEC peak (number 4) roughly corresponding to 23 kDa consistent with the size of the monomer (Supplemental Fig. S5c), which was also enriched for MnSOD (Supplemental Fig. S5d). Purified protein samples from the bacteria cells expressing pET21a-MnSODWT (peak number 2), showed significant superoxide dismutase activity (Fig. 4e, left bar) with minimal peroxidase activity (Fig. 4f, left bar). In contrast, bacterial cells expressing pET21a-MnSODK68TAG and pEVOL-AcKRS (peak number 4) exhibited minimal superoxide activity (Fig. 4e, right bar) and significant peroxidase activity (Fig. 4f, right bar)".

In closing, the authors would like to thank reviewer two for his/her comments, which have very significantly improved the overall quality of the manuscript, as well as strengthened the *in vitro* and *in vivo* genetic, biochemical, and physiological conclusions.

Reviewer Comments to the article “Lysine 68 acetylation regulates the dichotomous role of MnSOD: a protective tetrameric detoxification complex versus a monomeric tumor promotor”

Dr. Guis et al submitted the revision of the manuscript “Lysine 68 acetylation regulates the dichotomous role of MnSOD: a protective tetrameric detoxification complex versus a monomeric tumor promotor”. It is an improvement that the authors used the amber-suppression technology to produce acetylated MnSOD protein. Anyways, I am still not convinced that the data really show what the authors main message is, i.e. acetylation at K68 drives monomerisation of MnSOD and thereby switching of the enzymatic activity from superoxide-dismutase to peroxidase activity. From my point of view, the data do not inconsistently show the validity of the model. Therefore, I cannot recommend publication of the manuscript in the current state. The authors have to work on the following points.

Comment 1:

The size-exclusion chromatography for non-acetylated MnSOD and K68-acetylated MnSOD were performed on a S75 10/300 column and therefore unsuitable to perform these assays. If expecting a tetramer, i.e. about 90 kDa, an S200 10/300 column would be better. The first peak is clearly running in the void volume of the column and it is not at all separated from peak 2. This could of course affect the activity of the protein in peak 2. The authors should repeat these experiments on a suitable SEC column (S200 10/300). How did the authors calculate the molecular weight? Did they perform a calibration of the column? Can the authors show this? How were the sequences of the purified proteins? Are they exactly the same for non-acetylated and acetylated mnSOD? Please provide the exact sequence, including purification tags etc.

Comment 2:

The authors should provide quality checks for the purified proteins. First, they should provide an immunoblotting using an anti-acetyl-lysine antibody for acetylated and non-acetylated MnSOD to see if it is acetylated. Second, they

should determine the molecular mass for both proteins by ESI-MS. Third, they should analyse both proteins by LC-MS/MS, i.e. tryptic digest followed by LC-MS/MS, to show the correct site of acetyl-lysine incorporation.

Comment 3:

Please name the column used for the size-exclusion chromatography (SEC) runs at least in the figure legend. Label the axes in the graph and show the elution volumes of the peaks. Are the peaks at the absorption at 280 nm of similar height? If not, the signal to noise ratio is different and it can be assumed that different amounts of proteins were loaded onto the column. Of course, this could affect the monomer-dimer equilibrium.

Comment 4:

In the graph of the SEC runs for MnSOD, peak 4 is labeled in the wildtype run. I cannot see any peak there. Instead it is the tailing of peak 3 what the authors look at. At which elution volume does peak 3 elute? According to the authors peak 3 is in between dimer and monomer, isn't it? The authors see peak 4 for acetylated MnSOD. To judge how much this fraction is with respect to the total protein, the authors should label the axes and show the relative absorption units at A280. How is the activity of peak 3 for both proteins?

Comment 5:

Along this line, if peak 4 for both proteins (both being monomers!!!) show different enzymatic activities, the model presented cannot be valid that the monomerisation itself is the driving force for switching it from superoxide-dismutase to peroxidase activity as both are monomers. Instead, the K68-acetylation itself must be the reason and the mechanism underlying this enzymatic switch is different from what the authors postulate. The exact mechanism needs further investigation. From my point of view this questions the whole message of the paper and the authors should rewrite it as the title etc. is not fitting to the data.

Comment 6:

The authors show the raw data, i.e. the Coomassie gel, only for the purification of wildtype MnSOD. Please include also the raw data for the acetylated MnSOD. I am asking myself why the authors only show these data for wildtype MnSOD? Is something wrong for acetylated MnSOD? Specifically, it would be good to see how pure the protein is over the whole molecular weight range and also if there is significant translational truncation product for the acetylated MnSOD, due to low incorporation efficiency.

Comment 7:

The authors say that MnSOD AcK122 is a Sirt3 target. There are papers showing that MnSOD AcK122 is not a direct substrate for Sirt3. Please write that a bit more carefully.

Comment 8:

The labeling of the SDS-PAGE gel for the SEC runs is not fitting the lanes (figure S5 b).

Dear Referee,

We are grateful for allowing our article to be considered for resubmission. I think it correct to write that the biochemical experiments in our last submission did not represent the quality of work consistent with the standards for *Nature Communications* and we have “gone back to the drawing board” to redo these experiments. As the corresponding author I take responsibility for this and I will outline below the issues with the experimental design and how it has been changed in this submission. Finally, I think it accurate to state the primary concern from our last resubmission was the biochemical data that was not rigorous and did not address reviewer number two’s primary concerns. As such, this will be the primary focus of resubmission and the new data added to the manuscript.

Reviewer #1, Specific Comments:

Concern # 1 - *The size-exclusion chromatography for non-acetylated MnSOD and K68-acetylated MnSOD were performed on a S75 10/300 column and therefore unsuitable to perform these assays. If expecting a tetramer, i.e. about 90 kDa, an S200 10/300 column would be better. The first peak is clearly running in the void volume of the column and it is not at all separated from peak 2. This could of course affect the activity of the protein in peak 2. The authors should repeat these experiments on a suitable SEC column (S200 10/300). How did the authors calculate the molecular weight? Did they perform a calibration of the column? Can the authors show this? How were the sequences of the purified proteins? Are they exactly the same for non-acetylated and acetylated MnSOD? Please provide the exact sequence, including purification tags, etc.*

The first issue was, due to my miscommunication to our research team and our new biochemical collaborators, we did not use the proper manuscripts, and the methods outlined, to conduct the requested biochemical experiments. We mistakenly used the methods from the research group that provided the bacterial expression vectors to us (Li et al., *Antioxid. Redox Signal.* 2015 and 2017) instead of the manuscript suggested by reviewer number two to guide our experiments methods: (1) de Boor et al., *Proc. Natl. Acad. Sci.*, 2015, (2) Knyphausen et al., *J Biol Chem*, 2016; and (3) Lammers, *Methods Mol Biol.* 2018. In addition, the second issue was the column used and as such, to address this a new Superdex 200 Increase 10/300 GL column (GE Healthcare, Product # GE28-9909-44) was purchased and all the experiments in this revision were done with this column.

As such, using the new Superdex 200 Increase 10/300 GL column the first experiments conducted was to isolate wild-type MnSOD from bacteria carrying pET21a-MnSOD^{WT}, using the specific methods, as outlined in the three manuscripts mentioned above. These experiments showed a single

band with a clear peak shown at an elution volume of roughly 13.75 ml, which appears to be very close to the expected size of the tetrameric MnSOD peak (i.e., roughly 92 kDa), which is very similar to the data shown in figure 1B, as shown in Knyphausen et al., 2016. The 92 kDa size was determined by performing a calibration of the column using a Gel Filtration Cal Kit High Molecular Weight GE Healthcare, 28-4038-42 and Low Molecular Weight GE Healthcare, 28-4038-41 kit, which was run on the Superdex 200 Increase 10/300 GL column. These results are presented directly under supplemental Figure 5b. The results for the bacteria expressed MnSOD-K68-WT are presented in figure 4e as peak 1 / blue peak with the y-axis as “Normalized Absorbance (280 nm)”, as suggested by the reviewer. In addition, the raw data is also presented in Supplemental Fig. S5a, which shows the raw chromatogram data with the y-axis as mAU (280 nm) as a means to show that the bands for MnSOD-K68-WT and MnSOD-K68-Ac were of somewhat different peak sizes, which is likely due to the slight less protein used in the two runs (5.5 mg versus 4.8 mg). However, we can change this if the reviewer would prefer both sets of data with the same y-axis.

Similar biochemical experiments were also done with recombinant protein expressed in bacteria carrying pEVOL-AcKRS and pET21a-MnSOD^{K68TAG}, isolated per the manuscripts cited above, and subsequently run on the Superdex 200 column. The chromatogram showed a peak estimated as 25 kDa, using the protein standards outlined directly above, and this data is presented in figure 4e as peak 2 / red peak as well as presented as “Normalized Absorbance (280 nm)” to match the data presented in figure 1B in Knyphausen, 2016. The raw data for these experiments is also presented in Supplemental Section, Fig. S5b and the standard curve is presented directly below.

To address the new data discussed above, this text has been added to the manuscript (in blue font) starting on page 11, line 1 to line 9 stating

The bacterially expressed proteins from the control (carrying pET21a-MnSOD^{WT}) and acetylated form (carrying pET21a-MnSOD^{K68TAG}) were purified by nickel affinity columns followed by size exclusion chromatography (SEC), as previously published^{1, 2, 3}. Purified protein from bacteria expressing wild-type MnSOD eluted at a volume roughly corresponding to 92 kDa (Fig. 4e, peak 1 / blue peak) on SEC consistent with the size of its known homotetrameric complex (Supplemental Fig. S5a, full chromatogram), as shown by others (Knyphausen et al., 2016)¹³. Purified protein from bacteria carrying pEVOL-AcKRS and pET21a-MnSOD^{K68TAG} eluted at a volume consistent with the monomeric form of MnSOD (Fig. 4e, peak 2 / red peak) roughly corresponding to 25 kDa (see Supplemental Fig. S5b, full chromatogram).

Lastly, a concern was raised in regards to how *were the sequences of the purified proteins? Are they exactly the same for non-acetylated and acetylated MnSOD? Please provide the exact sequence, including purification tags, etc.* I believe this is referring to the amino acid sequence of both bacterially expressed pET21a-MnSOD^{WT} and pET21a-MnSOD^{K68TAG}.

		25	30	40	50	60
MnSOD-K68-WT		MKHS	LPD	LPYDYG	ALEP	HINAQIMQLH HSKHHAAYVN
MnSOD-K68-AC		MKHS	LPD	LPYDYG	ALEP	HINAQIMQLH HSKHHAAYVN
		70	80	90	100	
MnSOD-K68-WT		NLNVTEEKYQ	EALAKGDVTA	QIALQPALKF	NGGGHINHSI	
MnSOD-K68-AC		NLNVTEEKYQ	EALAKGDVTA	QIALQPALKF	NGGGHINHSI	
		110	120	130	140	
MnSOD-K68-WT		FWTNLS	PNNGG	GEPK	GELLEA	IKRDFGSFDK FKEKLTAASV
MnSOD-K68-AC		FWTNLS	PNNGG	GEPK	GELLEA	IKRDFGSFDK FKEKLTAASV
		150	160	170	180	
MnSOD-K68-WT		GVQGS	GWGWL	GFNK	ERHGLQ	IAACPNDPL QGTTGLIPLL
MnSOD-K68-AC		GVQGS	GWGWL	GFNK	ERHGLQ	IAACPNDPL QGTTGLIPLL
		190	200	210	220	
MnSOD-K68-WT		GIDV	WEHAYY	LQYKN	VRPDY	LKAIWNVINW ENVTERYMAC
MnSOD-K68-AC		GIDV	WEHAYY	LQYKN	VRPDY	LKAIWNVINW ENVTERYMAC
		222				
MnSOD-K68-WT		KKHHHHHHH				
MnSOD-K68-AC		KKHHHHHHH				

As discussed below (Concern # 2) these proteins, after isolation, were sent for mass spectrometry to confirm for enrichment of K68-Ac. In addition, the sequencing for these two bacterially expressed proteins were also used to

confirm that they contain identical amino acid sequences (see above), except for K68-Ac levels (K68 in red text).

Concern # 2 - *The authors should provide quality checks for the purified proteins. First, they should provide an immunoblotting using an anti-acetyl-lysine antibody for acetylated and non-acetylated MnSOD to see if it is acetylated. Second, they should determine the molecular mass for both proteins by ESI-MS. Third, they should analyse both proteins by LC-MS/MS, i.e. tryptic digest followed by LCMS/MS, to show the correct site of acetyl-lysine incorporation.*

Two samples corresponding to peak 1 / blue peak (i.e., elution volume 13 and 14 ml) and peak 2 / red peak (i.e., elution volume 16 and 17 ml) were first analyzed to confirm MnSOD (Figs. 4f, g) by immunoblotting with an anti-MnSOD antibody (top panels) as well as by using Coomassie Brilliant Blue staining (bottom panels). In addition, these samples were also analyzed using mass spectrometry and the raw data / representative spectra for MnSOD-K68-WT and MnSOD-K68-Ac protein are shown (Supplemental Fig. S5c,d). The data from the mass spectrometry, showing the data for the non-acetylated and acetylated MnSOD analyzed via mass spectrometry, is presented in a new Supplemental Fig. S5e which is a table that shows (i) protein identification probability; (ii) number of unique peptides; (iii) number of unique spectra; (iv) number of total spectra; and (v) percentage of lysine 68 acetylation from bacteria expression pET21a-MnSOD^{WT} and pET21a-MnSOD^{K68TAG}. Finally, these samples peak 1 / blue peak (i.e., elution volume 13 and 14 ml) and peak 2 / red peak (i.e., elution volume 16 and 17 ml) were also immunoblotted with our anti-MnSOD-K68-Ac antibody and these results also confirm for the enrichment of MnSOD-K68-Ac protein (see Supplemental Fig. S5f). As such, it is proposed that this data confirms the enrichment of MnSOD-K68-Ac in the bacteria expressing both pEVOL-AcKRS and pET21a-MnSOD^{K68TAG} and eluted at volumes 16 and 17.

To address the new data discussed above, this text has been added to the manuscript (in blue font) starting on page 11, line 10 to line 17 stating

Prior to further analysis, two eluted fractions corresponding to peak 1 (elution volumes 13 and 14 ml) and peak 2 (elution volumes 16 and 17 ml) were analyzed to confirm MnSOD. Immunoblotting (Fig. 4f, top panel) and Coomassie staining (bottom panel) for purified wild-type bacterial expressed protein confirmed the presence of MnSOD. Similar experiments also confirmed the presence of MnSOD in bacteria carrying pET21a-MnSOD^{K68TAG} (Fig. 4g, top and bottom panel). These samples were also analyzed via mass spectrometry (Supplemental section, Fig. S5c,d) and by staining with the anti-MnSOD-K68-Ac antibody (Supplemental section, Fig. S5e) confirming that the peak 2 is enriched for MnSOD-K68-Ac protein.

Concern # 3 - *Please name the column used for the size-exclusion chromatography (SEC) runs at least in the figure legend. Label the axes in the graph and show the elution volumes of the peaks. Are the peaks at the absorption at 280 nm of similar height? If not, the signal to noise ratio is different and it can be assumed that different amounts of proteins were loaded onto the column. Of course, this could affect the monomer-dimer equilibrium.*

The legend for figure four has also been changed and new text has been added to outline the new experiments as well as name the column used (GE Healthcare, Product # GE28-9909-44) for the size-exclusion chromatography (see Figure legend 4e-i) starting on page 33, line 19 to line 28 stating

(e) *E. coli* were co-transformed with either pEVOL-AcKRS and pET21a-MnSOD^{K68TAG} or pET21a-MnSOD^{WT} and cells were lysed, and the extracts were purified using an equilibrated Ni²⁺-NTA

ÄKTA FPLC Purifier system with GE HisTrap HP column, concentrated using Amicon centrifugal filter, and subsequently run over a Superdex 200 Increase 10/300 GL column (GE Healthcare, Product # GE28-9909-44), as previously described^{1,2}.

(f,g) Elution volumes 13 and 14 ml, corresponding to peak 1 / blue peak **(f)** and elution volumes 16 and 17 ml, corresponding to peak 2 / red peak, were separated by SDS-PAGE, and immunoblotted with anti-MnSOD antibody (top panels) or stained with Coomassie Brilliant Blue (bottom panels). Representative images are shown.

(h,i) Peak 1 (elution volumes 13 and 14 ml) and peak 2 (elution volumes 16 and 17 ml) were analyzed for **(h)** superoxide dismutase activity **(i)** and peroxidase activity. Activity is the average of both samples. All experiments were done in triplicate. Errors represent ± 1 SEM. *** $p < 0.001$.

Finally, the raw data for the size-exclusion chromatography is included in Supplemental Fig. S5a,b to show that they peaks are of similar height. For MnSOD-K68-WT the peak is roughly 150 mAU (280 nm) and for MnSOD-K68-Ac it is roughly 120 (280 nm) and I think this is due to the slight difference in the protein run (5.5 mg versus 4.8 mg) with the Superdex 200 Increase 10/300 column.

Concern # 4 - *In the graph of the SEC runs for MnSOD, peak 4 is labeled in the wild-type run. I cannot see any peak there. Instead it is the tailing of peak 3 what the authors look at. At which elution volume does peak 3 elute? According to the authors peak 3 is in between dimer and monomer, isn't it? The authors see peak 4 for acetylated MnSOD. To judge how much this fraction is with respect to the total protein, the authors should label the axes and show the relative absorption units at A280. How is the activity of peak 3 for both proteins?*

I think it safe to say that with the use of the correct methods, and the manuscripts outlining these methods, that this issue has been addressed and we now have specific peaks corresponding to the tetramer (~92 kDa) and the monomer (~25 kDa). In addition, in figure 4e we have labeled the y-axis as “Normalized Absorbance (280 nm)”. Finally, in the Supplemental Section, Figs. S5a,b we have also presented the raw data as “absorption units at A280”.

Concern # 5 – *Along this line, if peak 4 for both proteins (both being monomers!!!) show different enzymatic activities, the model presented cannot be valid that the monomerisation itself is the driving force for switching it from superoxide dismutase to peroxidase activity as both are monomers. Instead, the K68-acetylation itself must be the reason and the mechanism underlying this enzymatic switch is different from what the authors postulate. The exact mechanism needs further investigation. From my point of view this questions the whole message of the paper and the authors should rewrite it as the title etc. is not fitting to the data.*

Since we have now performed these experiments using the correct methods, as outlined by the three manuscript outlined above, we now only observe a one peak for both proteins from bacteria expressing either pET21a-MnSOD^{K68TAG} or pET21a-MnSOD^{WT}. However, I think the main issue is the underlying mechanism of the switch from a superoxide dismutase to a peroxidase. In this regard, in the last few months we have identified several manuscripts, both in bacteria and yeast, showing that MnSOD can also bind iron and this binding inhibits superoxide activity^{4, 5, 6}, i.e., the binding between manganese and iron can be exchangeable based on context. In addition, a research group from the National Institute of Environmental Health Sciences has shown that when “bacterial manganese-dependent SOD is bound to iron (FeSOD) it can exhibit a peroxidase activity”⁷.

In addition to this molecular dynamics data, we also have some very preliminary molecular biology / biochemical data suggesting that monomeric MnSOD peroxidase activity requires the iron binding

to increase its peroxidase activity, suggesting, but clearly not providing causative data, that MnSOD-K68-Ac may require a potential change from a manganese binding protein to an iron binding protein. Therefore, I would very respectfully ask that this new data, and the subsequent story that more rigorously defines the mechanism for the peroxidase activity, be included in our next manuscript.

Finally, we have added two new figures to present the superoxide dismutase and peroxidase data for Peak 1 / blue peak versus peak 2 / red peak as shown in Fig. 4h,i. To address the new data discussed above, text has been added to the manuscript starting on page 11, line 19 to line 22 stating

Purified protein samples from the bacteria cells expressing pET21a-MnSOD^{WT} (elution volumes 13 and 14 ml) showed significant superoxide dismutase activity (Fig. 4h, left bar) with minimal peroxidase activity (Fig. 4i, left bar). In contrast, recombinant MnSOD-K68-Ac protein from bacterial cells expressing pET21a-MnSOD^{K68TAG} (elution volumes 16 and 17 ml) exhibited minimal superoxide activity (Fig. 4h, right bar) and significant peroxidase activity (Fig. 4i, right bar).

Concern # 6 – *The authors show the raw data, i.e. the Coomassie gel, only for the purification of wild-type MnSOD. Please include also the raw data for the acetylated MnSOD. I am asking myself why the authors only show these data for wild-type MnSOD? Is something wrong for acetylated MnSOD? Specifically, it would be good to see how pure the protein is over the whole molecular weight range and also if there is significant translational truncation product for the acetylated MnSOD, due to low incorporation efficiency.*

The raw data for the experiments shown in Fig. 4f,g, which show the two fractions from peak 1 (13 / 14) and peak 2 (16 / 17) either immunoblotted with an anti-MnSOD antibody (in color) or stained with Coomassie Blue (in black and white), are shown on the bottom of page 19 of the supplemental section. The data that I think shows ‘how pure the protein is over the whole molecular weight range’ is the Coomassie Brilliant Blue raw data, in addition to the immunoblotting gel for the western blot from for volumes 13 / 14 and 16 /17 immunoblotted with our anti-MnSOD-K68-Ac antibody, as shown in the Supplemental Section, Fig. S5e and the raw data blot, which are in both grey and color backgrounds, are shown on the top of page 20.

We have also added new text to the methods section to outline how these experiments were done starting on page 27, line 4 to line 18 stating

Incorporation and isolation of N-(ε)-Acetyl-Lysine into MnSOD-K68

BL21 (DE3) pMAGIC bacteria were co-transformed with pEVOL-AcKRS, which expresses an acetyl-lysyl-tRNA synthetase/tRNA^{CUA} pair from *M. barkeri*, and pET21a-MnSOD^{K68TAG}, which expresses a site-specific mutation that allows incorporation of N-(ε)-acetyl-l-lysine (AcK) into K68. Cells were grown in 10 mM N-(ε)-acetyl-lysine (Sigma) and 20 mM nicotinamide (Sigma) and protein expression was induced by addition of 200 μM IPTG and cells were grown overnight³. Cells were harvested and lysed as previously described¹ and the cleared lysate was loaded onto an equilibrated Ni²⁺-NTA ÄKTA FPLC Purifier system with GE HisTrap HP columns (Product # GE17524701). After elution, proteins were concentrated using Amicon centrifugal filters (Product # UFC800324) to a concentration of 5.5 mg/mL for MnSOD-K68-Ac and 4.8 mg/mL MnSOD-WT. 200 μL of each eluted protein were run over a Superdex 200 Increase 10/300 GL column (GE Healthcare) (Product # GE28-9909-44), as previously described^{1, 2}. The eluted fractions were then subjected to further analysis. A calibration curve was generated using a gel filtration low and high

molecular weight kit (GE Healthcare) according to the manufacturer's instructions and is shown in Supplemental section, figure S5a,b, which was used to determine the relative size of peak 1 and 2.

Concern # 7 – *The authors say that MnSOD AcK122 is a Sirt3 target. There are papers showing that MnSOD AcK122 is not a direct substrate for Sirt3. Please write that a bit more carefully.*

To address this we have changed this sentence in the introduction to state “While multiple MnSOD acetylation sites have been identified recent publications seem to suggest that K68 is central to the regulation of MnSOD superoxide dismutase activity^{1, 8, 9, 10, 11, 12, 13}.”

Concern # 8 – The labeling of the SDS-PAGE gel for the SEC runs is not fitting the lanes (figure S5 b). This gel has been replaced with new data / gels is now presented as Figs. 4f,g.

In closing, the authors would like to thank the reviewer for these comments, which have significantly improved the overall quality of the manuscript, as well as strengthened this manuscript.

REFERENCES

1. Knyphausen P, *et al.* Insights into Lysine Deacetylation of Natively Folded Substrate Proteins by Sirtuins. *J Biol Chem* **291**, 14677-14694 (2016).
2. Lammers M. Expression and Purification of Site-Specifically Lysine-Acetylated and Natively-Folded Proteins for Biophysical Investigations. *Methods in molecular biology (Clifton, NJ)* **1728**, 169-190 (2018).
3. de Boor S, *et al.* Small GTP-binding protein Ran is regulated by posttranslational lysine acetylation. *Proc Natl Acad Sci U S A* **112**, E3679-3688 (2015).
4. Culotta VC, Yang M, O'Halloran TV. Activation of superoxide dismutases: putting the metal to the pedal. *Biochim Biophys Acta* **1763**, 747-758 (2006).
5. Naranuntarat A, Jensen LT, Pazicni S, Penner-Hahn JE, Culotta VC. The interaction of mitochondrial iron with manganese superoxide dismutase. *J Biol Chem* **284**, 22633-22640 (2009).
6. Kang Y, He YX, Zhao MX, Li WF. Structures of native and Fe-substituted SOD2 from *Saccharomyces cerevisiae*. *Acta Crystallogr Sect F Struct Biol Cryst Commun* **67**, 1173-1178 (2011).
7. Ganini D, Petrovich RM, Edwards LL, Mason RP. Iron incorporation into MnSOD A (bacterial Mn-dependent superoxide dismutase) leads to the formation of a peroxidase/catalase implicated in oxidative damage to bacteria. *Biochim Biophys Acta* **1850**, 1795-1805 (2015).
8. Qiu X, Brown K, Hirshey MD, Verdin E, Chen D. Calorie restriction reduces oxidative stress by SIRT3-mediated SOD2 activation. *Cell metabolism* **12**, 662-667 (2010).
9. Brown K, *et al.* SIRT3 reverses aging-associated degeneration. *Cell Rep* **3**, 319-327 (2013).
10. Chen Y, *et al.* Tumour suppressor SIRT3 deacetylates and activates manganese superoxide dismutase to scavenge ROS. *EMBO Rep* **12**, 534-541 (2011).
11. Vassilopoulos A, *et al.* SIRT3 deacetylates ATP synthase F1 complex proteins in response to nutrient- and exercise-induced stress. *Antioxid Redox Signal* **21**, 551-564 (2014).
12. Tao R, *et al.* Sirt3-mediated deacetylation of evolutionarily conserved lysine 122 regulates MnSOD activity in response to stress. *Molecular cell* **40**, 893-904 (2010).
13. Haigis MC, Deng CX, Finley LW, Kim HS, Gius D. SIRT3 Is a Mitochondrial Tumor Suppressor: A Scientific Tale That Connects Aberrant Cellular ROS, the Warburg Effect, and Carcinogenesis. *Cancer Res* **72**, 2468-2472 (2012).

REVIEWERS' COMMENTS:

Reviewer #2 (Remarks to the Author):

Zhu and co-workers presented in their revision novel data regarding the impact of MnSOD K68-acetylation on its oligomeric state and its enzymatic activity. Honestly, to me it is a bit surprising that the size-exclusion chromatography runs on the S200 10/300 column are now so clear and wildtype MnSOD appears tetrameric as published previously (Knyphausen et al., 2016) and MnSOD AcK68 is exclusively monomeric. However, these data were missing to prove the hypothesis of the manuscript saying that acetylation of MnSOD at K68 results in monomerization of the enzyme. However, these data are not the proof for monomerization being the underlying molecular mechanism for the observed enzymatic switch from a superoxide dismutase activity towards a peroxidase activity. As the authors suggest, the mechanism underlying this switch in the enzymatic activities might be due to the alteration of the binding affinities from manganese in the SOD towards iron in peroxidase. If the monomerization is the driving mechanism for these altered affinities cannot be confirmed by this manuscript. The authors should emphasize this. If they really wanted to show if monomerization itself is the driving force for enzymatic switch, they could make interfacial mutations resulting in monomerization and test this for their enzymatic activities. The authors should also clearly describe how the new size-exclusion chromatography runs were performed including the exact running buffer composition for both acetylated and non-acetylated proteins for which I assume the authors used the same buffer so that other researchers are able to repeat the experiments. If the authors add this missing information and emphasize that the underlying mechanism for the enzymatic switch is not clear, I recommend publication of this manuscript in Nature Communications at this stage.

Dear Referee,

We are grateful for allowing our article to be considered for resubmission. We are very appreciative of your thoughtful reviews, which were mostly supportive, but specifically recommended additional data and changes to the text to improve the overall quality and presentation of the manuscript. In addition, the manuscript has been reworked with additional text to outline the additional experiments in the manuscript to specifically address the concerns for the manuscript.

Specific Comments:

Concern # 1 - *For the SEC in figure 4, please provide a Coomassie staining of the fractions or western blot if there is not enough protein. The Standards of the SEC should be indicated in the graph.*

To address this concern we have separated the retention/elution volumes (11 through 20) from the bacterial expressing MnSOD-K68-WT which were subsequently either stained with Coomassie Blue (Fig. S5a, middle panel) or immunoblotted with an anti-MnSOD antibody (lower panel). Identical experiments were also done with elution volumes 11 through 20 for bacterial isolated MnSOD-K68-Ac with either the Coomassie Blue staining (Fig. S5b, middle panel) or immunoblotting with an anti-MnSOD antibody (lower panel) as the results are shown.

Concern # 2 -*Please update the figure legends, as it is they describe the method, not the result shown. Please describe the results in the Figure legend and move the methods to the methods section.*

To address this concern we have completely rewritten the figure legend for Supplemental figure S5, which is shown in blue font, and is presented near the top of page 13 of the supplement.

Supplemental Figure S5. MnSOD-K68 acetylation exhibits peroxidase activity. BL21(DE3) bacteria were transformed with pET21a-MnSOD^{WT}, or pEVOL-AcKRS together with pET21a-MnSOD^{K68TAG}. Cells were harvested and lysed, and eluted protein were run over a Superdex 200 Increase 10/300 GL column and fractions, as previously described^{2, 3, 4}. A more significant detailed methods section is outlined above in the “Bacterial expression and purification system for MnSOD-K68-Ac and MnSOD-WT” and these samples were subsequently used for further analysis. **(a)** Chromatogram from the size exclusion column of purified protein from bacteria carrying pET21a-

MnSOD^{WT} (top panel), retention volumes fractions 11 through 20 were further analyzed by either Coomassie staining (middle panel) or immunoblotted with anti-MnSOD antibody (lower panel) to confirm MnSOD levels. **(b)** Chromatogram of purified protein from bacteria carrying pEVOL-AcKRS and pET21a-MnSOD^{K68TAG}, all the fractions were further analyzed by either Coomassie staining (middle panel) or immunoblotted with an anti-MnSOD-K68-Ac antibody (lower panel) The raw data are presented with the y-axis as mAU (280 nm) to show that peak 2 is smaller than peak 1 which is likely due to the slightly less protein run on the Superdex 200 Increase 10/300 GL column (5.5 mg vs. 4.8 mg). **(c)** Three separate MnSOD-K68-WT samples were analyzed via mass spectroscopy and 32 exclusive unique peptides, 164 spectra, and 999 total spectra, 100% coverage which is an average of each run. **(d)** Three separate MnSOD-K68-Ac samples showed 24 exclusive unique peptides, 99 unique spectra, 531 total spectra were identified, 95% coverage which is an average of each run. **(e)** Table showing the average percentage of total number of unique K68 acetylated peptides, as a ratio of the total number of unique peptides. The data for total number of unique peptides, unique spectra, and total spectra from bacteria expressing pET21a-MnSOD^{WT} or expressing pET21a-MnSOD^{K68TAG} are also shown. **(f)** Peak 1 (volumes 13, 14 ml) and peak 2 (volumes 16, 17 ml) were separated by SDS-PAGE and immunoblotted with anti-MnSOD-K68-Ac antibody. All experiments done in triplicate. Representative images are shown.

Concern # 3 - Please implement the changes/additions requested by Rev#2 (see below).

Concern #3a - The authors should also clearly describe how the new size-exclusion chromatography runs were performed including the exact running buffer composition for both acetylated and non-acetylated proteins for which I assume the authors used the same buffer so that other researchers are able to repeat the experiments.

To address this important concern we have added a short sentence at the end of the methods section in the manuscript on page 27 starting on line 19 to 21 “For a more complete description of the methods, buffers, and other techniques please see Supplemental Section, Methods, “Isolation of wild-type and N-(ε)-Acetyl-Lysine into MnSOD-K68”, to direct the reader to a more thorough outline of the methods, which we have placed into the supplemental section methods. In this regard, a more detailed outline of the methods has been added to the Supplemental Section starting on page 4, line 23 stating

“Bacterial expression and purification system for MnSOD-K68-Ac and MnSOD-WT

The text below is a more complete description of the methods to isolate the bacterially expressed MnSOD-K68-WT and MnSOD-K58-Ac exogenous protein. In this regard, BL21 (DE3) chemically competent E.coli cells were co-transformed with pEVOL-AcKRS and pET21a-MnSODK68TAG plasmids or pET21a-wtMnSOD to express MnSOD-K68-Ac and MnSOD-WT proteins. The cells harboring pEVOL-AcKRS and pET21a-MnSODK68TAG plasmids were incubated in 100 ml LB with 100 µg/ml Ampicillin and 50 µg/ml Chloramphenicol (37 °C, rpm) for 3 h at 37 °C, and 50 mM nicotinamide (Sigma) was added to this culture. When OD600 reached 1.1, 2 mM Nε-acetyl-lysine (Sigma) was added to the culture and cells were induced by the addition of 0.4 mM IPTG and 0.2% Arabinose (25°C, 180 r/min) for another 20 hours, as described (The bacterial MnSOD expression and lysine acetylation tRNA mutant plasmids were a kind gift from Dr. Jiangyun Wang, Institute of Biophysics, Chinese Academy of Sciences, Beijing, China).

The BL21 (DE3) cells harboring pET21a-wtMnSOD plasmid were cultured in 5 ml LB media with 100 µg/ml Ampicillin and 1 ml of this culture was incubated in 100 ml LB with 100 µg/ml Ampicillin (37 °C, pm) overnight. The next day, 1 L of LB with 100 µg/ml Ampicillin was inoculated with 10 ml of overnight culture, for ~ 2.5 h until OD600 = 0.6, cells were induced with

0.4 mM IPTG (25 °C, pm) overnight as described. All purification steps were performed on ice. E.coli cells in 1 L LB were harvested by centrifugation (6000 g, 10 min, 4 °C) and washed with 50 ml Buffer I containing 20 mM imidazole, 50 mM Tris-HCl, 200 mM NaCl, 5 mM MgCl₂, 50 mM nicotinamide, pH = 8.0. Then pellets after centrifugation were suspended with 50 ml Buffer I supplemented with 1 mM PMSF and approximately 1 mg/ml lysozyme, and the lysates were incubated at 4°C for 10 min. Then protein was extracted by sonication (5 sec on, 6 sec off cycle, 25 min). The extract was clarified by centrifugation (13,000 g, 30 min, 4 °C) and pellet was discarded. 0.2 ml Ni²⁺-NTA beads were added to the supernatant and incubated with agitation at 4 °C for 1 h.

Beads were transferred into a column and washed three times with Buffer I containing increasing imidazole gradient (50, 75, 100mM) and protein was eluted in 1 ml Buffer I supplemented with 200 mM imidazole. The proteins were analyzed by SDS-PAGE and then concentrated using Ultra-15 Centrifugal Filter Unit (10kDa, Millipore Amicon™, USA UFC800324). The eluted protein were then re-buffered to Buffer II (50 mM Tris-HCl, 200 mM NaCl, 5 mM MgCl₂, 50 mM nicotinamide, pH = 8.0) and loaded onto an equilibrated Ni²⁺-NTA ÄKTA FPLC Purifier system with GE HisTrap HP columns (Product # GE17524701) and further purified by Superdex 200 Increase 10/300 GL column (GE Healthcare, Product#GE28-9909-44) in a buffer containing 50 mM potassium phosphate (pH = 7.8). Peak fractions were collect by using an automated fraction collector. A280 as a function of elution volume/time were also recorded^{1, 2, 3}. The peak protein fractions concentrations were determined and immunoblotted with anti-MnSOD and anti-MnSOD-K68-Ac. The remaining purified proteins were measured for peroxidase activity and MnSOD activity.

Concern #3b - *As the authors suggest, the mechanism underlying this switch in the enzymatic activities might be due to the alteration of the binding affinities from manganese in the SOD towards iron in peroxidase. If the monomerization is the driving mechanism for these altered affinities cannot be confirmed by this manuscript. The authors should emphasize this.*

To address this important concern we have added a sentence in the results section of the manuscript on page 12 starting on line 2 to 3 “However, more research is required to definitively identify that monomerization is the underlying molecular mechanism for the enzymatic switch to peroxidase activity.” as well as another paragraph in the discussion section of the manuscript on page 20 starting on line 4 to 11 “In this manuscript we found that acetylation of K68 drives MnSOD conformational changes and induces a significant increase in the monomeric form of MnSOD. In addition, our biochemical tissue culture and bacterial experiments clearly show that MnSOD can, under specific condition, function as a peroxidase, however this not, a priori, causative data that monomerization is the underlying molecular mechanism for the observed enzymatic switch towards a peroxidase. In this regard, several manuscripts in bacteria and yeast suggest, based on context that MnSOD binding between manganese and iron is exchangeable, suggesting this may be, at least in a part, a potential additional mechanism.” to emphasize that the specific mechanism for underlying this switch in the enzymatic activities might be due to the alteration of the binding affinities from manganese in the SOD towards iron in peroxidase is not fully understood and will be studied in the future.

In closing, the authors would like to thank the reviewer for these comments, which have significantly improved the overall quality of the manuscript, as well as strengthened this manuscript.